# Clonal hematopoiesis with JAK2V617F promotes pulmonary hypertension with ALK1 upregulation in lung neutrophils

Yusuke Kimishima [1], Tomofumi Misaka [1,2 ✉], Tetsuro Yokokawa [1,3], Kento Wada[1], Koki Ueda [4], Koichi Sugimoto[1,3], Keiji Minakawa[4], Kazuhiko Nakazato[1], Takafumi Ishida[1], Motohiko Oshima[5], Shuhei Koide[5], Kotaro Shide[6], Kazuya Shimoda [6], Atsushi Iwama [5], Kazuhiko Ikeda [4 ✉] & Yasuchika Takeishi[1]

Pulmonary hypertension (PH) is a progressive cardiopulmonary disease characterized by pulmonary arterial remodeling. Clonal somatic mutations including *JAK2*V617F, the most frequent driver mutation among myeloproliferative neoplasms, have recently been identified in healthy individuals without hematological disorders. Here, we reveal that clonal hematopoiesis with JAK2V617F exacerbates PH and pulmonary arterial remodeling in mice. JAK2V617F-expressing neutrophils specifically accumulate in pulmonary arterial regions, accompanied by increases in neutrophil-derived elastase activity and chemokines in chronic hypoxia-exposed JAK2V617F transgenic (JAK2$^{V617F}$) mice, as well as recipient mice transplanted with JAK2$^{V617F}$ bone marrow cells. JAK2V617F progressively upregulates *Acvrl1* (encoding ALK1) during the differentiation from bone marrow stem/progenitor cells peripherally into mature neutrophils of pulmonary arterial regions. JAK2V617F-mediated STAT3 phosphorylation upregulates ALK1-Smad1/5/8 signaling. ALK1/2 inhibition completely prevents the development of PH in JAK2$^{V617F}$ mice. Finally, our prospective clinical study identified *JAK2*V617F-positive clonal hematopoiesis is more common in PH patients than in healthy subjects. These findings indicate that clonal hematopoiesis with JAK2V617F causally leads to PH development associated with ALK1 upregulation.

[1] Department of Cardiovascular Medicine, Fukushima Medical University, Fukushima, Japan. [2] Department of Advanced Cardiac Therapeutics, Fukushima Medical University, Fukushima, Japan. [3] Department of Pulmonary Hypertension, Fukushima Medical University, Fukushima, Japan. [4] Department of Blood Transfusion and Transplantation Immunology, Fukushima Medical University, Fukushima, Japan. [5] Division of Stem Cell and Molecular Medicine, Center for Stem Cell Biology and Regenerative Medicine, The Institute of Medical Science, The University of Tokyo, Tokyo, Japan. [6] Department of Gastroenterology and Hematology, University of Miyazaki, Miyazaki, Japan. ✉email: misaka83@fmu.ac.jp; kazu-ike@fmu.ac.jp

Pulmonary hypertension (PH) is a complex cardio-pulmonary disease characterized by increases in pulmonary vascular resistance and pulmonary arterial pressure. Despite recent advances in diagnosis and treatment, PH remains a serious condition, eventually leading to right heart failure with high mortality[1]. A pathological feature of PH is structural remodeling of the small pulmonary arteries, which is associated with intimal thickening, muscularization and the formation of plexiform lesions[2]. Bone marrow (BM)-derived progenitor cells, as well as perivascular inflammatory infiltrates, contribute to the process of pulmonary arterial remodeling[3]. It has been also reported that several hematological disorders, including myeloproliferative neoplasms (MPNs), are often complicated with PH[4]. The incidence of PH has been reported to be higher in MPN patients than in the general population, and high mortality due to cardiovascular diseases has been observed in MPN patients with PH[5,6]. PH is categorized into five etiological groups according to the WHO clinical classification[7]. Based on the above observations, MPN-associated PH is classified into WHO Group V, which is an important heterogenous group that encompasses unclear multifactorial mechanisms[7].

MPNs including polycythemia vera (PV), essential thrombocythemia (ET), and primary myelofibrosis (MF) are characterized by chronic proliferation of mature myeloid cells[8], and the myeloproliferative phenotype is driven by somatic mutations in *JAK2*, *CALR*, and *MPL*. Among MPNs, *JAK2*V617F, an activating somatic mutation in *JAK2*, is the most frequently observed driver mutation; it has been observed in over 95% of PV patients as well as 50–60% of ET and primary MF patients[9–11]. *JAK2*V617F causes cytokine-independent activation of the JAK–STAT pathway, resulting in proliferation of mature myeloid cells[11].

Recent advances in genetic analyses have led to the discovery of clonal hematopoiesis, whose hematopoietic stem/progenitor cells harbor somatic mutations in genes often mutated in myeloid cancers, including MPNs, in healthy individuals without any hematologic disorders[12,13]. Among clonal hematopoiesis, age-related clonal hematopoiesis implies the presence of any detectable clonal events in hematopoietic cells, and its incidence increases with age. Clonal hematopoiesis of indeterminate potential (CHIP) is defined by somatic mutations with a variant allele frequency (VAF) of at least 2%. Clonal hematopoiesis is quite common, and more than 15% of individuals are affected at age ≥70 years[14]. Whereas the rate of patients who progress from CHIP to myeloid malignancies is estimated to be only 0.5–1%, patients with CHIP exhibit markedly increased cardiovascular diseases such as atherosclerosis[12,15]. Most frequently mutated genes in clonal hematopoiesis are epigenetic modifiers; *DNMT3A*, *TET2*, and *ASXL1*. *JAK2* is the next most often mutated gene, and the vast majority of these mutants are *JAK2*V617F in clonal hematopoiesis. Murine studies have suggested that CHIP with somatic mutations in epigenetic modifiers, as well as *JAK2*V617F, played causal roles in acceleration of atherosclerosis[16,17]. MPN patients often show venous and arterial vascular complications[18]. In particular, MPN patients with *JAK2*V617F showed higher incidence of vascular complications compared to those with other driver mutations[18]. However, mechanistic relevance of clonal hematopoiesis with *JAK2*V617F in PH has yet to be elucidated.

Herein, we provide the evidence that clonal hematopoiesis with JAK2V617F plays causal roles in the development of PH with ALK1 upregulation in lung neutrophils.

## Results

**JAK2V617F expression accelerates pulmonary hypertension in response to chronic hypoxia exposure in mice.** To know the involvement of the JAK-STAT pathway in PH development, adult wild-type (WT) C57BL/6 J mice were exposed to chronic hypoxia (10% $O_2$), which is a well-established method to induce PH in mice[19,20]. STAT3 phosphorylation levels on whole lung homogenates, not fractionated cells, were significantly increased after exposure to chronic hypoxia for 3 weeks (Supplementary Fig. 1), suggesting that JAK-STAT activation may play a pathophysiological role in chronic hypoxia-induced PH. To clarify the effects of JAK2V617F expression on the pathogenesis of PH, we used JAK2$^{V617F}$ female mice with transgenic expression of *Jak2*V617F[21] after exposure to normoxia or chronic hypoxia. Starting from 2 weeks after chronic hypoxia exposure, we observed noticeable signs of cardio-respiratory distress such as reduced activity, diminished appetite, and piloerection in JAK2$^{V617F}$ mice, but not in WT mice. We determined to analyze the mice at the 2-week point to minimize the secondary alternation for investigation of the molecular mechanisms that cause PH (Fig. 1a). After normoxia exposure, JAK2$^{V617F}$ mice had significantly higher white blood cell and platelet counts, in comparison to WT littermates, indicating an MF-like phenotype in JAK2$^{V617F}$ mice (Fig. 1b), which is consistent with the results of our previous studies[21,22]. Right ventricular systolic pressure (RVSP) and the ratio of right ventricle weight to left ventricle weight plus septum weight (RV/LV + S) did not differ between WT and JAK2$^{V617F}$ mice after normoxia exposure (Fig. 1c). Although chronic hypoxia significantly elevated hemoglobin values in both WT and JAK2$^{V617F}$ mice, there was no significant difference between them. Notably, we found that RVSP was significantly elevated in JAK2$^{V617F}$ mice compared to WT mice in response to continuous hypoxia (Fig. 1c) in line with the echocardiographic evaluation of pulmonary hemodynamics (Supplementary Fig. 2). In addition, RV/LV + S in JAK2$^{V617F}$ mice was significantly greater than that in WT mice, indicating more severe RV hypertrophy due to PH in chronic hypoxia-exposed JAK2$^{V617F}$ mice (Fig. 1c). LV fractional shortening or LV + S values were not different among the groups, suggesting that chronic hypoxia was not associated with LV systolic dysfunction or LV hypertrophy in JAK2$^{V617F}$ mice (Supplementary Figs. 2 and 3). Of note, we found that even male JAK2$^{V617F}$ mice showed significant increases in RVSP and RV/LV + S compared to male WT mice 2 weeks after chronic hypoxia (Supplementary Fig. 4). Considering the clinical relevance of PH patients that women are more likely to be affected than men[23], we thereafter used female mice in a whole series of the present study unless otherwise indicated.

**JAK2$^{V617F}$ mice exhibit pulmonary vascular remodeling accompanied by the increased perivascular neutrophil infiltration in the lungs after chronic hypoxia.** Histological analyses revealed significant increases in medial wall thickness and muscularization of pulmonary vessels in JAK2$^{V617F}$ mice compared to WT mice after exposure to chronic hypoxia (Fig. 1d, e). The numbers of proliferating smooth muscle cells in the pulmonary arteries were significantly increased in JAK2$^{V617F}$ mice compared to WT mice after chronic hypoxia (Supplementary Fig. 5a). These data suggest that the JAK2V617F expression promoted PH with pulmonary arterial structural remodeling in response to chronic hypoxia, rather than spontaneous development of PH under normoxia. We observed increased cellular infiltration surrounding the pulmonary arteries in both normoxia- and chronic hypoxia-exposed JAK2$^{V617F}$ mice in H&E staining (Supplementary Fig. 5b). Next, we characterized the infiltrating cells by immunohistochemical staining. There were significant increases in Ly6G$^+$ neutrophils specifically in pulmonary arterial regions of JAK2$^{V617F}$ lungs compared to WT lungs (Fig. 1f, g), and more Ly6G-expressing cells within CD45$^+$ cells than F4/80$^+$ macrophages or CD45R$^+$ B cells (Supplementary Fig. 6). Of note, the numbers of Ly6G$^+$ cells in the perivascular and non-perivascular

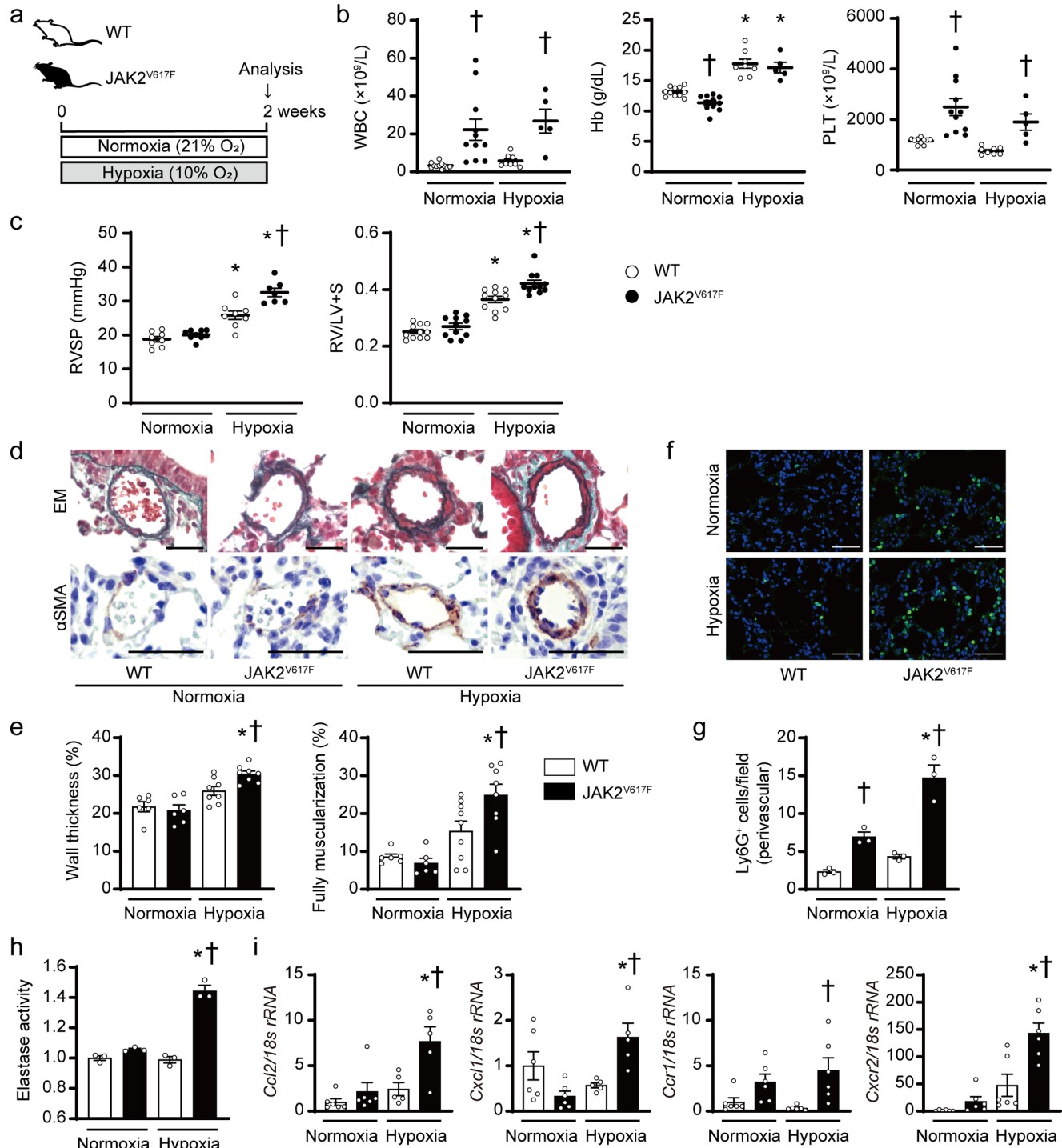

regions of JAK2[V617F] lungs were further increased after chronic hypoxia exposure compared to those after normoxia exposure (Supplementary Fig. 7). CD41[+] megakaryocytes and TER-119[+] erythroblasts were rarely observed in both WT and JAK2[V617F] lungs (Supplementary Fig. 5b). The activity of elastase, which mainly originates from neutrophils[24], and mRNA expression levels of neutrophil-related chemokines and chemokine receptors, including *Ccl2*, *Cxcl1*, *Ccr1*, *Cxcr2*, as well as cytokines such as *Pdgfrb* and *Tgfb1*, were significantly increased in the lungs of JAK2[V617F] mice after chronic hypoxia (Fig. 1h, i, Supplementary Fig. 5c). Thus, the infiltrated neutrophils in perivascular regions accompanied by their increased functional activities might play important roles in pulmonary arterial structural remodeling in

JAK2[V617F] mice. We confirmed that in the Sugen-hypoxia model, which is another PH model[25], RVSP and RV/LV + S were significantly elevated in JAK2[V617F] mice compared to WT mice (Supplementary Fig. 8). There was no statistical significance regarding RVSP and RV/LV + S between aged WT and JAK2[V617F] mice (8–9 months old) without hypoxia stimulus, but some of the aged JAK2[V617F] mice displayed comparatively high RVSP and RV/LV + S (Supplementary Fig. 9).

**Hematopoietic cell clone with JAK2V617F exacerbates the development of pulmonary hypertension in response to chronic hypoxia in mice.** We next investigated whether a hematopoietic cell clone, rather than lung tissue with JAK2V617F

**Fig. 1 JAK2$^{V617F}$ mice accelerate pulmonary hypertension accompanied by perivascular neutrophil infiltration in response to chronic hypoxia.**
**a** Experimental design. Wild-type (WT) mice and mice with transgenic expression of *Jak2*V617F (JAK2$^{V617F}$) exposed to normoxia (21% O$_2$) or hypoxia (10% O$_2$) for 2 weeks were analyzed. **b** Peripheral blood cell counts in WT mice or JAK2$^{V617F}$ mice under normoxia or hypoxia for 2 weeks ($n = 11, 11, 8, 5$, $^\dagger P = 0.0036$ [left], $0.0185$ [right] for WBC, $n = 11, 11, 8, 5$, $^*P < 0.0001$ [left], $< 0.0001$ [right], $^\dagger P = 0.0335$ for Hb, $n = 10, 11, 8, 5$, $^\dagger P = 0.0008$ [left], $0.0396$ [right] for PLT). **c** Right ventricular systolic pressure (RVSP, $n = 8, 9, 8, 7$, $^*P < 0.0001$ [left], $< 0.0001$ [right], $^\dagger P = 0.0002$) and right ventricular hypertrophy determined by the ratio of right ventricle (RV) weight to left ventricle weight plus septum weight (RV/LV + S, $n = 11$ in each group, $^*P < 0.0001$ [left], $< 0.0001$ [right], $^\dagger P = 0.0027$). **d** Representative images of Elastica-Masson (EM)-stained sections and sections immunostained with anti-α–smooth muscle actin (αSMA) antibody from WT mice and JAK2$^{V617F}$ mice. Scale bars, 25 μm. **e** Quantitative analysis of medial wall thickness in EM-stained sections (left, $n = 6, 6, 8, 8$, $^*P < 0.0001$, $^\dagger P = 0.0413$) and the percentage of muscularized distal pulmonary vessels in αSMA-immunostained sections (right, $n = 6, 6, 9, 8$, $^*P = 0.0001$, $^\dagger P = 0.0263$). **f** Representative immunofluorescence images of lung sections stained with anti-Ly6G (green) antibody and DAPI (blue). Scale bars, 50 μm. **g** Quantitative analysis of Ly6G-positive cells in perivascular regions ($n = 3$ in each group, $^*P = 0.0015$, $^\dagger P = 0.0318$ [left], $0.0002$ [right]). **h** Elastase activity in the lungs from WT and JAK2$^{V617F}$ mice. The average value from normoxia-exposed WT mice was set to 1 ($n = 3$ in each group, $^*P < 0.0001$, $^\dagger P < 0.0001$). **i** Relative mRNA expression levels of *Ccl2*, *Cxcl1*, *Ccr1*, and *Cxcr2* in the lungs. The 18 s *rRNA* was used for normalization. The average value from normoxia-exposed WT mice was set to 1 ($n = 6, 6, 5, 5$, $^*P = 0.0049$, $^\dagger P = 0.0105$ for *Ccl2*, $n = 6, 6, 5, 5$, $^*P = 0.0044$, $^\dagger P = 0.0284$ for *Cxcl1*, $n = 6, 6, 6, 6$, $^\dagger P = 0.0139$ for *Ccr1*, $n = 6, 6, 6, 6$, $^*P < 0.0001$, $^\dagger P = 0.0008$ for *Cxcr2*). All data are presented as mean ± SEM. $^*P < 0.05$ versus the corresponding normoxia-exposed group and $^\dagger P < 0.05$ versus the corresponding WT mice by the one-way ANOVA with Tukey post-hoc analysis. WBC white blood cell count, Hb hemoglobin concentration, PLT platelet count. Source data are provided as a Source Data file.

expression, contributes to the development of PH, by means of BM transplantation (BMT)[22]. Donor BM cells from JAK2$^{V617F}$ mice or control WT mice were injected into lethally irradiated recipient WT mice, so that the recipient mice had WT lungs (Fig. 2a). The BMT mice were exposed to chronic hypoxia for 3 weeks. The *Jak2*V617F VAF in blood leukocytes in the recipient mice transplanted with JAK2$^{V617F}$ BM cells (JAK2$^{V617F}$-BMT) gradually elevated from 4 to 8 weeks after BMT; from 25.5 ± 1.1% to 34.9 ± 6.7% in normoxia-exposed mice, and from 24.5 ± 0.8% to 51.1 ± 5.4% in chronic hypoxia-exposed mice (Fig. 2b), suggesting the nearly complete engraftment of hematopoietic cells with heterozygous *Jak2*V617F. However, blood cell counts in JAK2$^{V617F}$-BMT mice did not exhibit significant increases compared to those in the recipient mice transplanted with WT BM cells (WT-BMT) after normoxia exposure (Fig. 2c), differently from those in individual JAK2$^{V617F}$ mice (Fig. 1b). This finding was consistent with the previously established evidence that recipient mice transplanted with hematopoietic stem/progenitor cells carrying JAK2V617F often fail to show MPN-like phenotypes[26]. Although RVSP and RV/LV + S did not differ between WT-BMT and JAK2$^{V617F}$-BMT mice after normoxia exposure, JAK2$^{V617F}$-BMT mice showed significant increases in both RVSP and RV/LV + S compared to WT-BMT mice in response to exposure to chronic hypoxia for 3 weeks (Fig. 2d), which is consistent with the echocardiography used to assess pulmonary hemodynamics (Supplementary Fig. 10). LV + S values did not differ among the groups (Supplementary Fig. 11). Medial wall thickness, percentage of muscularized vessels and numbers of proliferating smooth muscle cells of pulmonary arteries were significantly increased in JAK2$^{V617F}$-BMT mice compared to WT-BMT mice after hypoxia exposure (Fig. 2e, f, Supplementary Fig. 12a). These findings strongly indicate that a hematopoietic cell clone with JAK2V617F could accelerate PH with pulmonary arterial remodeling in WT lung tissues in response to chronic hypoxia, even without phenotypic MPNs, mimicking PH due to clonal hematopoiesis, such as CHIP. The numbers of Ly6G$^+$ neutrophils in pulmonary arterial regions were significantly increased in JAK2$^{V617F}$-BMT mice compared to WT-BMT mice either after normoxia or chronic hypoxia exposure, and the numbers of Ly6G$^+$ cells in both perivascular and non-perivascular regions in chronic hypoxia-exposed JAK2$^{V617F}$-BMT mice were further increased compared to those in normoxia-exposed JAK2$^{V617F}$-BMT mice (Fig. 2g, h, Supplementary Figs. 13, 14). Ly6G$^+$ cells significantly contributed to CD45$^+$ cells rather than F4/80$^+$ or CD45R$^+$ cells in hypoxia-exposed JAK2$^{V617F}$-BMT lungs (Supplementary Fig. 13). The numbers of CD41$^+$ or TER-119$^+$ cells were not different between

WT and JAK2$^{V617F}$-BMT mice (Supplementary Fig. 12b). Notably, elastase activity, neutrophil-related chemokines and chemokine receptors, and cytokines were significantly elevated in the lungs of JAK2$^{V617F}$-BMT mice in response to chronic hypoxia compared to the other groups (Fig. 2i, j, Supplementary Fig. 12c). Taken together, these data suggest that the neutrophils specifically infiltrating in pulmonary arterial regions induced by clonal hematopoiesis with JAK2V617F are involved in the development of PH.

**Characterization of bone marrow-derived hematopoietic cells with JAK2V617F in the lungs by using GFP-transgene.** To visualize and further characterize BM-derived hematopoietic cells carrying JAK2V617F in pulmonary arterial remodeling, we generated double transgenic mice (JAK2$^{V617F}$/CAG-EGFP mice) by crossing JAK2$^{V617F}$ mice with CAG-EGFP mice[27]. We transplanted BM cells from JAK2$^{V617F}$/CAG-EGFP mice or control WT/CAG-EGFP littermates into lethally irradiated WT mice. After BMT followed by exposure to chronic hypoxia for 3 weeks, immunostaining showed that the GFP$^+$ cells were substantially accumulated in pulmonary arterial regions in BMT recipients transplanted with BM cells from JAK2$^{V617F}$/CAG-EGFP mice (JAK2$^{V617F}$-GFP-BMT), whereas recipients transplanted with BM cells from WT/CAG-EGFP mice (WT-GFP-BMT) showed fewer GFP$^+$ cells in the lungs (Fig. 3a, b). There was no co-localization between GFP and α-smooth muscle actin (αSMA) in the lungs of either WT-GFP-BMT or JAK2$^{V617F}$-GFP-BMT mice. In JAK2$^{V617F}$-GFP-BMT mice, nearly half of the GFP$^+$ cells expressed Ly6G in pulmonary arterial regions, and Ly6G$^+$ cells predominantly contributed to BM-derived cells rather than F4/80$^+$ or CD45R$^+$ cells (Fig. 3c, d, Supplementary Fig. 15). The percentage of these Ly6G-expressing GFP$^+$ cells was significantly higher than that in WT-GFP-BMT mice, while all Ly6G$^+$ cells expressed GFP in both WT-GFP-BMT and JAK2$^{V617F}$-GFP-BMT mice (Fig. 3e). These data indicate that the accumulated Ly6G$^+$ neutrophils carrying JAK2V617F are originated from BM to pulmonary arterial regions.

**Small hematopoietic clones with JAK2V617F lead to development of pulmonary hypertension.** We next performed a competitive transplantation using different ratios of a mixture of WT-GFP or JAK2$^{V617F}$-GFP BM cells and WT without GFP BM cells (Fig. 4a, Supplementary Fig. 16a). In the control non-competitive group, flow cytometry showed that chimerism assessed by GFP$^+$ cells within CD45$^+$ cells in the blood was significantly elevated at 8 weeks compared to that at 4 weeks in 100% WT-GFP-BMT and 100% JAK2$^{V617F}$-GFP-BMT mice (Supplementary Fig. 16b). To determine the minimum threshold of PH aggravation in JAK2$^{V617F}$-GFP-BMT

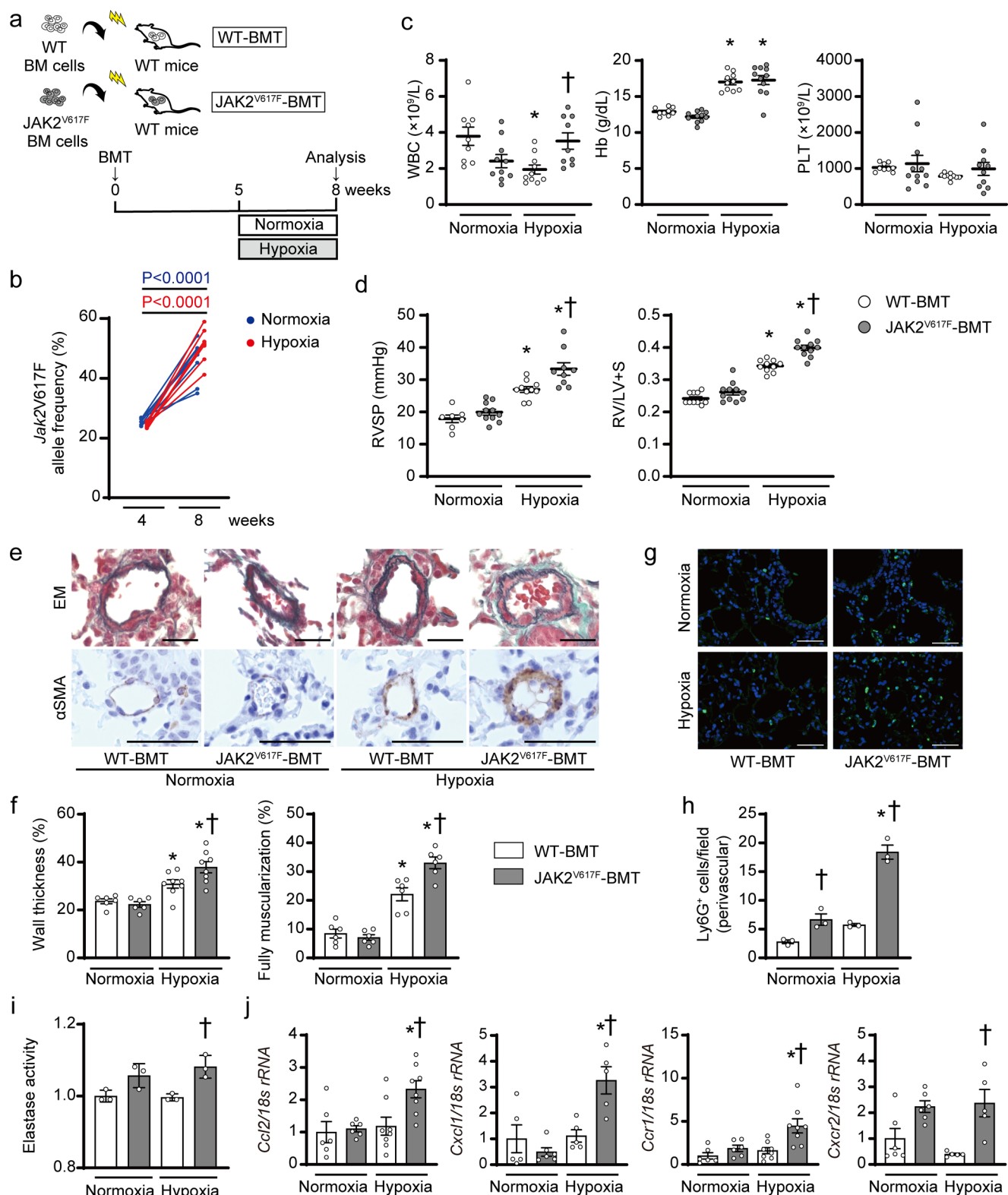

mice, we categorized the recipient mice according to the chimerism level 8-weeks after BMT. Interestingly, when we analyzed the recipients limited to the chimerism of 1–19% as well as 20–49% and 50–100%, the JAK2[V617F]-GFP-BMT mice showed significant increases in RVSP and RV/LV + S compared to the WT-GFP-BMT mice (Fig. 4b, Supplementary Fig. 16c–e). Moreover, the JAK2[V617F]-GFP-BMT mice with lower chimerism of <1% tended to display increases in RVSP and RV/LV + S compared to the WT-GFP-BMT

mice (Supplementary Fig. 16f). These data suggest that even small clones with *Jak2*V617F are associated with PH development.

**JAK2V617F is associated with selective migration of neutrophils into the lungs and maturation for the myeloid lineage from hematopoietic precursors in the lungs.** We isolated cell fraction from the lungs and the blood in WT-GFP-BMT and

**Fig. 2 Clonal hematopoiesis with JAK2V617F exacerbates pulmonary hypertension and infiltration of perivascular neutrophils in bone marrow transplanted recipients with wild-type lungs in response to chronic hypoxia. a** Schematic diagram of the experimental design. Bone marrow (BM) cells from WT or JAK2[V617F] mice were injected into lethally irradiated recipient WT mice with the same C57BL/6 J background. Five weeks after BM transplantation (BMT), the recipient mice transplanted with JAK2[V617F] BM cells (JAK2[V617F]-BMT) or WT BM cells (WT-BMT) were exposed to normoxia or hypoxia for 3 weeks. **b** Jak2V617F allele frequencies (%) in peripheral blood of each JAK2[V617F]-BMT mouse at 4 and 8 weeks after BMT at normoxia (blue circles, $n = 8$) or chronic hypoxia exposure (red circles, $n = 8$). Statistical comparison was performed by the paired Student's $t$ test (two-sided). **c** Peripheral blood cell counts in WT-BMT or JAK2[V617F]-BMT mice after exposure to normoxia or hypoxia ($n = 9, 10, 10, 9$, $^*P = 0.0121$, $^†P = 0.0388$ for WBC, $n = 9, 11, 10, 11$, $^*P < 0.0001$ [left], $<0.0001$ [right] for Hb, $n = 9, 11, 10, 10$ for PLT). **d** RVSP and RV hypertrophy determined by RV/LV + S in WT-BMT or JAK2[V617F]-BMT mice after exposure to normoxia or hypoxia ($n = 7, 11, 10, 9$, $^*P = 0.0002$ [left], $< 0.0001$ [right], $^†P = 0.0054$ for RVSP, $n = 10, 11, 10, 11$, $^*P < 0.0001$ [left], $<0.0001$ [right], $^†P < 0.0001$ for RV/LV + S). **e** Representative images of EM-stained sections and sections immunostained with anti-αSMA antibody from WT-BMT and JAK2[V617F]-BMT mice. Scale bars, 25 μm. **f** Quantitative analysis of medial wall thickness in EM-stained sections (left, $n = 6, 6, 8, 8$, $^*P = 0.0465$ [left], $<0.0001$ [right], $^†P = 0.0346$) and the percentage of muscularized distal pulmonary vessels in αSMA-immunostained sections (right, $n = 6$ in each group, $^*P = 0.0001$ [left], $<0.0001$ [right], $^†P = 0.0016$). **g** Representative immunofluorescence images of lung sections stained with anti-Ly6G (green) antibody and DAPI (blue). Scale bars, 50 μm. **h** Quantitative analysis of Ly6G-positive cells in the perivascular regions ($n = 3$ in each group, $^*P < 0.0001$, $^†P = 0.0387$ [left], $<0.0001$ [right]). **i** Elastase activity in the lungs from WT -BMT and JAK2[V617F]-BMT mice. The average value of normoxia-exposed WT-BMT mice was set to 1 ($n = 3$ in each group, $^†P = 0.0128$). **j** Relative mRNA expression levels of *Ccl2*, *Cxcl1*, *Ccr1*, and *Cxcr2* in the lungs. The 18 s *rRNA* was used for normalization. The average value from the normoxia-exposed WT-BMT mice was set to 1 ($n = 6, 6, 8, 8$, $^*P = 0.0171$, $^†P = 0.0159$ for *Ccl2*, $n = 5, 6, 5, 5$ $^*P = 0.0004$, $^†P = 0.0065$ for *Cxcl1*, 6, 6, 8, 8, $^*P = 0.0171$, $^†P = 0.0040$ for *Ccr1*, $n = 6, 6, 5, 5$, $^†P = 0.0056$ for *Cxcr2*). The data are presented as mean ± SEM. $^*P < 0.05$ versus the corresponding normoxia-group and $^†P < 0.05$ versus the corresponding WT-BMT mice by the one-way ANOVA with Tukey post-hoc analysis. WT-BMT, recipient WT mice transplanted with BM cells of WT mice; JAK2[V617F]-BMT, recipient WT mice transplanted with BM cells of JAK2[V617F] mice. WBC white blood cell count, Hb hemoglobin concentration, PLT platelet count. Source data are provided as a Source Data file.

JAK2[V617F]-GFP-BMT mice with 1–19% chimerism at 8 weeks after BMT. The percentages of GFP[+] cells within Ly6G[+] neutrophils in JAK2[V617F]-GFP-BMT mice were significantly higher in the lungs than in the blood, while those in WT-GFP-BMT mice were not different between the lungs and the blood (Fig. 4c, d). These findings suggest that JAK2[V617F] neutrophils have an intrinsic capability of increased migration into the lungs, and this migration is enhanced in response to hypoxia. Accordingly, ex vivo analysis using chemotaxis assay revealed that JAK2[V617F]-Ly6G[+] cells in the blood displayed a higher capability of neutrophil migration than WT-Ly6G[+] cells (Fig. 4e). To investigate the involvement of hematopoietic progenitors in JAK2[V617F] lungs, CD117 (c-kit)[+] cells were sorted from the lungs and subjected to a colony-forming assay. There were substantial increases in the colony-forming ability of JAK2V617F-expressing progenitor cells, especially toward the myeloid lineage (Fig. 4f, Supplementary Fig. 17). These data indicate that the accumulated Ly6G[+] neutrophils carrying JAK2V617F are migrated from BM to pulmonary arterial regions, and potentially proliferated and maturated from the precursors in the lungs.

**Alternation of gene profiling during neutrophil differentiation with JAK2V617F.** To elucidate the underlying mechanisms of how BM-derived neutrophils carrying JAK2V617F were causally related to PH development, we performed gene expression profiling of the neutrophils at several stages of differentiation by RNA sequencing in sorted Ly6G[+] cells from BM, peripheral blood (PB) and lungs of JAK2[V617F] mice in comparison to WT mice. The purity of the lung Ly6G[+] cell enrichment was confirmed by immunofluorescence (Supplementary Fig. 18). To compare these data with the cells at the hematopoietic stem/progenitor cell level, we used the available RNA sequencing results of lineage-Sca1[+]Kit[+] (LSK) cells in BM from our previous study[22]. We found that 451, 849, 1142, and 1022 genes were upregulated, and 580, 841, 1123, and 1006 genes were downregulated in LSK cells and Ly6G[+] cells of the BM, PB and lungs, respectively, in JAK2[V617F] mice compared to WT mice (Fig. 5a, Supplementary Data 1). Differentially expressed genes in JAK2[V617F] mice were more frequently overlapped among the BM, PB and lung Ly6G[+] cells than between the LSK cells and BM Ly6G[+] cells. Next, we subjected these RNA sequencing results to the pathway analysis (Fig. 5b). Hierarchical clustering analysis

showed that the gene profiling was branched from the LSK cells, and diverged into BM myeloid cells and neutrophils in the lungs and PB, suggesting that the neutrophils were spread peripherally. Some of the canonical pathways were commonly up- and down-regulated at each stage. There were also pathways that were enhanced in accordance with differentiation and that were specifically enhanced in the final stage before peripherally. Thus, the gene expression profiles were differently altered from the LSK cells to lung neutrophils in JAK2[V617F] mice.

**Ly6G[+] cells carrying JAK2V617F progressively increased *Acvrl1* gene expression during the process of differentiation into peripheral pulmonary arterial regions of the lungs.** A gene set enrichment analysis revealed that the canonical IL6-JAK-STAT3 pathway was upregulated at each stage of neutrophil differentiation in JAK2[V617F] mice compared to WT mice (Fig. 5c), with some alterations of differentially expressed individual genes (Fig. 5d). Interestingly, *Activin A receptor like type 1* (*Acvrl1*), which encodes ALK1 and is known as a type I transmembrane serine/threonine kinase receptor, is associated with the pathogenesis of PH[7,28] and has been found to be the most upregulated gene in the canonical IL6-JAK-STAT3-pathway in Ly6G[+] neutrophils of the lungs and PB of JAK2[V617F] mice (Fig. 5d). *Acvrl1* was slightly upregulated in the BM Ly6G[+] myeloid cells and LSK cells of JAK2[V617F] mice (Fig. 5d). Furthermore, the genes associated with neutrophil functions such as protein secretion, degranulation, and granulation were exclusively enriched in the periphery, especially in the lung Ly6G[+] neutrophils with JAK2V617F (Fig. 5e–g).

**Acvrl1 mRNA expressions and phosphorylation of Smad1/5/8 and STAT3 in the lungs of JAK2[V617F] mice in response to chronic hypoxia.** *Acvrl1* mRNA expression levels in the lung homogenates of JAK2[V617F] mice were higher than those of WT mice after exposure to normoxia (Fig. 6a). In response to chronic hypoxia, *Acvrl1* levels were increased in both WT and JAK2[V617F] lungs, but the levels in JAK2[V617F] lungs were greater than those in WT lungs. *Acvrl1* mRNA levels in sorted Ly6G[+] neutrophils were significantly elevated in JAK2[V617F] lungs compared to WT lungs after both normoxia and hypoxia, but not in CD31[+] endothelial cells, suggesting that the changes in *Acvrl1* in the

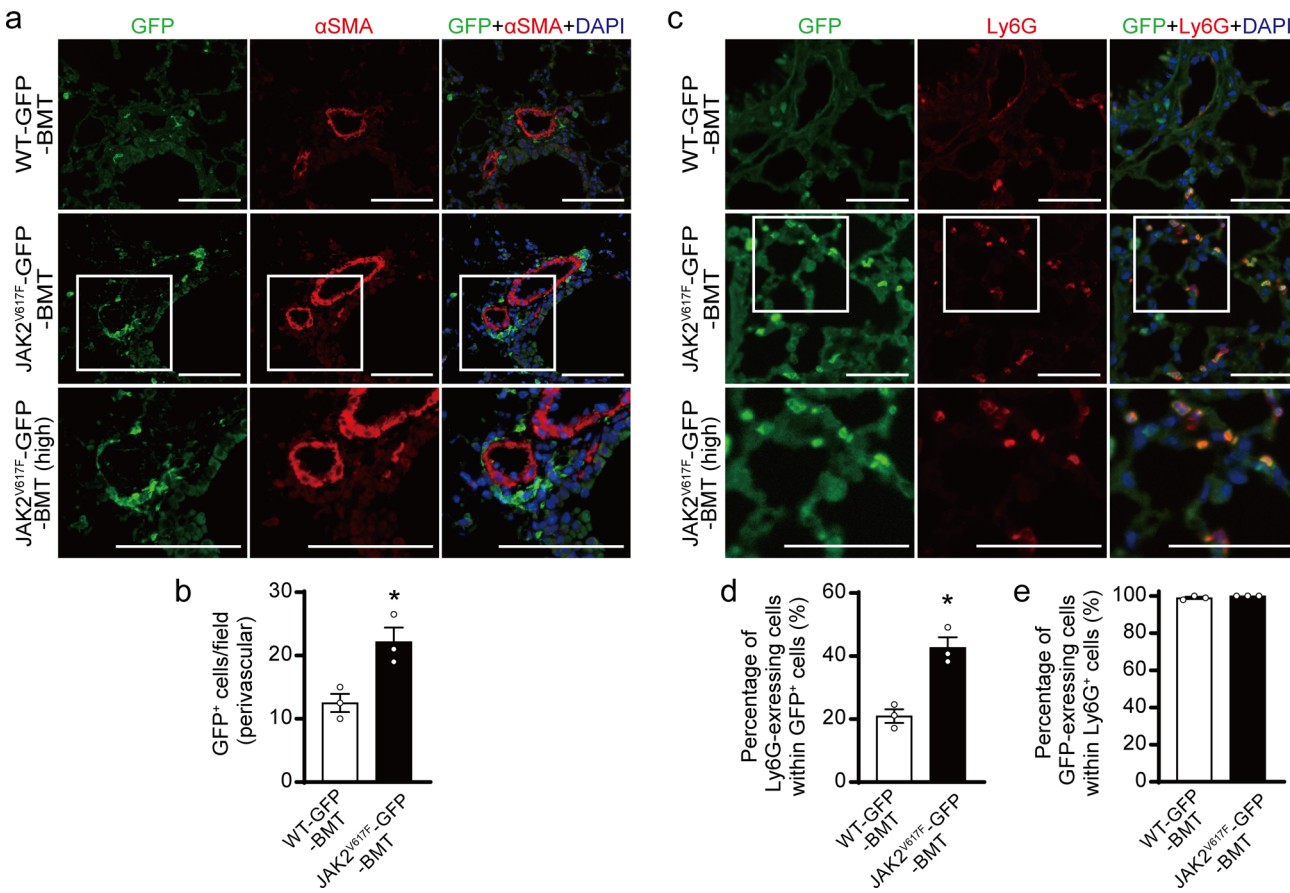

**Fig. 3 Characterization of bone marrow-derived JAK2V617F hematopoietic cells in the lungs by the use of GFP-transgene. a** Lethally irradiated WT mice were transplanted with bone marrow (BM) cells from control WT/CAG-EGFP (WT-GFP) or JAK2$^{V617F}$/CAG-EGFP (JAK2$^{V617F}$-GFP) double transgenic mice. Five weeks after BM transplantation (BMT), the recipient mice were subjected to chronic hypoxia for 3 weeks, and then the lungs were fixed and stained with the indicated antibodies. Representative immunofluorescence images of lung sections stained with anti-GFP (green) and anti-αSMA (red) antibodies and DAPI (blue) in WT-GFP-BMT or JAK2$^{V617F}$-GFP-BMT mice. The boxed areas from JAK2$^{V617F}$-GFP-BMT mice at higher magnifications (high) are shown in the bottom panels. Scale bars, 100 μm. **b** Quantitative analysis of Ly6G$^+$ cells in the perivascular regions ($n = 3$ in each group, $^{*}P = 0.0223$). **c** Representative immunofluorescence images of lung sections stained with anti-GFP (green) and anti-Ly6G (red) antibodies, as well as DAPI (blue) in WT-GFP-BMT or JAK2$^{V617F}$-GFP-BMT mice. The boxed areas from JAK2$^{V617F}$-GFP-BMT mice at higher magnifications are shown in the bottom panels (high). Scale bars, 100 μm. Quantitative analysis of Ly6G-exressing cells in GFP$^+$ cells (**d**, $n = 3$ in each group, $^{*}P = 0.0052$) and GFP-expressing cells in Ly6G$^+$ cells (**e**, $n = 3$ in each group). More than 100 GFP$^+$ cells and Ly6G$^+$ cells were counted in each section and expressed as the percentage of the cells. All data are presented as mean ± SEM. WT-GFP-BMT, recipient WT mice transplanted with WT-GFP BM cells; JAK2$^{V617F}$-GFP-BMT, recipient WT mice transplanted with JAK2$^{V617F}$-GFP BM cells. $^{*}P < 0.05$ versus WT-GFP recipients by the unpaired *t* test (two-sided). Source data are provided as a Source Data file.

lungs resulted from different expression levels of *Acvrl1* in Ly6G$^+$ neutrophils, although *Acvrl1* expression levels were higher in CD31$^+$ cells than in Ly6G$^+$ cells (Fig. 6a, Supplementary Fig. 19a). Similarly, phosphorylation levels of Smad1/5/8, which is down-stream of ALK1, were significantly elevated in JAK2$^{V617F}$ lungs compared to WT lungs after chronic hypoxia (Fig. 6b). There was a significant difference between the tenfold increase in *Acvrl1* mRNA levels versus the twofold increase in phosphory-lated Smad1/5/8 levels in chronic hypoxia-exposed JAK2$^{V617F}$ lungs, indicating that the relationship of *Acvrl1* mRNA expression and the phosphorylation of Smad1/5/8 was not completely linear, and Smad1/5/8 phosphorylation may be regulated by multiple pathways. *Acvr1* mRNA encoding the ALK2 in the lung homo-genates was significantly increased after chronic hypoxia in both WT and JAK2$^{V617F}$ mice, but there were no differences between the groups after normoxia or hypoxia. However, *Acvr1* mRNA in Ly6G$^+$ cells was decreased in both WT and JAK2$^{V617F}$ mice after hypoxia, and the changes in *Acvr1* levels were observed in the opposite direction to those seen in *Acvrl1* (Supplementary Fig. 19b). There were no differences in *Bmpr2* mRNA between

WT and JAK2$^{V617F}$ mice in the lung homogenates, Ly6G$^+$, or CD31$^+$ cells (Supplementary Fig. 19c). STAT3 phosphorylation levels were significantly increased in JAK2$^{V617F}$ lungs compared to WT lungs after normoxia exposure; however, after exposure to chronic hypoxia, these levels in JAK2$^{V617F}$ lungs were even more upregulated compared to the other groups (Fig. 6c). HIF1α expression levels in the lungs were increased in both WT and JAK2$^{V617F}$ mice after chronic hypoxia, but there was no differ-ence between the groups (Supplementary Fig. 20). Immunopre-cipitation analysis showed that STAT3 protein weakly interacted with HIF1α in JAK2$^{V617F}$ lungs at normoxia, and chronic hypoxia increased the bindings (Supplementary Fig. 21). The conditioned medium from hypoxia-exposed JAK2$^{V617F}$ neu-trophils pretreated with a HIF1α inhibitor partly attenuated the increases in the proliferation of pulmonary arterial smooth muscle cells (Supplementary Fig. 22). Thus, JAK-STAT3 signaling in the lungs was constitutively activated in JAK2$^{V617F}$ mice at baseline, whereas both the JAK-STAT3 and ALK1-Smad1/5/8 pathways were further upregulated in JAK2$^{V617F}$ lungs in response to chronic hypoxia, which may be associated with

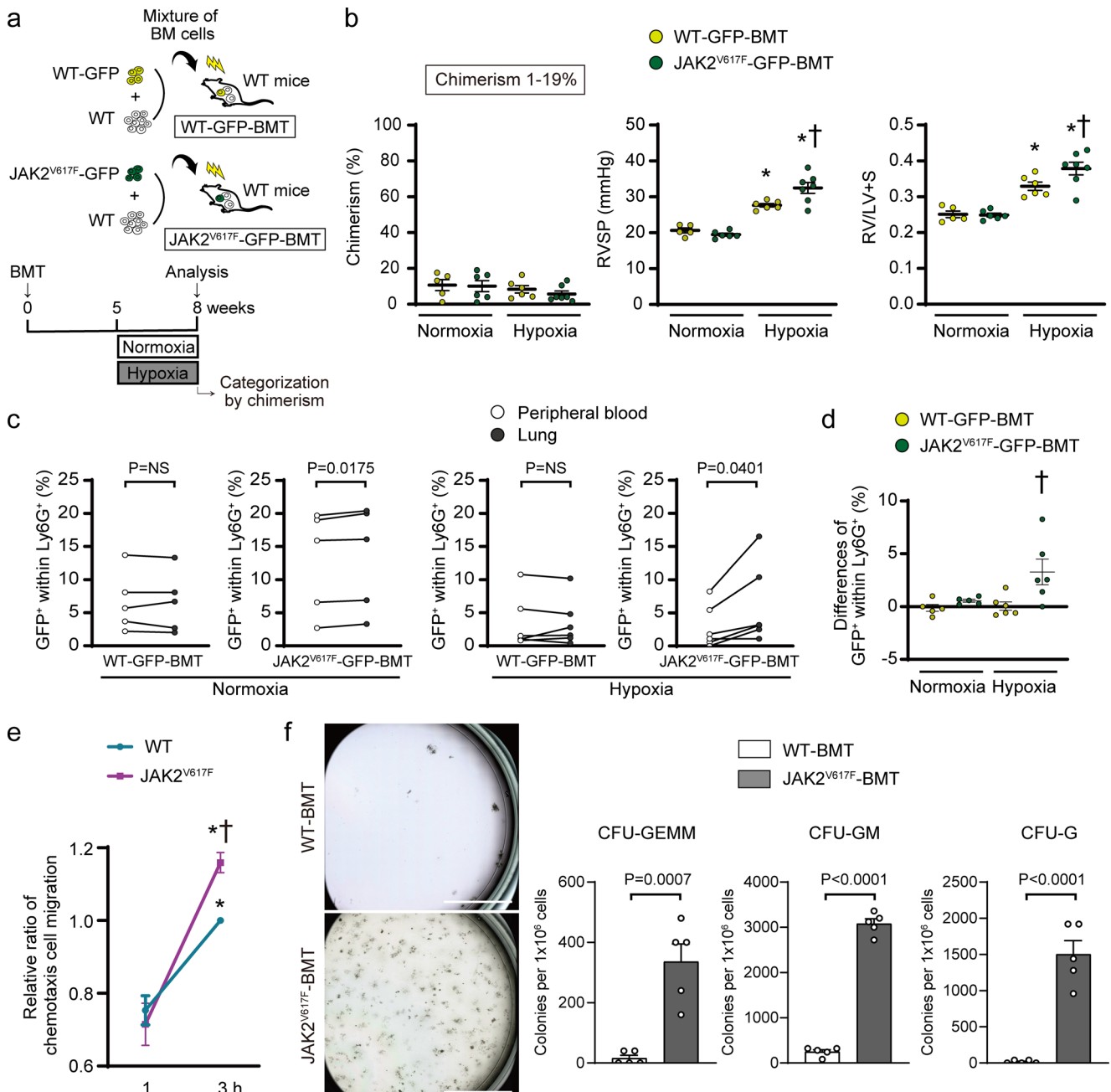

HIF1α. These data suggest that ALK1-Smad1/5/8 in the lungs is associated with PH development due to clonal hematopoiesis with JAK2V617F.

**JAK2V617F transcriptionally upregulates *ACVRL1* by STAT3-binding**. To investigate the regulatory mechanisms of *ACVRL1* by *JAK2*V617F, heterozygous *JAK2*V617F knock-in (*JAK2*V617F/+) HCT116 cell lines were analyzed. Smad1/5/8 was phosphorylated by stimulation of BMP9, a high affinity in ALK1 ligand, in HCT116 cells (Supplementary Fig. 23). Phosphorylation levels of STAT3 in *JAK2*V617F/+ cells were significantly elevated compared to those in *JAK2*+/+ cells (Fig. 7a). *JAK2*V617F/+ cells exhibited significant increases in the expression levels of *ACVRL1* mRNA as well as ALK1 protein and phosphorylation levels of Smad1/5/8 compared to *JAK2*+/+ cells (Fig. 7b, c, Supplementary Fig. 24), but not in *ACVR1* (ALK2) expressions (Supplementary Fig. 25). To assess the effects of *JAK2*V617F on the transcriptional activity of

*ACVRL1*, an in silico analysis was performed, which identified putative STAT3 binding sites in the *ACVRL1* promoter region in both humans and mice (Fig. 7d). The chromatin immunoprecipitation (ChIP) coupled with quantitative PCR showed that the bindings of STAT3 and the putative *ACVRL1* promoter regions were significantly increased in *JAK2*V617F/+ HCT116 cells compared to *JAK2*+/+ HCT116 cells (Fig. 7e). Next, we performed the luciferase reporter assay using the luciferase construct containing the human *ACVRL1* putative promoter sequence from −1035 bp to +210 bp of the transcriptional start site[29]. The promoter activity of *ACVRL1* in *JAK2*V617F/+ cells was significantly increased compared to those of *JAK2*+/+ cells (Fig. 7f). Ruxolitinib, a specific JAK1/2 inhibitor, decreased the *ACVRL1* promoter activity in a dose-dependent manner in *JAK2*V617F/+ HCT116 cells (Fig. 7g). In addition, the administration of stattic, an inhibitor of STAT3, attenuated the *ACVRL1* promoter activity (Fig. 7h). Taken together, *JAK2*V617F increased *ACVRL1*

**Fig. 4 Small hematopoietic clones with JAK2V617F lead to development of pulmonary hypertension, associated with selective migration of neutrophils into the lungs and maturation from the lung hematopoietic precursors for the myeloid lineage. a** Schematic depiction of the competitive transplantation. The different ratios of WT-GFP or JAK2[V617F]-GFP and WT without GFP competitor were transplanted into the lethally irradiated recipient WT mice. **b** The recipients with donor chimerism of 1–19% at 8 weeks after bone marrow transplantation (BMT), determined by the percentages of GFP[+] cells within CD45[+] cells by flow cytometry, were enrolled for statistical comparison ($n = 5, 6, 6, 7$). The other categories of the donor chimerism are presented in Supplementary Fig. 16. RVSP and RV/LV + S are shown ($n = 5, 6, 6, 7$, [*]$P = 0.0008$ [left], <0.0001 [right], [†]$P = 0.0113$ for RVSP, $n = 5, 6, 6, 7$ [*]$P = 0.0029$ [left], <0.0001 [right], [†]$P = 0.0049$ for RV/LV + S). [*]$P < 0.05$ versus the corresponding normoxia-group and [†]$P < 0.05$ versus the corresponding WT-GFP-BMT mice by the one-way ANOVA with Tukey post-hoc analysis. **c, d** JAK2V617F neutrophils showed an intrinsic increased migration capability into the lungs. The percentages of GFP[+] cells within CD45[+] cells in the peripheral blood and the lungs were analyzed at 8 weeks in the BMT mice with 1–19% chimerism by flow cytometry (**c**, $n = 5, 5, 6, 6$). The comparison was performed by the paired Student's $t$ test (two-sided). NS, not significant. The differences of GFP[+] cells within CD45[+] cells between the lungs and the peripheral blood are shown (**d**, $n = 5, 5, 6, 6$). [†]$P = 0.0173$ versus the corresponding WT-GFP-BMT mice by the one-way ANOVA with Tukey post-hoc analysis. **e** Chemotaxis migration assay. The sorted Ly6G[+] neutrophils from the blood in WT or JAK2[V617F] mice were placed on the top of Transwell in triplicate and were allowed to migrate for 1 or 3 h. Data are expressed as a relative ratio to WT-3 h from six independent experiments and presented as mean ± SEM. [*]$P < 0.01$ versus corresponding 1 h ([*]$P = 0.0009$ for WT, < 0.0001 for JAK2[V617F]) and [†]$P = 0.0342$ versus WT-3 h by the two-way ANOVA with Tukey post-hoc analysis. **f** Colony-forming assay of the hematopoietic progenitors in the lungs. CD117 (c-kit)[+] cells sorted from the lungs of WT-BMT and JAK2[V617F]-BMT mice were grown in the methylcellulose-based medium for 7 days. Representative images of the 35 mm plates are shown in the left panels. Scale bars, 10 mm. Right, quantification of numbers of the colonies derived from colony-forming unit (CFU)-granulocyte, -erythroid, -macrophage, -megakaryocyte (CFU-GEMM), CFU-granulocyte, -monocyte (CFU-GM), CFU-granulocyte (CFU-G). The comparison was performed by the two-sided unpaired Student's $t$ test ($n = 5$ in each group). All data are presented as mean ± SEM. Source data are provided as a Source Data file.

transcriptional activity via STAT3-binding, resulting in phosphorylation of Smad1/5/8 in HCT116 cells.

**Inhibition of ALK1/2 prevents chronic hypoxia-induced pulmonary hypertension in JAK2[V617F] mice.** We investigated whether the inhibition of ALK1 could ameliorate chronic hypoxia-induced PH in JAK2[V617F] mice (Fig. 8a). K02288, a chemical inhibitor of ALK1/2[30,31] clearly decreased the phosphorylation levels of Smad1/5/8 in chronic hypoxia-exposed JAK2[V617F] lungs as well as JAK2[V617F/+] HCT116 cells (Supplementary Fig. 26). Administration of K02288 did not affect blood cell counts in JAK2[V617F] mice (Fig. 8b). Remarkably, K02288 treatment significantly decreased RVSP and RV/LV + S in JAK2[V617F] mice compared to DMSO-treated JAK2[V617F] mice after exposure to chronic hypoxia (Fig. 8c, Supplementary Fig. 27). In contrast, K02288 administration did not significantly change the levels of RVSP or RV hypertrophy in chronic hypoxia-exposed WT mice. There were significant decreases in medial wall thickness and muscularization, as well as in the numbers of proliferating smooth muscle cells in pulmonary arteries of K02288-treated JAK2[V617F] mice compared to DMSO-treated JAK2[V617F] mice (Fig. 8d, e, Supplementary Fig. 28). The numbers of Ly6G[+] neutrophils in perivascular regions were decreased in K02288-treated JAK2[V617F] lungs compared to DMSO-treated JAK2[V617F] lungs (Fig. 8f, g). In addition, K02288 treatment significantly decreased elastase activity in JAK2[V617F] lungs (Fig. 8h). Of note, we found that the treatment of LDN-212854, another ALK1/2 inhibitor[31], significantly decreased RVSP and RV/LV + S in chronic hypoxia-exposed JAK2[V617F] mice, similar to K02288 (Supplementary Figs. 29, 30). K02288 or LDN-212854 did not affect the levels of RVSP and RV/LV + S in WT or JAK2[V617F] mice after normoxia (Supplementary Fig. 31). A higher dose of K02288 did not attenuate the PH levels of hypoxia-exposed WT mice (Supplementary Fig. 32). Collectively, these results suggest that the ALK1/2 pathway is involved in chronic hypoxia-induced PH in JAK2[V617F] mice.

**Prevalence of JAK2V617F-clonal hematopoiesis in PH patients.** To clarify the clinical relevance of clonal hematopoiesis with JAK2V617F in PH, we prospectively recruited PH patients, and examined the prevalence of clonal hematopoiesis with JAK2V617F in 70 PH patients by allele specific quantitative PCR

analysis[32]. Strikingly, we found that 7.1% of the PH patients ($n = 5$) showed JAK2V617F somatic mutation in peripheral leukocytes, which was significantly higher than that of the age- and sex-matched control subjects (Fig. 9a, Supplementary Table 1). Among these five PH patients with JAK2V617F, three patients, who were categorized into WHO Group IV (chronic thromboembolic pulmonary hypertension), were regarded as CHIP with a JAK2V617F VAF of ≥ 2% (Fig. 9b, Supplementary Table 2). The JAK2V617F VAF was < 2% in the remaining two patients, who were classified into WHO Group I (pulmonary arterial hypertension). These two patients were in their 50 s and 30 s; younger than the average age of patients with age-related clonal hematopoiesis. Of note, none of the JAK2V617F-positive PH patients met the criteria of hematological disorders including MPNs[33]. There were no significant differences in clinical characteristics, laboratory data including blood cell counts, echocardiographic parameters, or hemodynamics between the PH patients with and without the JAK2V617F mutation (Fig. 9c, d, Supplementary Table 3). These data indicate that clonal hematopoiesis with JAK2V617F is related to the onset and development of PH in the carriers of this mutant, regardless of blood cell counts or PH severity.

**Discussion**

The present study demonstrates that clonal hematopoiesis with JAK2V617F accelerated PH in both the absence and presence of phenotypic MPNs in mice. Neutrophils-derived vascular remodeling was involved in JAK2V617F-mediated PH development. JAK2V617F progressively upregulated *Acvrl1* expression from BM stem/progenitor cells into neutrophils in pulmonary arterial regions in the lungs. JAK2V617F further increased ALK1-Smad1/5/8 signaling accompanied with increases in neutrophil-derived elastase activity and multiple chemokines, resulting in pulmonary arterial remodeling after chronic hypoxia. Correspondingly, JAK2V617F-positive clonal hematopoiesis was more common in the PH patients than in the healthy subjects, despite no signs of hematological disorders.

In the current study, we employed two experimental mouse models mimicking hematological clinical scenarios. Namely, in one model, JAK2[V617F] mice which displayed an MPN-like phenotype were used, and in the other, recipient mice transplanted with JAK2[V617F] BM cells were used to model clonal hematopoiesis without hematologic phenotypes. Both JAK2[V617F] mice

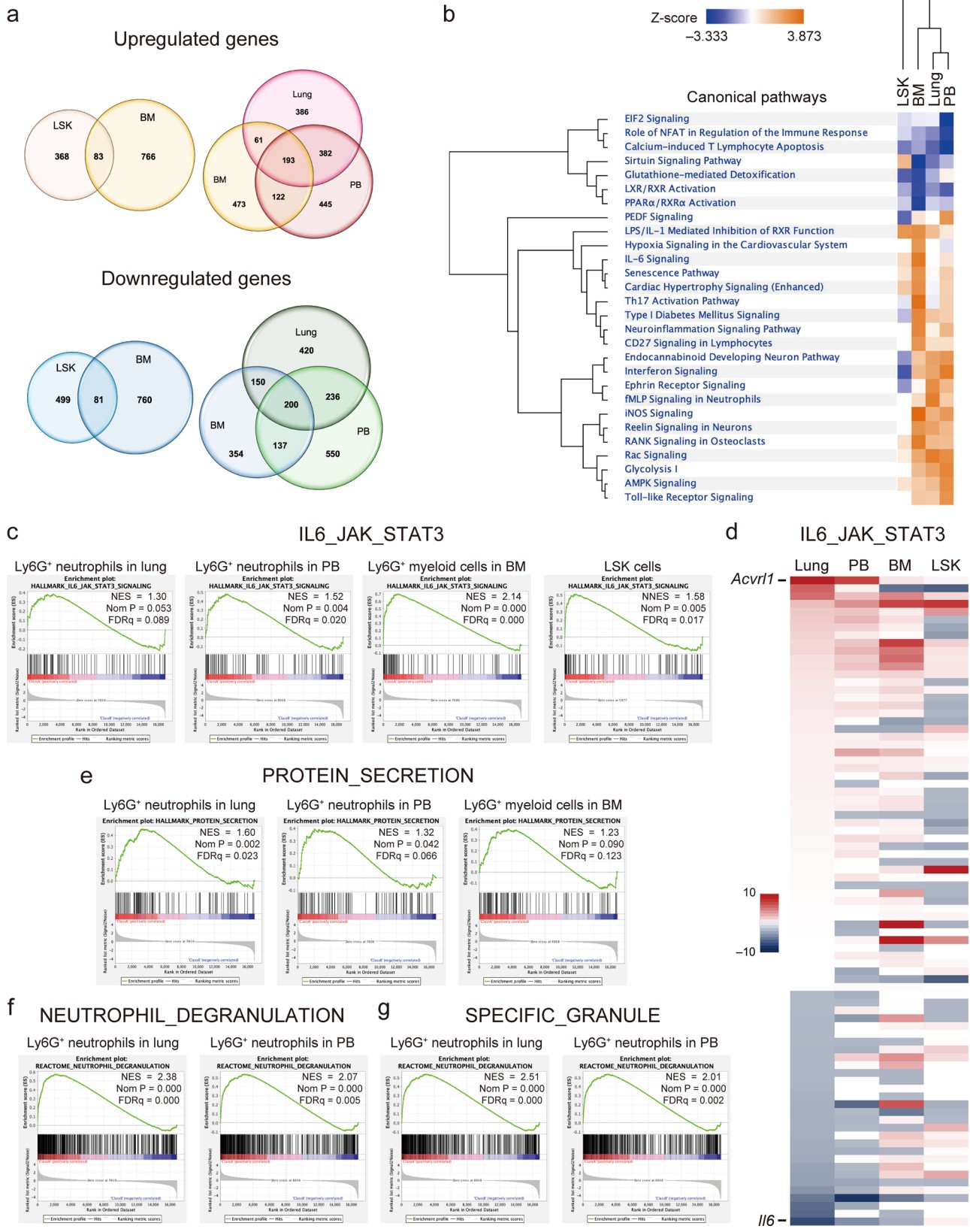

and JAK2$^{V617F}$-BMT mice similarly showed that the number of neutrophils was prominently increased specifically in pulmonary arterial regions, accompanied by vascular remodeling after chronic hypoxia, suggesting that JAK2 activation in neutrophils play a central role in PH. It is likely that the JAK2$^{V617F}$ neutrophils largely migrated into pulmonary arterial regions from

BM. JAK2V617F may increase the adhesion and rolling of neutrophils partly due to increases in formyl peptide receptor (FPR)[17], as our RNA sequencing data demonstrated that both *Fpr1* and *Fpr2* were higher in the Ly6G$^{+}$ lung neutrophils of JAK2$^{V617F}$ mice (3.1- and 3.8-fold, respectively) than in those of WT mice. Moreover, it is possible that the JAK2$^{V617F}$

**Fig. 5 Gene expression profiles of neutrophils with JAK2V617F at several differential stages. a** Venn diagrams show the numbers of upregulated and downregulated genes (>1.5-fold) in Ly6G⁺ neutrophils in lungs and peripheral blood (PB), and Ly6G⁺ myeloid cells in BM, and lineage⁻Sca1⁺Kit⁺ (LSK) cells isolated from JAK2$^{V617F}$ mice ($n = 3$) compared to those from WT mice ($n = 5$) by RNA sequencing. **b** Strongly affected pathways (|z | >2.58) at least one cell type according to the gene expression of Ly6G⁺ neutrophils and LSK cells from JAK2$^{V617F}$ mice relative to those from WT mice. Hierarchical clustering of pathways and cell types are also shown. **c–g** A gene set enrichment analysis (GSEA) of RNA sequencing. Among Hallmark analyses, the IL6-JAK-STAT3 pathway was consistently enriched in JAK2$^{V617F}$ myeloid cells at each differential stage (**c**), but the expression profiles of the individual genes were different between the stem/progenitor and periphery levels (**d**). The expression level of *Acvrl1* was the highest in the lung and PB neutrophils, while slightly upregulated in the BM myeloid cells and LSK cells in this pathway. Gene sets of PROTEIN-SECRETION (**e**), NEUTROPHIL-DEGRANULATION (**f**), and SPECIFIC-GRANULE (**g**) were enriched in mature Ly6G⁺ neutrophils.

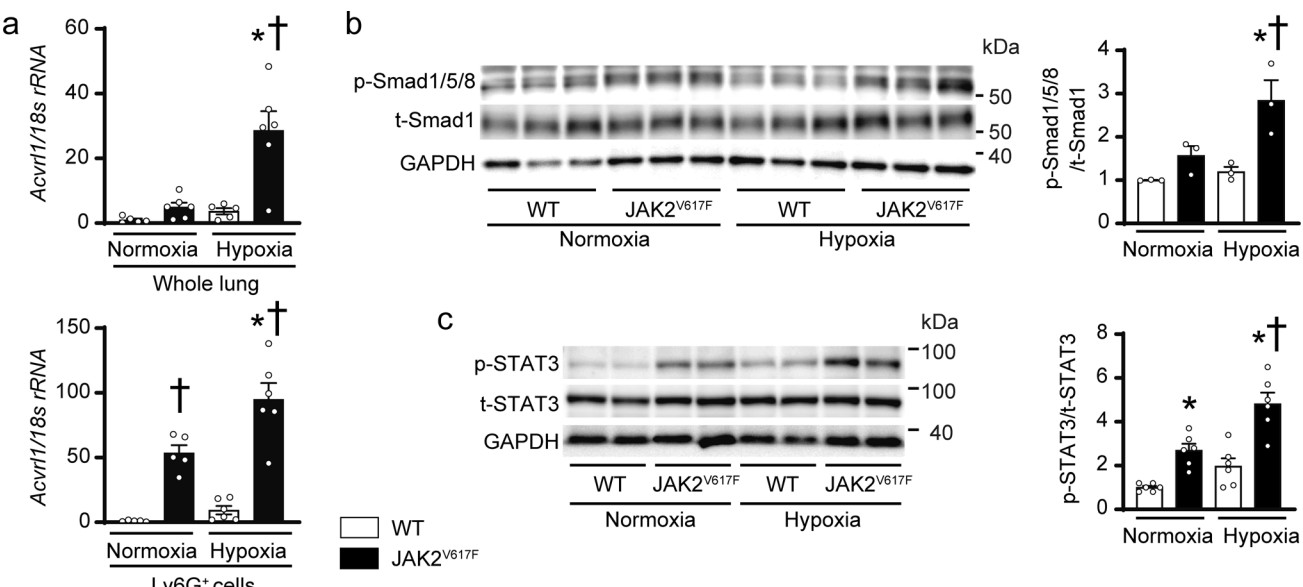

**Fig. 6 *Acvrl1* mRNA expressions and phosphorylation of Smad1/5/8 and STAT3 in the lungs of JAK2$^{V617F}$ mice in response to chronic hypoxia. a** mRNA expression of *Acvrl1* in whole lung extracts (top) or the sorted Ly6G⁺ cells from the lungs (bottom) of WT mice and JAK2$^{V617F}$ mice exposed to normoxia or hypoxia. The data were normalized to *18 s rRNA* levels ($n = 5, 6, 5, 6$, *$P = 0.0004$, †$P = 0.0004$ for whole lung extracts, $n = 5, 5, 6, 6$, *$P = 0.0069$, †$P = 0.0012$ [left], <0.0001 [right] for sorted Ly6G⁺ cells). Western blot analysis on the SMAD (**b**) and STAT (**c**) pathways in the lungs. Lung extracts from WT mice or JAK2$^{V617F}$ mice were immunoblotted with the indicated antibodies. The ratios of phosphorylated Smad1/5/8 (p-Smad1/5/8) to total Smad1 (t-Smad1) and phosphorylated-STAT3 (p-STAT3) to total STAT3 (t-STAT3) are shown in the bar graphs. The average value for normoxia-WT mice was set to 1 (**b**, $n = 3$ in each group, *$P = 0.0382$, †$P = 0.0100$; **c**, $n = 6$ in each group, *$P = 0.0125$ [left], 0.0019 [right], †$P < 0.0001$). GAPDH was used as the loading control. All data are presented as mean ± SEM. *$P < 0.05$ versus the corresponding normoxia-group and †$P < 0.05$ versus the corresponding WT mice by the one-way ANOVA with Tukey post-hoc analysis. Source data are provided as a Source Data file.

hematopoietic precursor cells in the lungs can display the capacity to lodge and complete maturation there. PH patients with or without MPNs have increased circulating CD34⁺ hematopoietic stem/progenitor cells[34]. Engraftment of hematopoietic progenitors from PH patients who did not display any hematological disorder into xenografts showed increases in the growth of myeloid colonies and the expression of myeloid transcription factors, resulting in pulmonary vascular remodeling and right heart hypertrophy[35], suggesting that the intrinsic capability of hematopoietic progenitors is associated with PH. In line with our JAK2$^{V617F}$-BMT model that did not show elevation of white blood cells or platelets, the activation of JAK-STAT in myeloid cells may lead to PH phenotypes even without elevation of leukocyte or platelet counts. The rheological effects of leukocytes and thrombocytes on PH need to be clarified. JAK2$^{V617F}$ mice developed a PH pathology in response to chronic hypoxia, but did not develop PH in normoxia, indicating that JAK2V617F alone is not sufficient to induce PH, and that a trigger such as chronic hypoxia is required for PH phenotypes in JAK2$^{V617F}$ mice. In contrast, patients with MPNs can develop PH in the setting of normoxia. However, not all MPN patients develop PH. As MPNs occurs later in life[36], an additional genetic and/or environmental

hit in addition to JAK2V617F may be needed for the onset and development of PH in predisposed subjects.

Enhanced neutrophil-derived elastase activity is involved in the response of pulmonary arterial smooth muscle cells, resulting in excessive muscularity of the vessels[3,24,37]. Neutrophils produce a wide range of substances that could contribute to exaggerated contractility and proliferation of vascular cells, leading to vascular remodeling in the lungs[38]. While the infiltration of neutrophils was increased by hematopoietic JAK2V617F expression even after normoxia exposure, the increased JAK2$^{V617F}$ neutrophils did not induce pulmonary vascular remodeling or elevate RVSP. RNA sequencing indicated that the differentiated JAK2$^{V617F}$ neutrophils in the lungs and PB, but not in BM myeloid cells or LSK cells, were activated in terms of protein secretion, degranulation and granulation after normoxia exposure. However, elastase activity or neutrophil-derived chemokines were not elevated in JAK2$^{V617F}$ lungs after normoxia exposure. These findings raise the possibility that biological mechanisms such as elastase activity by neutrophils, leading to pulmonary vascular remodeling, might be compensated, unless there is an additional factor, such as chronic hypoxia. Increased physical interactions of HIF1α and STAT3[39] in response to hypoxia might trigger PH phenotypes in

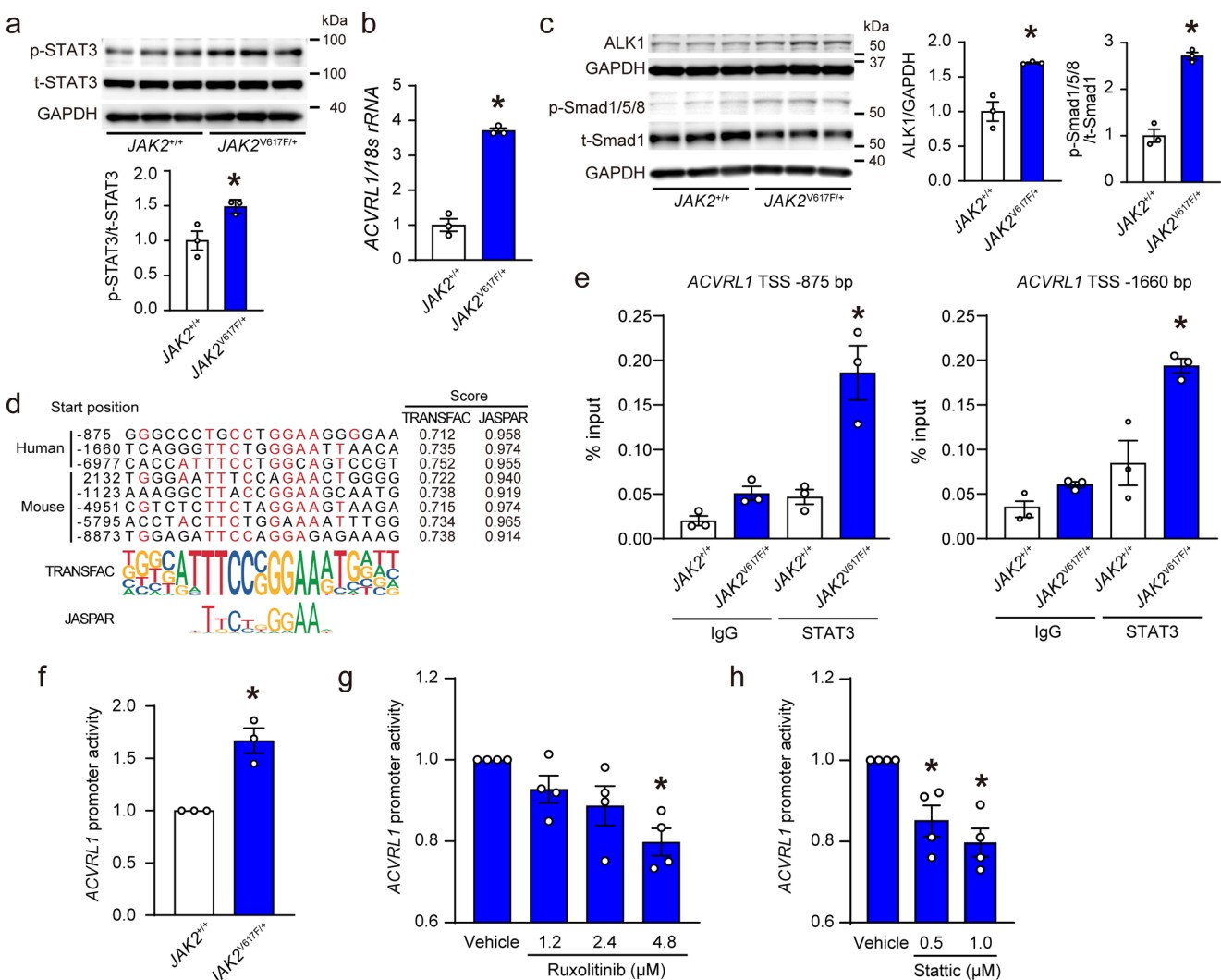

**Fig. 7 JAK2V617F transcriptionally upregulates ACVRL1 by STAT3-binding. a** Western blot analysis of STAT3 in $JAK2^{V617F/+}$ knock-in HCT116 cells. p-STAT3 and t-STAT3 indicate phosphorylated and total STAT3, respectively. p-STAT3 to t-STAT3 ratios are shown in the bar graph ($n = 3$, *$P = 0.0296$). The average value of $JAK2^{+/+}$ HCT116 cells was set to 1. **b** mRNA expression in ACVRL1 in $JAK2^{V617F/+}$ cells. The data were normalized to 18 s rRNA levels. The average value of $JAK2^{+/+}$ cells was set to 1 ($n = 3$, *$P = 0.0001$). **c** Western blot analysis of the ALK1-SMAD pathway. The graphs show the densitometric analysis for ALK1, p-Smad1/5/8 and t-Smad1 ($n = 3$ in each, *$P = 0.0070$, $0.0004$, respectively). p-Smad1/5/8 and t-Smad1 indicate phosphorylated Smad1/5/8 and total Smad1, respectively. GAPDH was used as the loading control. **d** Sequence alignments of putative STAT3 binding sites of Acvrl1 in human (hg19) and mouse (m10). Numbers are given according to the genomic sequence from transcriptional start site (TSS). The sequences of the STAT3 binding motifs are highlighted in red. Sequence logos for the motifs analyzed by TRANSFAC and JASPAR databases are displayed. **e** ChIP-quantitative PCR analysis for STAT3 binding to the putative ACVRL1 promoter. Chromatin was extracted from $JAK2^{+/+}$ and $JAK2^{V617F/+}$ HCT116 cells, and then precipitated with an anti-STAT3 antibody or IgG (negative control). The genomic DNA fragments of ACVRL1 promoter were evaluated for enrichment by quantitative PCR using the specific primers to the Acvrl1 promoter given from TSS. Data are expressed as the respective DNA inputs ($n = 3$ independent experiments, *$P = 0.0015$, $0.0026$, respectively). **f** Dual luciferase reporter assays for the ACVRL1 gene promoter. The pGL3-basic vector containing the putative ACVRL1 promoter region (TSS $-875$ bp) and pNL1.1.TK [Nluc/TK] as a control vector were co-transfected in $JAK2^{V617F/+}$ HCT116 cells. Twenty-four h after transfection, cell lysates were collected, and relative luciferase activity was determined by the ratio of firefly luciferase to Nano luciferase activity ($n = 3$ independent experiments, *$P = 0.0051$). **g, h** Inhibition of JAK1/2 or STAT3 reduced the elevated ACVRL1 promoter activity in $JAK2^{V617F/+}$ cells. Twenty-four h after transfection, the $JAK2^{V617F/+}$ HCT116 cells were incubated with a specific JAK1/2 inhibitor, ruxolitinib or a specific STAT3 inhibitor, stattic, at the indicated concentration for a further 24 h, and then luciferase activity was measured ($n = 4$ independent experiments, **g**, *$P = 0.0059$; **h**, $n = 4$ independent experiments, *$P = 0.0164$ [left], $0.0027$ [right]). All data are presented as mean ± SEM. *$P < 0.05$ versus $JAK2^{+/+}$ cells or vehicle by the unpaired Student's t test (two-sided) or the one-way ANOVA with Tukey post-hoc analysis. Source data are provided as a Source Data file.

$JAK2^{V617F}$ mice, but further mechanisms and investigation by other PH models such as the monocrotaline-pyrrole need to be clarified.

The binding of STAT3 to ACVRL1 promoter regions induced by $JAK2$V617F upregulated ACVRL1 gene expression at the transcriptional level, in addition to the previously reported finding of transcriptional regulation of ACVRL1[29]. It is known that

ACVRL1 is one of the genes affected by germline mutations identified in patients with pulmonary arterial hypertension[28]. Germline mutations of ACVRL1 also cause hereditary hemorrhagic telangiectasia, a dominant autosomal vascular dysplasia, and PH is recognized as a severe complication of this disease[40,41]. It has been reported that ACVRL1 mutations in hereditary hemorrhagic telangiectasia led to a loss of function[42,43]. Most of

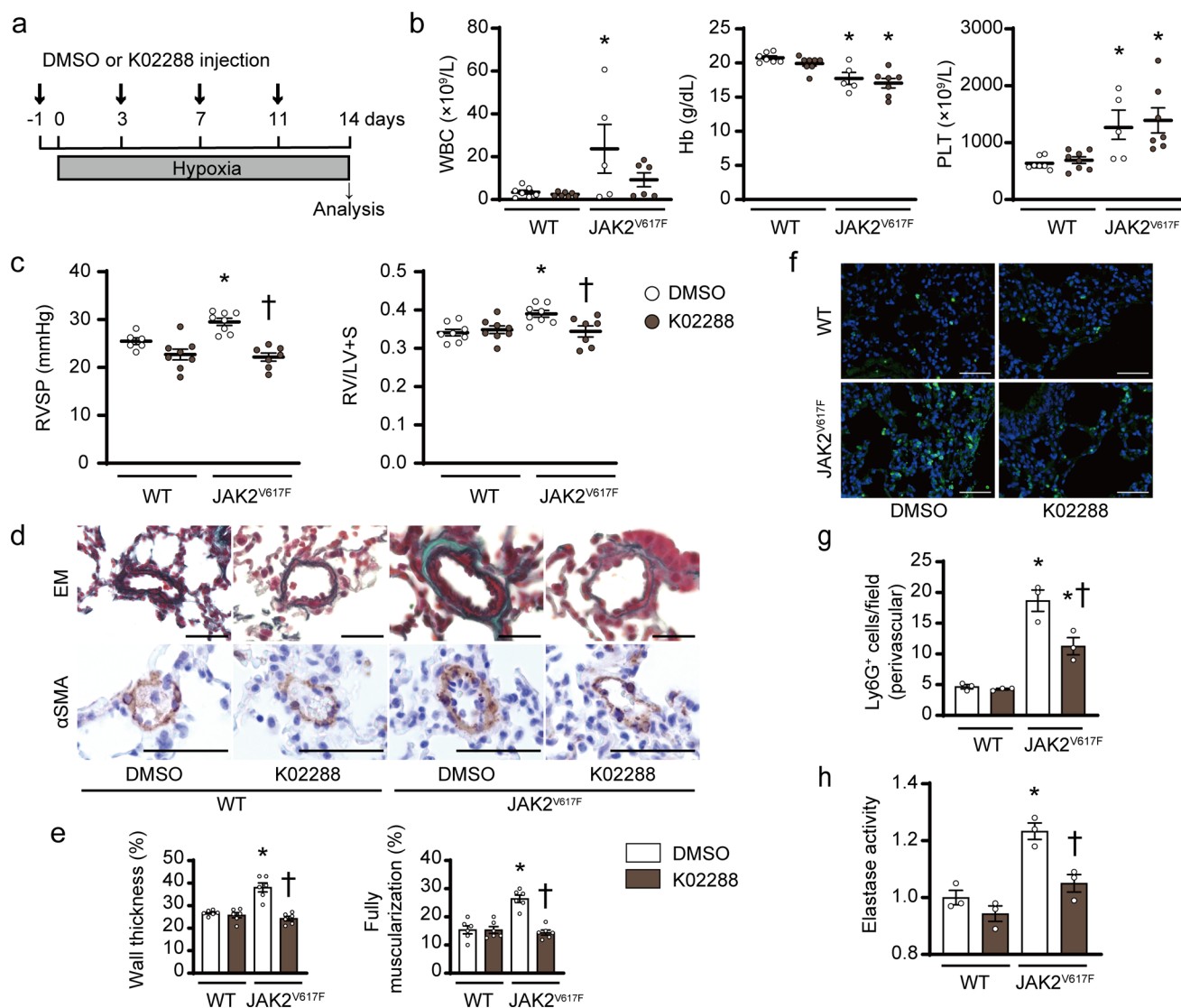

**Fig. 8 Inhibition of ALK1/2 improves chronic hypoxia-induced pulmonary hypertension in JAK2$^{V617F}$ mice. a** Schematic protocol. Vehicle (DMSO) or an ALK1/2 inhibitor, K02288 was administered via an intraperitoneal injection of 12 mg/kg body weight during 2-week chronic hypoxia-exposure, as indicated. **b** Peripheral blood cell counts in DMSO- or K02288-treated WT mice and JAK2$^{V617F}$ mice after exposure to chronic hypoxia for 2 weeks ($n = 7$, 7, 5, 6, *$P = 0.0381$ for WBC, $n = 7$, 8, 5, 7, *$P = 0.0074$ [left], 0.0037 [right] for Hb, $n = 7$, 8, 5, 7, *$P = 0.0401$ [left], 0.0120 [right] for PLT). **c** RVSP and RV hypertrophy determined by RV/LV + S in DMSO- or K02288-treated WT mice and JAK2$^{V617F}$ mice ($n = 6$, 8, 8, 7, *$P = 0.0238$ for RVSP, $n = 8$, 8, 8, 7, *$P = 0.0112$, †$P = 0.0240$ for RV/LV + S). **d** Representative images of EM-stained sections and sections immunostained with anti-αSMA antibody from DMSO- or K02288-treated WT mice and JAK2$^{V617F}$ mice. Scale bar, 25 μm. **e** Quantitative analysis of medial wall thickness in EM-stained sections (left, $n = 6$ in each group, *$P < 0.0001$, †$P < 0.0001$) and the percentage of muscularized distal pulmonary vessels in αSMA-immunostained sections (right, $n = 6$ in each group, *$P < 0.0001$, †$P < 0.0001$). **f** Representative immunofluorescence images of lung sections stained with anti-Ly6G (green) antibody and DAPI (blue). Scale bars, 50 μm. **g** Quantitative analysis of the numbers of Ly6G$^+$ cells in the perivascular regions ($n = 3$ in each group, *$P = 0.0001$ [left], 0.0103 [right], †$P = 0.0074$). **h** Elastase activity in the lung extracts from DMSO- or K02288-treated WT mice and JAK2$^{V617F}$ mice. The average value for DMSO-treated WT mice was set to 1 ($n = 3$ in each group, *$P = 0.0017$, †$P = 0.0075$). All data are presented as mean ± SEM. *$P < 0.05$ versus the corresponding WT mice and †$P < 0.05$ versus DMSO-treated JAK2$^{V617F}$ mice by the one-way ANOVA with Tukey post-hoc analysis. WBC white blood cell count, Hb hemoglobin concentration, PLT platelet count. Source data are provided as a Source Data file.

*ACVRL1* mutations found in pulmonary arterial hypertension are the same mutations described in HHT which result in a loss of function. The loss-of-function mutations in *ACVRL1* are important causes of heritable pulmonary arterial hypertension[44]. Consistently, heterozygous ALK1 knockout mice developed PH in adulthood[45]. In contrast, the inhibition of BMP9 partly protected chronic hypoxia-induced PH in the adult mice and systemic administration of ALK1 inhibitor, a ligand trap targeting ALK1, prevented the monocrotaline and Sugen hypoxia-induced PH in the adult rats[46], suggesting that systemic blockade of the

BMP9/ALK1 pathways is beneficial for PH in the adult rodents. In the present study, we showed that ALK1/2 inhibitor administration prevented the progression of chronic hypoxia-induced PH in JAK2$^{V617F}$ mice, indicating that JAK2V617F-related ALK1 upregulation in myeloid cells had detrimental effects in PH. Although the molecular roles of ALK1 have been investigated particularly in endothelial cells, ALK1 expressions in myeloid cells may have a different impact on PH from the lung endothelial cells. As the functional relevance of ALK1 in PH is not fully understood, a conditional knockout model of hematopoietic cells

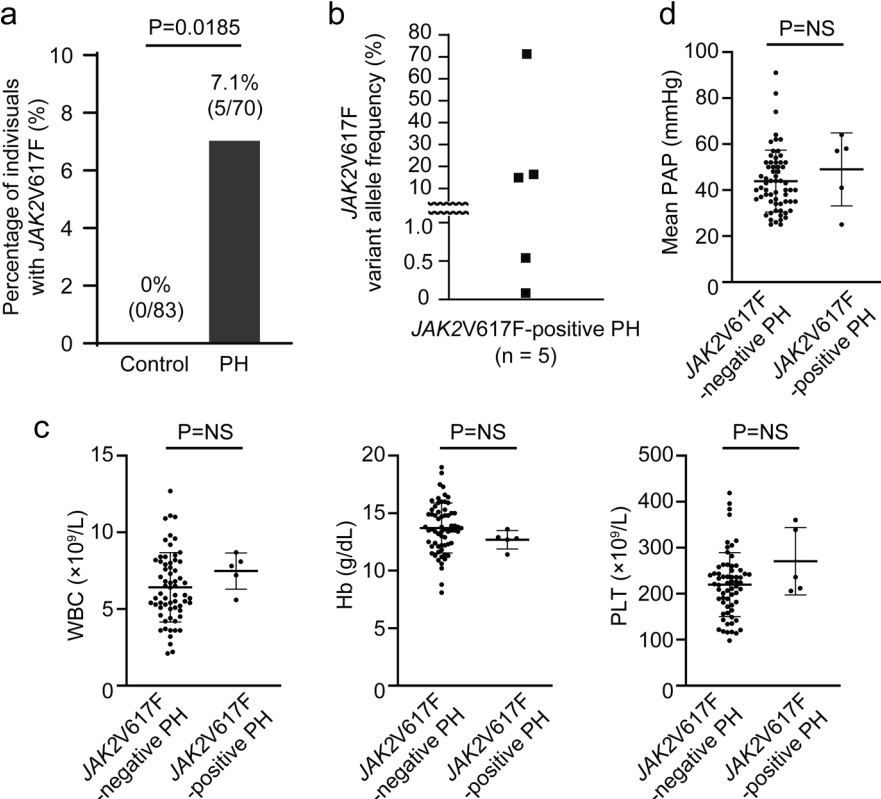

**Fig. 9 Prevalence of *JAK2*V617F-positive clonal hematopoiesis in PH patients. a** *JAK2*V617F-positive clonal hematopoiesis was more common in PH patients. The comparison between PH patients ($n = 70$) and age- and sex-matched control subjects ($n = 83$) was made by Fisher's exact test (two-sided). **b** *JAK2*V617F variant allele frequency. **c, d**, Peripheral blood cell counts, and mean PAP (pulmonary arterial pressure) evaluated by right heart catheterization between *JAK2*V617F-negative and *JAK2*V617F-positive PH patients ($n = 64$, 5 for WBC, Hb, PLT and $n = 63$, 5 for mean PAP). **d**. Data are presented as mean ± SD. Comparisons of values between the two groups were performed by the unpaired Student's *t*-test (two-sided). WBC white blood cell count, Hb hemoglobin concentration; PLT platelet count, NS not significant. Source data are provided as a Source Data file.

is needed to clarify the role of ALK1 on PH in the hematopoietic system.

It has recently been reported that clonal hematopoiesis was especially associated with atherosclerotic cardiovascular diseases[12,15]. Among the somatic mutations related to clonal hematopoiesis, individuals with *JAK2*V617F showed a higher risk of coronary heart disease compared to those without CHIP and those with mutations other than *JAK2*V617F[15]. We showed here the association between clonal hematopoiesis and PH. Importantly, five out of the 70 PH patients (7.1%) were carriers of *JAK2*V617F-positive clonal hematopoiesis, three of whom fulfilled the criteria of CHIP, with a *JAK2*V617F VAF exceeding 2%. The remaining two patients were in their 30 s and 50 s, younger than the average age of patients with age-related clonal hematopoiesis. Our murine study demonstrated that even small clones with *JAK2*V617F led to PH development. Given that *JAK2*V617F VAF as low as 0.1–2% was associated with the elevation of blood cell counts, manifestations of MPNs, thrombotic events, and survival in both *JAK2*V617F-positive general populations without MPNs and patients with MPNs[47–50], the presence of *JAK2*V617F, even with low VAF, may have a clinically biological impact. Further study into the relationship between *JAK2*V617F VAF levels and PH prevalence is required.

Currently, no treatment has yet been established to prevent or directly modify clonal hematopoiesis-associated cardiovascular diseases. The JAK1/2 inhibitor ruxolitinib is now routinely used in patients with MF and PV for improvements of splenomegaly and disease-related symptoms[51,52]. For MPN patients with PH complications, ruxolitinib has been shown efficacy to ameliorate

PH only in a small number of patients[53,54] or has actually exacerbated PH in some cases[55]. There are concerns, such as hematologic toxicities, dysfunction of lymphocytes, and reactivation of viral infections, regarding the use of ruxolitinib for patients with clonal hematopoiesis without any hematologic disorders. The only possible treatment to eliminate clones with somatic mutations, including *JAK2*V617F, is hematopoietic stem cell transplantation, which is however often associated with serious comorbidity and treatment-related mortality. Moreover, it has recently been reported that PH is associated with poor outcome of hematopoietic stem cell transplantation in patients with MPNs[56]. Therefore, transplantation may not be a suitable strategy for PH patients with MPNs or clonal hematopoiesis, unless the patient is in a severe hematologic condition, such as acute leukemia.

Although medical therapies for PH, such as prostanoids and endothelin receptor antagonists, have been greatly improved, PH remains a progressive and fatal disease[1]. Precision medicine may be a novel approach that identifies *JAK2*V617F-positive PH patients regardless of the etiologies of PH. In the present study, we did not find any significant differences in clinical characteristics, including blood cell counts or hemodynamics, between the *JAK2*V617F-positive and -negative PH patients. In turn, these findings suggest that examination of *JAK2*V617F may be a potential strategy, which may help in the diagnosis and treatment of *JAK2*V617F-positive PH patients. Furthermore, three patients with *JAK2*V617F were categorized into Group IV, implying that JAK2V617F promotes venous thrombosis resulting in pulmonary embolisms[57]; however, our murine data showed that pulmonary

arterial structural remodeling was accelerated in the presence of hematopoietic JAK2V617F with no distinct features of venous thrombosis in the lungs. Notably, ALK1/2 inhibitors completely prevented chronic hypoxia-induced PH in JAK2V617F-mediated clonal hematopoiesis, without causing hematologic toxicity. Inhibition of ALK1/2 may be effective especially in the JAK2V617F lung neutrophils. Although we cannot exclude the potential effects of ALK2 on PH, ALK1 is a promising therapeutic target for PH patients with clonal hematopoiesis induced by *JAK2*V617F.

A limitation of this study was that overexpression levels of the transgenic mice expressing murine JAK2V617F might non-specifically affect the varieties of individual phenotypes; therefore, we also used BMT models. In human studies, inherited genetic backgrounds or other CHIP-related mutations could not be determined in the PH patients in a small sample size. Future work is needed to validate our findings in larger cohorts.

In conclusion, we unveiled that a hematopoietic cell clone with JAK2V617F was involved in the development of PH with neutrophil-derived vascular remodeling with ALK1 upregulation. Our study provides an approach for precision medicine that identifies *JAK2*V617F in PH patients, and suggests ALK1 as a possible candidate of therapeutic target.

## Methods

**Animals**. JAK2[V617F] mice of transgenic-*Jak2*V617F with a C57BL/6 J background were obtained as described previously[21,22]. Female JAK2[V617F] mice aged between 8 and 10 weeks (body weight range, 18–24 g) were used in the present study. WT littermates were used as controls. CAG-EGFP reporter mice with a C57BL/6 J background were purchased from Japan SLC. JAK2[V617F] mice were crossed with CAG-EGFP mice to generate JAK2[V617F]/CAG-EGFP double transgenic mice (JAK2[V617F]-GFP)[27], and WT littermates were used as controls (WT-GFP). We used female mice unless otherwise indicated. Mice were housed with food and water ad libitum during 12-h light/12-h dark cycles (light, 7:00–19:00; dark, 19:00–7:00), and ambient temperature (21.5 °C) and humidity (55 ± 10%) were monitored.

**Peripheral blood analysis**. Blood was collected from the tail vein and blood cell counts were determined using Sysmex pocH-100i (Sysmex).

**Exposure to chronic hypoxia**. The mice were exposed to normoxia (21% $O_2$) or hypoxia (10% $O_2$) for 2 or 3 weeks in a ventilated chamber[20]. The hypoxic environment was kept in a mixture of air and nitrogen (Teijin Ltd.). The chamber was kept closed, and was only opened to supply food and water as well as for cleaning twice a week. In a Sugen-hypoxia model, the mice received a single weekly injection of a VEGF inhibitor, SU-5416 (HY-10374, Med Chem Express), at 20 mg/kg followed by 2 weeks of hypoxia (10% $O_2$).

**Echocardiography**. Transthoracic echocardiography was performed using Vevo 2100 High-Resolution In Vivo Imaging System (Visual Sonics Inc.) with a 40-MHz imaging transducer. Mice were lightly anesthetized by titrating isoflurane (0.5–1.5%) to achieve a heart rate of around 400/min. RV fractional area change, RV diastolic dimension, PA acceleration time, PA ejection time, RV anterior wall diameter, tricuspid annular plane systolic excursion, cardiac output, and LV fractional shortening were determined[58].

**Hemodynamics and assessment for right ventricular hypertrophy**. After chronic exposure to normoxia or hypoxia, the mice were anesthetized by intra-peritoneal injection of 2,2,2-tribromo-ethanol (0.25 mg/g per body weight)[20]. A 1.2 F micromanometer catheter (Transonic Scisense Inc.) was inserted from the right jugular vein, and RVSP was continuously measured. The RVSP was blindly analyzed by LabScribe3 software (IWORX) and averaged over 10 sequential beats. To evaluate RV hypertrophy, the RV was dissected from the LV, including the septum, and RV/LV + S was calculated.

**Histological analysis**. Lung samples were fixed in 4% paraformaldehyde solution for paraffin embedding. Frozen lung tissues were embedded in the O.C.T. compound (Tissue-Tek). The paraffin-embedded sections were stained with H&E or Elastica-Masson (EM), or they were used for immunostaining. In the EM-stained sections, the wall area between the internal and external lamina of the pulmonary arteries with a diameter between 50 and 100 µm was measured and expressed as the percentage of medial wall thickness divided by the vessel area using ImageJ software (National Institutes of Health)[58]. In the sections stained with αSMA (M0851, Dako), the pulmonary vessels with a diameter of less than 50 µm were classified into three groups; the vessels with αSMA-positives throughout the entire circumference of the vessel cross-section was defined as "fully" muscularized, the vessels with αSMA-positives with 5–99% around the vessel was defined as "partially" muscularized, and the vessels with αSMA-positives with <5% around the vessel was classified as "non" muscularized[58]. Based on the anatomical characteristics, pulmonary arteries, distributed along the bronchi and displayed an eccentric morphology with thick and elastic walls, are distinguishable from pulmonary veins. The percentage of muscularized pulmonary vessels was determined by dividing the sum of the partially and fully muscular vessels by the total number of vessels[58]. For immunofluorescence staining, the paraffin-embedded tissue sections were incubated with primary antibodies against Ly6G (1:100, ab25377, Abcam), CD45 (1:100, 70257, Cell Signaling Technology; sc-53665, Santa Cruz Biotechnology Inc.), F4/80 (1:100, 70076, Cell Signaling Technology), CD45R (1:100, 103201, Biolegend), Ki67 (1:100, ab15580, Abcam), αSMA (1:100, M0851, Dako; 19245, Cell Signaling Technology), or GFP (1:100, NBP2-22111, Novus Biologicals). This was followed by incubation with the appropriate secondary antibodies, including Alexa Fluor 488 (1:1000, ab150105, Abcam), Alexa Fluor 594 (1:1000, R37119, A-21211; Thermo Fisher Scientific), and Alexa Fluor 647 (1:1000, ab150159, Abcam), then mounted with DAPI-containing mounting media (Fluoro Gel II, Electron Microscopy Sciences). Immunohistochemical staining of the paraffin-embedded or O.C.T.-embedded sections was performed with the following primary antibodies; CD41 (1:100, ab63983, Abcam) or TER-119 (1:100, 116201, BioLegend) followed by anti-rabbit or anti-rat IgG antibody labeled with peroxidase (14341 F, 414311 F, Nichirei Bioscience) with DAB peroxidase substrate system (Dojin Co., Ltd.) and counterstaining with hematoxylin. For quantification of perivascular cellular infiltration, more than 100 cells were counted around the distal pulmonary arteries, with a diameter of 50–100 µm in each mouse[59]. All images were acquired by a microscope (BZ-X700, Keyence Co.) using Keyence BZ II Viewer software (Keyence Co.).

**Western blot analysis**. Snap frozen mouse lung samples or cultured cells were initially homogenized in lysis buffer (Cell Lysis Buffer, Cell Signaling Technology) containing protease inhibitor cocktail (BD Biosciences)[60]. Protein concentration was determined using a Pierce BCA Protein Assay Kit (Thermo Fisher Scientific). Aliquots of proteins were subjected to SDS-polyacrylamide gel electrophoresis, transferred onto polyvinylidene difluoride membranes (Merck Millipore), and probed with the following primary antibodies; HIF1α (1:1000, 36169, Cell Signaling Technology), Phospho-STAT3 (1:1000, 9145, Cell Signaling Technology), STAT3 (1:1000, 4904, Cell Signaling Technology), Phospho-Smad1/Smad5/Smad8 (1:1000, AB3848-I, Merck Millipore), Smad1 (1:1000, 9743, Cell Signaling Technology), ALK1 (1:1000, 14745-1-AP, Proteintech), ALK2 (1:1000, MAB637, R&D Systems) and GAPDH (1:1000, 60004-1-Ig, Proteintech) followed by appropriate goat anti-rabbit or mouse horseradish peroxidase-conjugated secondary antibodies (1:10000, sc-2357, sc-516102, Santa Cruz Biotechnology Inc.). Immunoreactive bands were visualized by an Amersham ECL system (Amersham Pharmacia Biotech UK Ltd.), and signals were detected with an ImageQuant LAS-4000 digital imaging system (GE Healthcare). Or fluorescent immunoreactive bands were detected by an Odyssey CLX imaging system (LI-COR Biosciences) when the appropriate IRDye 680 or IRDye 800 secondary antibodies (1:20000, 925-68070, 925-68071, 925-32210, 925-32210, LI-COR Biosciences) were used. Optical densities of individual bands were analyzed using ImageJ software or Image Studio software (LI-COR Biosciences).

**Reverse transcription-quantitative polymerase chain reaction (RT-qPCR)**. Total RNA was extracted from mouse lungs, sorted cells or cultured cells using Trizol reagent according to the manufacture's protocol (Thermo Fisher Scientific). The RNA from the lung samples was further purified using RNeasy Fibrous Tissue Mini Kit (Qiagen Inc.). cDNA was synthesized using ReverTra Ace qPCR RT Master Mix (Toyobo Co., Ltd.). Quantitative PCR was performed to determine the mRNA expression of *Ccl2*, *Cxcl1*, *Ccr1*, *Cxcr2*, *Pdgfrb*, *Tgfb1*, *Acvrl1*, *Acvr1*, and *Bmpr2* using THUNDERBIRD SYBR qPCR Mix (Toyobo Co., Ltd.) in a CFX Connect real-time PCR System (Bio-Rad) with Bio-Rad CFX Manager 3.1 software (Bio-Rad). A standard curve method on serially diluted templates was applied for the lung samples, and a delta CT method was used for the cell samples. All data were normalized to 18 s *rRNA* and expressed as a fold increase of the control group. Primer sequences are described in Supplementary Table 4.

**Elastase assay**. Elastase activity in the lung tissue was evaluated using the EnzChek Elastase Assay Kit (Molecular Probes)[24,61]. Briefly, the frozen lung samples (20 mg) were homogenized and mixed with the extraction buffer containing NaAc and Na azide, and then rotated overnight at 4 °C. After centrifuge, the pellet was reextracted by adding $(NH_4)_2SO_4$ buffer. After overnight precipitation, the centrifuged pellet was resuspended in 50 mM TrisHCl assay buffer (pH 8.0) to reactivate the elastase. Elastase activity was then measured by adding bovine DQ-Elastin as a fluorogenic substrate in duplicate wells.

**Bone marrow transplantation (BMT)**. Recipient female C57BL/6 J mice aged between 8 and 10 weeks (Charles River Japan, Inc.) were lethally irradiated with a

total dose of 9.0 Gy 24 h before BMT[22]. Whole BM cells were harvested from donor femurs and tibiae. The cells were washed with PBS and $5.0 \times 10^6$ of BM cells were injected in the recipient mice via the tail vein. Peripheral blood parameters and chimerism were analyzed at 4 weeks after transplantation and at the termination of the experiments. DNA was isolated using the QuickGene DNA whole blood kit (KURABO) and quantitative PCR was performed using THUNDERBIRD SYBR qPCR Mix with the following primers; forward primer for donor and recipients, 5′-CTTTCTTCGAAGCAGCAAGCATGA-3′, reverse primer for recipients; 5′-CTGGCTTTACTTACTCTCCTCTCCACAGAC-3′ reverse primer for donors; 5′-AACCAGAATGTTCTCCTCTCCACAGAA-3′. Delta Ct ($Ct_{donor}$− $Ct_{total}$) was calculated to estimate *Jak2*V617F VAF in JAK2[V617F]-BMT mice.

**Magnetic-activated cell sorting (MACS)**. Myeloid cells and neutrophils from the BM, PB and lungs were isolated by using MACS MS columns (Miltenyi Biotec GmbH) with Ly6G MicroBeads according to the manufacturer's protocols. To form a cell suspension from the lungs, the tissues were minced and digested in 2 mg/mL collagenase type II (Worthington Biochemical) for 30 min. Then the tissues were passed through an 18-gauge needle and a 70 μm cell strainer. The purity of the neutrophils was >98% as determined by May-Giemsa staining, and the specificity was confirmed with positive immunostaining by anti-Ly6G (ab25377, Abcam) and anti-Myeloperoxidase (ab9535, Abcam) antibodies and with negative immunostaining by an anti-CD31 antibody (102401, BioLegend). The hematopoietic stem progenitor cells from the lungs were isolated using CD117 MicroBeads. The endothelial cells from the lungs were isolated by CD31 MicroBeads. All MicroBeads were purchased from Miltenyi Biotec GmbH.

**Flow cytometry**. Leukocytes were isolated from the peripheral blood and the lungs. The single cell suspensions from the lung tissues were prepared by the same methods described in MACS. After lysing red blood cells using an ammonium chloride-containing buffer, cells were stained with the relevant antibodies (CD45.2, 109814, BioLegend; Ly6G, 560599, BD Biosciences), assessed by flow cytometry using a FACSCanto II (BD Biosciences) and analyzed by FlowJo (version 10.2, Tree Star Inc.)[22]. HCT116 cells were collected and incubated with an anti-ALK1 antibody (14745-1-AP, Proteintech) followed by R-PE-conjugated donkey anti-rabbit secondary antibody (711-116-152, Jackson ImmunoResearch). The gating strategies are provided in Supplementary Fig. 33.

**Transwell chemotaxis assays**. Chemotaxis in neutrophils from mouse blood was assessed using CytoSelect 96-well (3 μm, Fluorometric Format) according to the manufacturer's protocol.

**Colony assay**. The MACS-isolated lung CD117+ cells were cultured in 1 mL of MethoCult M3434 (Stemcell Technologies) on a 35-mm plate. After 7 days, types of colonies and colony numbers were determined based on manufacturer's instructions. Images were captured by BZ-X700 microscope.

**RNA sequencing**. RNA from MACS-isolated Ly6G+ cells from the BM, PB and lungs was purified using an RNeasy Plus Micro Kit (Qiagen) according to the manufacturer's protocol. RNA concentrations and integrities were evaluated using the TapeStation (Agilent). Total RNA was subjected to reverse transcription and amplification with the SMARTer Ultra Low Input RNA Kit for Sequencing (Clontech). After sonication using ultrasonicator (Covaris), the libraries for RNA sequencing were generated from fragmented DNA with 10 cycles of amplification using a NEB Next Ultra DNA Library Prep Kit (New England BioLabs). After the libraries were quantified using the TapeStation (Agilent), the samples were subjected to sequencing with Hiseq2500 (Illumina) and 61 cycles of the sequencing reactions were performed. TopHat2 (version 2.0.13; with default parameters) and Bowtie2 (version 2.1.0) were used for alignment to the reference mouse genome (mm10 from the University of California, Santa Cruz Genome Browser; http://genome.ucsc.edu/). Levels of gene expression were quantified using Cuffdiff (Cufflinks version 2.2.1; with default parameters). We also used the data from our previous study's RNA sequencing of flow cytometry-sorted LSK cells[22].

**Analyses of pathways and gene set enrichment**. Affected pathways or gene set enrichment were compared among Ly6G+ cells from the lungs, PB, and BM, and LSK cells from our previous study[22] using the comparison analysis in IPA™ (Ingenuity Pathways Analysis, Qiagen) or Gene Set Enrichment Analysis (GSEA, Broad Institute), respectively, according to the RPKM + 1 value for each gene determined by RNA sequencing.

**Immunoprecipitation**. Samples of JAK2[V617F] mouse lung tissue were lysed with lysis buffer (75 mmol/L NaCl, 50 mmol/L Tris-HCl, 0.5% Nonidet P-40, pH 8.0) with a protease inhibitor cocktail. Protein was subjected to immunoprecipitation using protein A-coupled magnetic beads (Thermo Fisher Scientific) and an anti-HIF1α antibody (36169, Cell Signaling Technology) for 1 h at room temperature. Rabbit IgG was used as control.

**Preparation of primary pulmonary arterial smooth muscle cells (PASMCs) and assessment of proliferation using neutrophil-derived conditioned medium**. Mouse PASMCs were isolated from WT mice with a C57BL/6 J background by enzymatic dissociation of the minced lung with collagenase type II (Worthington)[58] and cultured in DMEM (Wako) containing 20% fetal bovine serum. The PASMCs were seeded in 96-well plates or on coverslips in 24-well plates. Conditioned medium from neutrophils in hypoxia incubator chamber (10% $O_2$, ASTEC) 3 h after incubation was collected. The neutrophils were pretreated with Echinomycin (Sigma) prior to hypoxia for 1 h. Then, the PASMCs were incubated with the conditioned medium for 48 h and then subjected to CellTiter 96 AQueous One Solution Cell Proliferation Assay (Promega) and immunofluorescent analysis with anti-Ki67 (NB600-1252, Novus Biologicals) and αSMA (19245, Cell Signaling Technology) antibodies.

**Cell culture**. JAK2[V617F/+] knock-in HCT116 cells as well as wild-type *JAK2*+/+ HCT116 cells were purchased from Horizon Discovery Ltd. The cells were cultured in RPMI 1640 (Sigma) containing 2 mM L-glutamine and 25 mM sodium bicarbonate supplemented with 10% FBS, 100 mg/mL of streptomycin and 100 IU/mL of penicilin at 37 °C in the presence of 5% $CO_2$. Recombinant human BMP9 was purchased from Biolegend, Inc. Cells were transfected with scrambled negative control siRNA (1022076, Qiagen) or ACVRL1-specific siRNA (VHS41063, 129001, Thermo Fisher Scientific) using Lipofectamine RNAiMAX (Thermo Fisher Scientific) according to manufacturer's instructions.

**Prediction of STAT binding sites on *ACVRL1* promotor**. To search for putative STAT binding sites on *ACVRL1* promotor, the in silico analysis was performed using the online databases JASPAR and TFBIND/TRANSFAC[62].

**ChIP-qPCR**. ChIP assays were performed using SimpleChIP enzymatic chromatin IP kit with magnetic beads (9003, Cell Signaling Technology). The crosslinked chromatin was digested with micrococcal nuclease followed by sonication to break into 150–900 bp fragments. Immunoprecipitation was performed using anti-STAT3 (4904, Cell Signaling Technology) or Rabbit IgG. The enriched fragments were purified and analyzed by qPCR. The signal relative to input was evaluated using the formula as follows; percent input = 2% × $2^{(CT\ 2\%\ input\ sample\ –\ CT\ IP\ sample)}$, where CT indicates threshold cycle of qPCR reaction; IP, immunoprecipitation. The qPCR primers used are listed in Supplementary Table 5.

**Construction of DNA plasmid and dual luciferase assay**. The putative human *ACVRL1* promoter sequence (GeneBank: NC_000012.12, position 51906383 to 51907627) was amplified by the forward primer; 5′- GGGGGTACCATAACCAGGA GGCTAGG-3′ and the reverse primer; 5′-TTTAAGCTTCGCGGCCGCAGTTG-3′. The obtained fragment was then subcloned into pGL3-basic vector (Promega) at the KpnI and HindIII sites[29]. The construct was verified by restriction digestion and DNA sequencing. The pGL3-basic vector containing the putative *ACVRL1* promoter region and pNL1.1.TK [Nluc/TK] as a control vector were co-transfected by using ScreenFect A Plus (Wako) according to the manufacturer's protocol. The promoter activity of *ACVRL1* was determined by using Dual-Glo Luciferase Assay System (Promega). The cells were incubated with ruxolitinib (Novartis Pharmaceuticals) or stattic (Cayman Chemical) for 24 h prior to the luciferase assay. Each experiment was performed in duplicate.

**Administration of ALK1/2 inhibitors**. The ALK1/2 inhibitor, K02288 (12 or 24 mg/kg body weight, Selleck Chemicals) or LDN-212854 (9 mg/kg body weight, Selleck Chemicals), dissolved in DMSO was administered to mice via an intraperitoneal injection a week for 2 weeks. DMSO was used as a control.

**Human blood samples and clinical data**. We prospectively analyzed the blood samples taken from patients with PH ($n = 70$) and control subjects ($n = 83$) between April 2018 and April 2020 at Fukushima Medical University Hospital. PH was diagnosed according to the 2015 European Respiratory Society guidelines[1] by independent cardiologists. For the control group, we recruited healthy volunteers or patients with no history of PH or no history of cardiopulmonary diseases. The blood samples were collected in a polypropylene tube containing EDTA-2Na (TERUMO). Genomic DNA was extracted from 200 μL whole blood by using a QuickGene DNA whole blood kit. The *JAK2*V617F VAF was determined by an allelic discrimination PCR assay using THUNDERBIRD Probe qPCR Mix (TOYOBO) in a QuantStudio 3 real-time PCR system (Thermo Fisher Scientific). We used the primers, probe and protocols described in Assay 5 in previous literature (Supplementary Table 6)[32]. The *JAK2*V617F VAF was calculated by Delta Ct ($Ct_{JAK2V617F}$ - $Ct_{wild-type}$) and expressed as the percentage of *JAK2*V617F divided by total *JAK2* (*JAK2*V617F / *JAK2*V617F + *JAK2*wild-type)[63]. Clinical information, including hospital laboratory data, echocardiographic analysis and hemodynamic assessment by right heart catheterization, was collected with our standard clinical practice[64,65].

**Ethical statement**. All animal studies were reviewed and approved by the Fukushima Medical University Animal Research Committee (approval number;

2019084). The protocols were compliant with relevant ethical regulations, and all experiments were performed in accordance with the guidelines provided in the Guide for the Use and Care of Laboratory Animals from the Institute for Laboratory Animal Research. All efforts were made to minimize the suffering of the animals. The protocols for human participants were approved by the institutional ethics committee of Fukushima Medical University Hospital (approval number; 29348). Written informed consent was given by all subjects. This study complied with all relevant regulations regarding the use of human study participants and was conducted in accordance to the criteria set by the 1975 Declaration of Helsinki.

**Statistical analysis**. Comparisons of values between two groups were performed by the unpaired or paired Student's *t*-test, or Mann-Whitney U-test. When more than two groups were evaluated, one-way ANOVA or two-way ANOVA was performed followed by multiple comparisons with the Tukey test. Categorical variables were compared using Fisher's exact test or Chi-square test. Statistical analyses were performed using the Statistical Package for Social Sciences version 26 (SPSS Inc) or GraphPadPrism version 8.1.2 (GraphPad Software). A value of $P < 0.05$ was considered statistically significant.

**Reporting summary**. Further information on research design is available in the Nature Research Reporting Summary linked to this article.

## Data availability

The RNA sequencing data generated in this study have been deposited in the DNA Data Bank of Japan database under accession code DDBJ PRJDB9389. The putative STAT binding sites were assessed using JASPAR [http://jaspar.genereg.net/] and TFBIND/ TRANSFAC [https://tfbind.hgc.jp/] databases. Source data are provided with this paper. Any remaining raw data will be available from the corresponding author upon reasonable request. Source data are provided with this paper.

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

## Acknowledgements

We thank Ms Tomiko Miura and Ms Shoko Sato in the Department of Cardiovascular Medicine, Fukushima Medical University, and Ms Chisato Kubo in the Office for Gender Equality Support, Fukushima Medical University, for their technical assistance. This work was supported by JSPS KAKENHI grant JP19K17609 to YK and JP19K08523 to YT, and research grant from the Uehara Memorial Foundation 201890006 to KI.

## Author contributions

Y.K. and T.M. designed the research, performed the experiments, analyzed the results, and wrote the manuscript. T.Y., K.W., K.U., K.S., and K.M. performed the experiments and analyzed the results. M.O., S.K., and A.I. performed and analyzed the RNA sequencing, supervised the research, and wrote the manuscript. KN and TI supervised the study. K.S. and K.S. provided JAK2V617F mice and interpreted the results. K.I. designed the research, analyzed the data, and wrote the manuscript. Y.T. designed and supervised the research and approved the final version of the manuscript.

## Competing interests

T.M. department is supported by Fukuda Denshi Co., Ltd., Japan. T.Y. and K.S. department is supported by Actelion Pharmaceuticals Japan, Ltd., Japan. Ruxolitinib was provided by Novartis Pharmaceuticals to K.I.. These companies were not associated with the contents of this study. All other authors declare no competing interests.
