## [Peer Review File · Nature Communications]

REVIEWER COMMENTS

Reviewer #1 (Remarks to the Author):

MAJOR COMMENTS

General: There is a major problem throughout this paper with the term “clonal hematopoiesis”. The authors seem to use the term to mean the presence of a mutation ie in JAK2 in the absence of overt hematologic abnormalities. However, the highly artificial situation they have produced in their mouse model and the experiments performed has little relevance to actual human CH. Mice do not develop CH naturally. The models utilized in the paper involve mice that are transgenic, or a transplant model with ablative radiation conditioning into WT mice, to document the intrinsic HSPC origin of the PH (which is important but not relevant to CH per se!). They have done NO experiments to document that PH can develop with the much lower allele frequencies associated with CH without overt hematologic abnormalities in humans, and no experiments to show that the neutrophils that accumulate in the lung are specifically the mutant ones (which is what they assume) versus potentially a paracrine activation effect on both mutant and WT cells. See specific major experimental and interpretive suggestions below. Throughout the Abstract and Discussion, findings are completely oversold and continually confused by this emphasis on clonal hematopoiesis. This is unfortunate, because the core findings as presented in the paper are interesting, novel, possibly clinically relevant, and worthy of rigorous extension and follow-up.

Specific:

1. The core experiments in the paper are the transplantation of mutant vs WT marrow into WT hosts (but Figure 2A has an error, shows transplant of mutant into mutant recipients, should be WT). It seems like in the first experiment using the non-double TG non-GFP marrow, despite 900rads, chimerism was not complete, at least in the mutant into WT transplants, with JAK2 mutant allele frequencies at 4 weeks less than 50%, increasing by 8 weeks whether in hypoxia or not. You don't use another measure for engraftment/chimerism which is unfortunate (ie Ly 5./5.2 congenic system) so can't tell if WT into WT also increased chimerism by 8 weeks, to know whether increase had anything to do with mutation or just the transplant model. Regardless, the mutant into WT mice got the PH phenotype, with hypoxia and at time chimerism was on average 50%. This is NOT typical CHIP type clonal hematopoiesis as seen in humans. These studies must be repeated using different ratios of mutant to WT in the transplanted marrow to have ANY relevance to human CH-what is the minimum level of mutant HSPCs that results in the phenotype? One could look at 2%, 10% and 50% and 100% mutant/WT ratios (perhaps these could be combined with next set of experiments by using a mixture of GFP-JAK2 double TG cells and WT without GFP cells)

2. In the double transgenic studies regarding source of neutrophils in lungs, it would be of great interest to show whether the mutant neutrophils have an intrinsic increased migration capability into the lungs, or whether the effect is indirect and some paracrine effect makes WT neutrophils migrate as well. Your experiment showed 100% of the neutrophils in lung were GFP + in both WT-GFP and mutant-GFP transplants, (I assume also 100% chimeric in blood and BM in this experiment versus the prior one as well, but no data included on that). So you can't distinguish any selective migration (despite what you claim in the text). Thus as in point #1, carrying out a set of experiments with different ratios of mutant/WT then sorting neutrophils from lung and looking at allele ratio there versus in the blood or as I noted above, %GFP positive versus the blood if you mix GFP-JAK HSPCs with WT non-GFP HSPCs, to prove selective migration into the lung, would be important.

3. You have done no experiments that prove the neutrophils migrated in from blood and were produced in BM. That seems likely but proliferation or maturation from precursors in lung seems possible. PH patients with or without MPDs have increased circulation of CD34+ HSPCs, so they could lodge in lung and complete maturation there.

4. You continually refer to "different levels of myeloid maturation" being looked at, but all you have compared are bulk LSKs from the BM, and then mature neutrophils in BM, PB and lung. You assume that lung are more mature than PB which are more mature than BM, but that is not an assumption you can support by any data shown. Senescent neutrophils return to the BM to die. No need to repeat all these studies, with further fractionation of myeloid development in my opinion, not crucial, but just stop referring to having investigated a full developmental differentiation pathway.

5. The ALK1 target is fascinating, but confusing, given that in congenital PH the ALK1 mutation is a LOSS of function, so how does increase in ALK1 expression lead to the same phenotype? The Discussion only mentions the paradox, and seems to suggest that expression in lung may have different (opposite?) impact from myeloid expression. Just discuss this more clearly and admit the paradox very explicitly.

6. The data on humans is not very convincing, although interesting. Just VERY hard to imagine how miniscule clones of less than 1% which would not even "count" as CHIP could possibly have an impact (especially given the lack of data in your mouse model that anything less than 50% can lead to a phenotype). If you drop out those two patients, there is no statistical significance. You need larger cohorts, or better an independent validation cohort from elsewhere (at least sequencing PH patients even if no concurrent controls). If you don't do the major studies suggested above to show smaller levels of mutant cells could have an impact, this human CH part of the paper doesn't even fit because none of the mouse studies are relevant to human CH as presented.

MINOR

1. In Figure S1, why were whole lung homogenates utilized rather than hematopoietic cells? Not perhaps that important to repeat with fractions of lung vs hematologic cells, but at least make it clear in text that this was on whole lung and not further fractionated to find the source of the increased STAT3P

2. Are the authors sure the right heart changes/PH markers in Figure 1 are specific to JAKV617 and not to the chronic elevations of WBC and platelets in the transgenic animals? Are there models with elevated counts but without JAK2 activation that have been shown not to develop PH? This is again not a super important set of experiments to perform if not straightforward, but at least point out this possible caveat. The results of the transplantation studies in Figures 2 and 3 make another explanation based on WBC and plt elevation less likely.

3. The mice only develop PH pathology in all the experiments with chronic hypoxia yet humans with MPDs and PH develop it in the setting of normoxia. At least make this discrepancy clearer in the Discussion.

4. English syntax throughout is understandable, but needs work.

5. Figure 2A-mistake in labeling? Weren't recipients WT not JAK2V617F mutant?

6. Figure 3DE poorly explained-also how many cells counted?

7. Figure 8 should show allele fractions- instead of only having in supplement

8. There is some pretty relevant literature not cited or discussed regarding transfer to PH phenotype in various murine PH genetic models, as well as via engraftment of HSPC from PH patients into xenografts resulting in PH (10.1182/blood-2012-03-419275). So the myeloid/HSPC intrinsic capability to produce PH not limited to JAK2 activating mutations.

Reviewer #2 (Remarks to the Author):

The present study claims that clonal haematopoiesis with JAK2V617F mutation accelerates pulmonary hypertension (PH) in mice via neutrophils accumulation in pulmonary arterial regions through the receptor ALK1. Although the PH phenotype of these mice seems clear, the mechanism of action presented by the authors involving ALK1 is far from being demonstrated and requires further experiments to be validated. The manuscript addresses a very important question for the scientific community as pulmonary hypertension is a life-threatening disease without cure and a better understanding of its underlying mechanisms is an essential point for cardiovascular community. Here the authors address the link of myeloproliferative neoplasms (MPNs) and PH.

Major points

1) The authors wanted to know the involvement of JAK-STAT pathway in PH development and saw that STAT3 phosphorylation was significantly increased in PH treated mice. To clarify this effect, they tested their JAK2V617F mice, whose mutation is the most frequent mutation among MPNs and that causes JAK-STAT activation, in a model of PH which chronic hypoxia (10%O₂). Strikingly, they found that JAK2V617F exacerbated PH in mice and that JAK2V617F-expressing neutrophils were specifically accumulated in pulmonary arterial regions. This is a very interesting result that would be strengthened if confirmed in another PH model such as monocrotaline injection.

2) To elucidate the underlying mechanisms, gene expression profiling of neutrophils, at several stages of differentiation, was performed by RNA sequencing in sorted Ly6G⁺ cells from BM, peripheral blood (PB) and lungs in JAK2V617F mice in comparison to WT mice. A gene set enrichment analysis revealed the canonical IL6-JAK-STAT3 pathway and interestingly the receptor ALK1 was found to be the highest upregulated gene in this pathway in Ly6G⁺ neutrophils of the lung as well as PB. This result is very interesting but rather surprising as ALK1 is an endothelial-specific receptor. This unexpected result needs to be supported by further experiments.

- First, the purity of the Ly6G⁺ cells enrichment (98%) should be checked by other means than just May-Giemsa staining. They should test some endothelial cell markers in this purification by PCR and /or IF, to check if they find any vascular enrichment.

- Second, they show that *Acvr11* mRNA level in whole lung extract is also increased in JAK2V617F mice under hypoxia (Fig5a). As *Acvr11* expression in the lungs is predominantly expressed by endothelial cells, it is rather surprising to observe such a high increase in *Acvr11* that would be due to neutrophils. ALK1 levels are particularly high in lungs as this tissue contains a very high number of endothelial cells.

It is interesting to note, although not discussed in the text, that *Acvr11* levels in this figure is also increase in wt mice under hypoxia. The authors should isolate endothelial cells and neutrophils under normoxia and hypoxia (the RNAseq analysis was performed under normoxia) from the lungs

of these mice and check in which cell type is *Acvr1* increased. It would be interesting to compare the number of *Acvr1* copies in neutrophils versus endothelial cells.

- The authors should comment on the difference between the 10-fold increase in *Acvr1* mRNA levels versus the 2-fold increase in pSmad1/5 levels (Fig5a and b).

- The authors should measure the level of *Acvr1* mRNA encoding the receptor *Alk2* expression, which highly related to ALK1 and phosphorylates Smad1/5/8, and in contrast to ALK1 ubiquitous. A pSmad1/5/8 immunostaining in lungs would be interesting to discriminate neutrophils pSmad1/5/8 from endothelial pSmad1/5/8 staining.

3) To confirm ALK1 upregulation by JAK2V617F, they then use the commercial cell line HCT116 carrying the JAK2V617F mutation. HCT116 is colorectal carcinoma cell line that should not express *Acvr1* so the choice of the cell line is rather surprising in particular when they want to study the promoter of this gene which is endothelial specific. Please comment. Again, it will be important to know the number of copies of *Acvr1* in these cells as compared to endothelial cells (Fig 6a). In order to check whether there is a significant level of active/functional ALK1 on these cells, the authors could stimulate these cells with BMP9, the high affinity ALK1 ligand.

4) The in-silico analysis identified several putative STAT3 binding sites in *ACVRL1* promoter region in both humans and mice (Fig. 6d), that were not mentioned in the publication cited by the authors and only one STAT3 binding site is present in the putative promoter sequence from -1035 bp to +210 bp of the transcriptional start that the author studied. So, the involvement of STAT3 in the regulation of this promoter is not that strongly supported by in-silico data. Although the experiment with Ruxolitinib, a specific JAK1/2 inhibitor (Fig. 6f), supports a role for the JAK pathway in *ACVRL1* promoter activity, no experiment was performed to conclude that it is STAT3 as mentioned in the discussion (lane 361). The authors should just conclude that ALK1 promoter activity in HCT116 cells seems to be JAK1/2 dependent.

5) Very interestingly, to support their ALK1 hypothesis, the authors performed some in vivo experiments in order to block ALK1 and test whether it would reverse their phenotype. For this, they used the inhibitor K02288. First, it is quite surprising to see no reference concerning this inhibitor in the manuscript and this should be corrected (Santivale et al. 2013, Plos One) and second, they do not mention that this inhibitor is not specific to ALK1 but also inhibits ALK2 and that this inhibitor is most of the time proposed as an ALK2 inhibitor with a slightly higher affinity for ALK2 than ALK1. This is why, it is important to measure ALK2 levels in neutrophils as ALK2 is a more widely expressed receptor. *BMP2* levels which plays a much preponderant role than ALK1 in PH should also be measured. Together, from these experiments it can only be concluded that the ALK1/2 pathway is important for PH development in accordance with a recent paper that they cite (Li et al., *Circ Res*, 2019). It is however surprising that K02288 administration did not significantly change the levels of RVSP or RV hypertrophy in chronic hypoxia-exposed WT mice as recently published in the above cited paper. These experiments need to be performed using other ALK1 inhibitors and the effect of these inhibitors need to be also tested under normoxia.

Minor points:

- 1) Lane 257: please indicate in the text what cell line is used.
- 2) Fig6e, please precise the cell type transfected in the figure legend.

Reviewer #3 (Remarks to the Author):

General comments:

In this study, Kimishima et al. have investigated the effect of clonal somatic mutation JAK2V617F on PH development. The JAK-STAT3 pathway has since long been known to be involved in PH. JAK2V617F, even though not the most common mutation among patients suffering from myeloproliferative neoplasms, it is associated with vascular complications. The authors have demonstrated that clonal hematopoiesis with JAK2V617F could exacerbate chronic hypoxia-induced PH and pulmonary vascular remodeling in mice. Both JAK2V617F transgenic and wild type recipient mice transplanted with JAK2V617F developed an exacerbated PH upon chronic hypoxia via accumulation neutrophils in perivascular regions of lungs. They show that JAK2V617F-mediated STAT3 phosphorylation could upregulate ALK1-Smad1/5/8 signaling resulting in enhanced PH phenotype. The findings are interesting. However, several points need specific attention.

Major comments:

1. A main concern is a lack of data showing how chronic hypoxia can trigger JAK2V617F PH phenotype. Is it HIF regulated activation? Are these neutrophils more likely to invade the perivascular area in the lung?
2. The authors showed that approximately 40% of the perivascular GFP positive cells are Ly6G positive (neutrophils). However, it remains unclear what are the remaining GFP positive cells. In order to not underestimate the contribution of other BM-derived cell types, I would suggest to use either IF staining for other markers such as CD68 or extensive flow cytometry to identify these cells. Furthermore, CD68 IHC staining shown in Supplemental Figures 2 and 3 is not convincing. Double immune staining including CD45 and presenting different cell types of interest as a percentage of CD45 positive perivascular cells is recommended.
3. A more detailed characterization of pulmonary hemodynamics and cardiac function would be helpful. It is not clear if cardiac function is affected and whether changes in cardiac output contribute to the changes observed for RVSP. Also, it would be interesting to see whether rheological effects of leukocytes and thrombocytes affects PH.

4. In the experimental mouse model of chronic hypoxia-induced PH, it has been shown that the number of proliferating cells increased in the pulmonary vasculature. These proliferating cells are usually found in the vessel wall and could be characterized by PCNA positive nuclei. The IHC staining for PCNA in Supplemental Figures 2 and 3 was not successful and it could not be quantified (cytoplasm and even blood seems to be stained). I suggest repeating the staining or using another proliferation marker. Accordingly, the authors should change the statements through the manuscript such as line 130-133: Proliferating cell nuclear antigen (PCNA)-positive cells in peri-vascular regions were comparable between WT lungs and JAK2V617F lungs, suggesting that the accumulated Ly6G+ cells were not on the proliferative state (Supplementary Fig. 2b).

5. It is not clear why the authors used only female mice. Is there any particular reason why different time points of hypoxic exposure (2 and 3 weeks respectively) were used in different parts of this study?

The myeloproliferative neoplasms (MPNs) are diseases occurring in later ages (Srouf, PMID: 28387461). Therefore, it would be worth to characterise JAK2V617F transgenic mice in later age to identify if somatic mutations in JAK could drive PH development alone, without hypoxia stimulus.

6. The authors should provide data from normoxic experiments in figure 7.

7. As nicely pointed by authors, the role of ALK1 in PH development is complex. Recent publications of Tu et al. (PMID: 30636542) showed that ALK1 ligand trap counterintuitively inhibited PH. Could the authors explain why inhibition of ALK1 by K02288 attenuated chronic hypoxia-induced PH only in JAK2V617F mice and did not affect WT mice? How was the dose of K02288 chosen for? Is it possible that in high dose K02288 could also affect EC or PASMC proliferation? Could the authors prove that K02288 specifically affect neutrophils and did not affect directly EC/SMC?

8. Please clarify how pulmonary arteries were distinguished from veins for the quantification of pulmonary vascular remodelling. Furthermore, what was the selection criteria for characterization of non-, partially- or fully-muscularized?

9. Since GFP positive cells are stained using GFP antibody, it would be useful to include WT animals without GFP as a negative control.

Minor comments:

1. Please provide LV+S values as a change in LV+S weight affects $RV/(LV+S)$ and can thus suggest a false positive or negative effect on RV hypertrophy

2. Please unify the presentations of statistics through the figures.

To Reviewer #1:

Thank you very much for reviewing our manuscript (NCOMMS-20-18526) and providing us with valuable comments. Our responses to your comments are as follows,

MAJOR COMMENTS

General: There is a major problem throughout this paper with the term “clonal hematopoiesis”. The authors seem to use the term to mean the presence of a mutation ie in JAK2 in the absence of overt hematologic abnormalities. However, the highly artificial situation they have produced in their mouse model and the experiments performed has little relevance to actual human CH. Mice do not develop CH naturally. The models utilized in the paper involve mice that are transgenic, or a transplant model with ablative radiation conditioning into WT mice, to document the intrinsic HSPC origin of the PH (which is important but not relevant to CH per se!). They have done NO experiments to document that PH can develop with the much lower allele frequencies associated with CH without overt hematologic abnormalities in humans, and no experiments to show that the neutrophils that accumulate in the lung are specifically the mutant ones (which is what they assume) versus potentially a paracrine activation effect on both mutant and WT cells. See specific major experimental and interpretive suggestions below. Throughout the Abstract and Discussion, findings are completely oversold and continually confused by this emphasis on clonal hematopoiesis. This is unfortunate, because the core findings as presented in the paper are interesting, novel, possibly clinically relevant, and worthy of rigorous extension and follow-up.

Response;

Thank you very much for pointing out such important matters. We have performed further experiments according to the reviewer’s specific comments as below.

Specific:

1. The core experiments in the paper are the transplantation of mutant vs WT marrow into WT hosts (but Figure 2A has an error, shows transplant of mutant into mutant recipients, should be WT). It seems like in the first experiment using the non-double TG non-GFP marrow, despite 900rads, chimerism was not complete, at least in the mutant into WT transplants, with JAK2 mutant allele frequencies at 4 weeks less than 50%, increasing by 8 weeks whether in hypoxia or not. You don’t use another measure for engraftment/chimerism which is unfortunate (ie Ly 5./5.2 congenic system) so can’t tell if WT into WT also increased chimerism by 8 weeks, to know

whether increase had anything to do with mutation or just the transplant model. Regardless, the mutant into WT mice got the PH phenotype, with hypoxia and at time chimerism was on average 50%. This is NOT typical CHIP type clonal hematopoiesis as seen in humans. These studies must be repeated using different ratios of mutant to WT in the transplanted marrow to have ANY relevance to human CH-what is the minimum level of mutant HSPCs that results in the phenotype? One could look at 2%, 10% and 50% and 100% mutant/WT ratios (perhaps these could be combined with next set of experiments by using a mixture of GFP-JAK2 double TG cells and WT without GFP cells).

Response;

According to the reviewer's kind suggestion together with your major comment #2, we performed a competitive bone marrow transplantation (BMT) using GFP⁺ bone marrow cells, because we were not able to prepare for enough numbers of Ly5.1 congenic mice but GFP mice were available during the limited revision period. We evaluated the chimerism in the peripheral blood after BMT and expressed it as the ratio of GFP⁺ cells within CD45⁺ cells by flow cytometry. Firstly, in the control non-competitive BMT group, the flow cytometry revealed that the chimerism in the peripheral blood was 89.2 ± 0.6 and 92.4 ± 0.7 at 4 weeks, and 95.1 ± 1.3 and 95.1 ± 0.6 at 8 weeks after BMT using 100% WT-GFP BM cells at normoxia and chronic hypoxia, respectively. Likewise, the chimerism in the recipients transplanted with 100% JAK2^{V617F}-GFP was 80.8 ± 1.7 and 82.5 ± 2.5 at 4 weeks, and 93.5 ± 1.7 and 94.2 ± 0.7 at 8 weeks after BMT at normoxia and chronic hypoxia, respectively. The chimerism at 8 weeks after BMT was significantly increased compared to that in 4 weeks after BMT in the recipient mice transplanted with 100% WT-GFP BM cells and 100% JAK2^{V617F}-GFP BM cells. We note that the differences between the chimerism with proportions of GFP-positive cells and *Jak2*^{V617F} allele frequency in Fig. 2 were due to the heterozygosity of *Jak2*^{V617F} both in JAK2^{V617F} and JAK2^{V617F}-GFP donor BM cells and the difference in the methodological principle between flow cytometry of GFP⁺ cells within CD45⁺ cells and allelic-specific qPCR of genomic DNA. In relation to your minor comment #5, we apologize for the inadequate labeling and confusion in Fig. 2a. We have revised Fig. 2a, accordingly.

Next, we carried out the competitive BMT using different ratios of a mixture of JAK2^{V617F}-GFP and WT without GFP as well as WT-GFP and WT without GFP. We set the ratio of 100%, 50%, 25%, 10%, 2.0%, 0.4% to know the minimum threshold of aggravation of PH in JAK2^{V617F}-GFP-BMT mice. When we analyze the recipients limited to the chimerism with 50-100% as well as 20-49% at 8 weeks after BMT, we found that JAK2^{V617F}-GFP-BMT mice showed significant increases in RVSP and RV/LV+S compared to WT-GFP-BMT mice. Of note, JAK2^{V617F}-GFP-BMT mice with chimerism with 1-19% also showed significant increases in RVSP and RV/LV+S, suggesting that the small clones with JAK2^{V617F} as low as 1% could develop PH in response to chronic hypoxia. Moreover,

JAK2^{V617F}-GFP-BMT mice with lower chimerism less than 1% tended to display increases in RVSP and RV/LV+S compared to WT-GFP-BMT mice. Taken together, these data suggest that not only the recipient mice with JAK2^{V617F} clones with the nearly complete chimerism but also 1-19% chimerism could develop PH, and the clones with low JAK2^{V617F} less than 1% chimerism were potentially associated with development of PH. We have included these data in the Results section, Fig. 4 and Supplementary Fig. 16.

In the revised text;

We next performed a competitive transplantation using different ratios of a mixture of WT-GFP or JAK2^{V617F}-GFP BM cells and WT without GFP BM cells (Fig. 4a, Supplementary Fig. 16a). In the control non-competitive group, flow cytometry showed that the chimerism assessed by GFP⁺ cells within CD45⁺ cells in the blood was significantly elevated at 8 weeks compared to that of 4 weeks in 100% WT-GFP-BMT and 100% JAK2^{V617F}-GFP-BMT mice (Supplementary Fig. 16b). To know the minimum threshold of aggravation of PH in JAK2^{V617F}-GFP-BMT mice, we categorized the recipient mice according to the chimerism at 8 week-point after BMT. Interestingly, when we analyzed the recipients limited to the chimerism with 1-19% as well as with 20-49% and 50-100%, the JAK2^{V617F}-GFP-BMT mice showed significant increases in RVSP and RV/LV+S compared to the WT-GFP-BMT mice (Fig. 4b, c, Supplementary Fig. 16c-e). Moreover, the JAK2^{V617F}-GFP-BMT mice with lower chimerism with <1% tended to display increases in RVSP and RV/LV+S compared to the WT-GFP-BMT mice (Supplementary Fig. 16f). These data suggest that even small clones with *Jak2*^{V617F} are associated with development of PH. (*pages 14-15, lines 224-238*)

2. In the double transgenic studies regarding source of neutrophils in lungs, it would be of great interest to show whether the mutant neutrophils have an intrinsic increased migration capability into the lungs, or whether the effect is indirect and some paracrine effect makes WT neutrophils migrate as well. Your experiment showed 100% of the neutrophils in lung were GFP + in both WT-GFP and mutant-GFP transplants, (I assume also 100% chimeric in blood and BM in this experiment versus the prior one as well, but no data included on that). So you can't distinguish any selective migration (despite what you claim in the text). Thus as in point #1, carrying out a set of experiments with different ratios of mutant/WT then sorting neutrophils from lung and looking at allele ratio there versus in the blood or as I noted above, %GFP positive versus the blood if you mix GFP-JAK HSPCs with WT non-GFP HSPCs, to prove selective migration into the lung, would be important.

Response;

In line with your comment #1, we isolated cell fraction from the lungs and the peripheral blood in WT-GFP-BMT and JAK2^{V617F}-GFP-BMT mice with 1-19% chimerism at 8 weeks after BMT. Then, we performed flow cytometry to clarify whether JAK2V617F neutrophils have an intrinsic increased migration capability into the lungs. Flow cytometry demonstrated that the percentages of JAK2^{V617F}-GFP⁺ cells within Ly6G⁺ neutrophils were significantly higher in the lung than in the blood while that of WT-GFP⁺ cells was not different between the lung and the blood. These data suggest that the presence of JAK2V617F leads to selective migration of neutrophils from peripheral blood into the lung tissue, especially in response to chronic hypoxia. We have included these data in the Results section and Fig. 4.

In the revised text;

We isolated cell fraction from the lungs and the blood in WT-GFP-BMT and JAK2^{V617F}-GFP-BMT mice with 1-19% chimerism at 8 weeks after BMT. The percentages of GFP⁺ cells within Ly6G⁺ neutrophils in JAK2^{V617F}-GFP-BMT mice were significantly higher in the lungs than in the blood while those in WT-GFP-BMT mice were not different between the lungs and the blood (Fig. 4c, d). These findings suggest that JAK2^{V617F} neutrophils have an intrinsic increased migration capability into the lungs, and the migration is more enhanced in response to hypoxia. (pages 15, lines 242-248)

3. You have done no experiments that prove the neutrophils migrated in from blood and were produced in BM. That seems likely but proliferation or maturation from precursors in lung seems possible. PH patients with or without MPDs have increased circulation of CD34+ HSPCs, so they could lodge in lung and complete maturation there.

Response;

As we described in your comment #2, we showed significant increases in the percentages of JAK2^{V617F}-GFP within the Ly6G⁺ cells in the lungs in comparison to that in the peripheral blood, suggesting that JAK2V617F neutrophils selectively migrate from the blood into the lung. We carried out *ex vivo* analysis using the chemotaxis assay, showing that JAK2^{V617F} neutrophils in the peripheral blood displayed more capability of migration compared to WT neutrophils. To clarify the possibility of proliferation or maturation from hematopoietic progenitor cells in the lung, we performed a colony-forming assay using MACS-isolated CD117 (c-kit)⁺ cells from the lung. There were substantial increases in the colony-forming ability of JAK2V617F-expressing progenitor cells toward the myeloid lineage. These data suggest that precursor cells in JAK2^{V617F} lungs display the capacity to lodge and complete maturation in the lung. Thus, we suggest that JAK2V617F induced the selective neutrophil migration into the lungs as well as proliferation/maturation from precursors in the lungs. We agree

with the reviewer that PH patients with or without MPNs have increased circulation of CD34⁺ hematopoietic stem and progenitor cells. We have shown these data in the Results section and Fig. 4 and Supplementary Fig. 17. We have also described in the Discussion section and cited a new ref. #34 (Blood 2011; 117: 3485-3493).

In the revised text;

Accordingly, *ex vivo* analysis using the chemotaxis assay revealed that JAK2^{V617F}-Ly6G⁺ cells in the blood displayed more capability of neutrophil migration than in WT-Ly6G⁺ cells (Fig. 4e). To investigate the involvement of hematopoietic progenitors in JAK2^{V617F} lungs, CD117 (c-kit)⁺ cells were sorted from the lungs and subjected to a colony-forming assay. There were substantial increases in the colony-forming ability of JAK2V617F-expressing progenitor cells especially toward the myeloid lineage (Fig. 4f, Supplementary Fig. 17). These data indicate that the accumulated Ly6G⁺ neutrophils carrying JAK2V617F are migrated from BM into pulmonary arterial regions and potentially proliferated and matured from the precursors in the lung. (*pages 15-16, lines 248-257*)

Moreover, it is possible that the JAK2^{V617F} hematopoietic precursor cells in the lung can display the capacity to lodge and complete maturation there. PH patients with or without MPNs have increased circulation of CD34⁺ hematopoietic stem and progenitor cells (ref. #34). (*page 26, lines 434-436*)

4. You continually refer to “different levels of myeloid maturation” being looked at, but all you have compared are bulk LSKs from the BM, and then mature neutrophils in BM, PB and lung. You assume that lung are more mature than PB which are more mature than BM, but that is not an assumption you can support by any data shown. Senescent neutrophils return to the BM to die. No need to repeat all these studies, with further fractionation of myeloid development in my opinion, not crucial, but just stop referring to having investigated a full developmental differentiation pathway.

Response;

We agree with the reviewer that senescent neutrophils could return to the BM to die and thus the myeloid maturation not always directs from BM to the lung. Accordingly, we have revised the terms related to “different levels of myeloid maturation” or “full developmental differentiation pathway” throughout the texts.

5. The ALK1 target is fascinating, but confusing, given that in congenital PH the ALK1 mutation

is a LOSS of function, so how does increase in ALK1 expression lead to the same phenotype? The Discussion only mentions the paradox, and seems to suggest that expression in lung may have different (opposite?) impact from myeloid expression. Just discuss this more clearly and admit the paradox very explicitly.

Response;

Although it was reported that *ACVRL1* mutations in hereditary hemorrhagic telangiectasia led to a loss of function (ref. #42, 43), the functional relevance of congenital *ACVRL1* mutations in PH remains undetermined whether the mutations result in a gain of function or a loss of function. General ALK1 heterozygous knockout mice developed PH naturally in young adulthood (ref. #44) whereas systemic administration of ALK1 inhibitor, a ligand trap targeting ALK1's ligands, prevented the chronic hypoxia-induced PH in the adult mice (ref. #45). Accordingly, it remains undefined whether ALK1 expression has beneficial or detrimental effects in PH, but the present study revealed that JAK2V617F-related ALK1 upregulation in hematopoietic cells had detrimental effects in PH. Thus, as the reviewer suggested, ALK1 expressions in myeloid cells may have a different impact from those in the lung such as endothelial cells. We have described these points in the Discussion section.

In the revised text;

Although it was reported that *ACVRL1* mutations in hereditary hemorrhagic telangiectasia led to a loss of function, the functional relevance of congenital *ACVRL1* mutations in PH has not been fully determined (ref. #42, #43). (page 28-29, lines 478-481)

We revealed that JAK2V617F-related ALK1 upregulation in myeloid cells had detrimental effects in PH, suggesting that ALK1 expressions in myeloid cells may have a different impact from the lung endothelial cells. (page 29, lines 490-492)

6. The data on humans is not very convincing, although interesting. Just VERY hard to imagine how miniscule clones of less than 1% which would not even “count” as CHIP could possibly have an impact (especially given the lack of data in your mouse model that anything less than 50% can lead to a phenotype). If you drop out those two patients, there is no statistical significance. You need larger cohorts, or better an independent validation cohort from elsewhere (at least sequencing PH patients even if no concurrent controls). If you don't do the major studies suggested above to show smaller levels of mutant cells could have an impact, this human CH part of the paper doesn't even fit because none of the mouses studies are relevant to human CH as presented.

Response;

As we responded to your comment #1, we showed the clones with *JAK2V617F* as low as 1-19% chimerism were significantly associated with PH phenotypes. Moreover, the recipient mice with much lower *JAK2V617F* chimerism (0.1-1%) tended to display increases in RVSP and RV/LV+S. Thus, our murine data suggest that the small clones with *JAK2V617F* have an impact on PH development. Accordingly, other studies demonstrated that the minuscule *JAK2V617F* clones played a biological role in general populations and MPN patients, in terms of either elevation of blood cell counts or thromboembolic complications. In a general population of 19,958 participants from the Danish General Suburban Population Study, there were 613 (3.1%) participants carrying *JAK2V617F*; among them, there were 92 individuals with $\geq 1\%$ *JAK2V617F* VAF and 507 with $< 1\%$ VAF, and MPNs were present in only 14 participants (Blood 2019; 134: 469-479). The minimum VAF of *JAK2V617F* was 0.31% in the MPN group. In the comparison between *JAK2V617F*-positive participants with $< 1\%$ VAF and *JAK2V617F*-negative participants, the $< 1\%$ VAF group had significantly higher leukocyte, neutrophil, and platelet counts. In the VAF $\geq 1\%$ *JAK2V617F*-positive non-MPN participants, although the VAF was low as 1-10% in the majority of VAF $\geq 1\%$ group, the prevalence of venous thromboembolism (deep vein thrombosis and/or pulmonary embolism) was significantly higher than in the nonmutated individuals. In another study, among 42 patients suspected of hematological malignancies with low *JAK2V617F* VAF (0.1–3%), 24 patients received a diagnosis of classical MPNs in a year (Oncotarget 2017; 8: 37239-37249). Moreover, the patients with *JAK2V617F* VAF $< 2\%$, including some cases who eventually developed MPNs, showed similar survival and thrombotic incidence as those with VAF 2-10% (Clin Lymphoma Myeloma Leuk 2020; 20: e569-e578). Taken together, in humans, additional factors such as the genetic background and environmental exposure are involved in the onset and development of PH. The persistent exposure to *JAK2V617F* may be important to the pathogenesis for pulmonary arterial remodeling even with the small clones with *JAK2V617F*. Due to the COVID-19 problems and the limited revision period, we realized that it was difficult to enroll patients with a larger cohort, and there was no appropriate database available that we could refer to in PH patients. We completely agree with the reviewer that we will need to validate our findings in larger cohorts, but we would very much appreciate if you could kindly understand our situations and the investigation by another cohort is beyond the scope of the current study. Although our murine data and human data were not exactly matched, we performed additional experiments as you advised, and our murine data suggest that *JAK2V617F* even with low VAF are associated with PH development. We have discussed in the Discussion section.

In the revised text;

Our murine study demonstrated that even small clones with *JAK2V617F* led to PH development.

Given that *JAK2V617F* VAF as low as 0.1-2% was associated with the elevation of blood cell counts, manifestations of MPNs, thrombotic events, and survival in both *JAK2V617F*-positive general populations without MPNs and patients with MPNs (ref. #47-#50), the presence of *JAK2V617F* with even low VAF may have a clinically biological impact. The relation of *JAK2V617F* VAF levels with a prevalence of PH needs to be studied in these populations. (page 30, lines 502-508)

MINOR

1. In Figure S1, why were whole lung homogenates utilized rather than hematopoietic cells? Not perhaps that important to repeat with fractions of lung vs hematologic cells, but at least make it clear in text that this was on whole lung and not further fractionated to find the source of the increased STAT3P.

Response;

There were technical issues of Western blotting to detect the distinct phosphorylation of STAT3 using MACS-isolated cells from the lung homogenates after collagenase digestion for 30 minutes. Instead, we showed the data from the lung homogenates in Supplementary Fig. 1. We clearly described that the data were from whole lung homogenates, not fractionated cells, in the Results section and Figure legends.

In the revised text;

STAT3 phosphorylation levels on whole lung homogenates, not fractionated cells, were significantly increased after exposure to chronic hypoxia for 3 weeks. (page 7, lines 95-96)

2. Are the authors sure the right heart changes/PH markers in Figure 1 are specific to *JAKV617* and not to the chronic elevations of WBC and platelets in the transgenic animals? Are there models with elevated counts but without *JAK2* activation that have been shown not to develop PH? This is again not a super important set of experiments to perform if not straightforward, but at least point out this possible caveat. The results of the transplantation studies in Figures 2 and 3 make another explanation based on WBC and plt elevation less likely.

Response;

In our BMT studies, the recipient mice transplanted with *JAK2*^{V617F} BM cells did not show increases in the leukocytes or platelets. This finding indicates that the *JAK2* activation in the myeloid cells was associated with PH independently on the leukocyte or platelet counts, as the reviewer suggested. In

relation to your minor comment #8, in mouse xenograft models, hematopoietic progenitor cells from pulmonary arterial hypertension (PAH) patients contributed to pulmonary vascular remodeling and right heart hypertrophy (Blood 2012; 120: 1218-1227. ref. #35). Although these PAH patients had not displayed any hematological disorder, their hematopoietic progenitor cells showed increases in the growth of myeloid colonies and the expression of myeloid transcription factors. These data further support that the activation of myeloid cells including JAK-STAT may lead to PH phenotypes even without elevation of WBCs and PLTs. We have mentioned these in the Discussion section.

In the revised text;

Engraftment of hematopoietic progenitors from PH patients who did not display any hematological disorder into xenografts showed increases in the growth of myeloid colonies and the expression of myeloid transcription factors, resulting in pulmonary vascular remodeling and right heart hypertrophy (ref. #35), suggesting that the intrinsic capability of hematopoietic progenitors is associated with PH. In line with our JAK2^{V617F}-BMT model that did not show elevation of white blood cells or platelets, the activation of the myeloid cells including JAK-STAT may lead to PH phenotypes even without elevation of leukocyte or platelet counts. (*pages 26-27, lines 437-444*)

3. The mice only develop PH pathology in all the experiments with chronic hypoxia yet humans with MPDs and PH develop it in the setting of normoxia. At least make this discrepancy clearer in the Discussion.

Response;

JAK2^{V617F} mice aged 8- to 10-week-old developed PH pathology in response to chronic hypoxia but did not develop PH in the setting of normoxia, indicating that JAK2V617F alone is not sufficient to induce PH in young adulthood. Thus, an additional trigger such as chronic hypoxia was required for PH phenotypes in young JAK2^{V617F} mice. In contrast, patients with MPNs can develop PH in the setting of normoxia but not all MPN patients develop PH (ref. #6). These suggest that an additional genetic and/or environmental hit in addition to JAK2V617F is needed for the onset and development of PH in the predisposed subjects. We described these in the Discussion section.

In the revised text;

JAK2^{V617F} mice developed PH pathology in response to chronic hypoxia but did not develop PH in normoxia, indicating that JAK2V617F alone is not sufficient to induce PH and that a trigger such as chronic hypoxia is required for PH phenotypes in JAK2^{V617F} mice. In contrast, patients with MPNs can develop PH in the setting of normoxia. However, not all MPN patients develop PH. As MPN

occurs in later ages (ref. #36), an additional genetic and/or environmental hit in addition to JAK2V617F may be needed for the onset and development of PH in the predisposed subjects. (*page 27, lines 445-451*)

4. English syntax throughout is understandable, but needs work.

Response;

We have revised the manuscript with had it proofread by two native English-speaking scientific editors.

5. Figure 2A-mistake in labeling? Weren't recipients WT not JAK2V617F mutant?

Response;

We apologize for the inadequate labeling and confusion. We used WT mice as the recipient mice for bone marrow transplantation (BMT). To avoid confusion, we defined the recipient WT mice transplanted with donor JAK2^{V617F} bone marrow cells as “JAK2^{V617F}-BMT” and the recipient WT mice transplanted with donor WT bone marrow cells as “WT-BMT” in the revised manuscript. We have revised Fig. 2a and the text throughout the manuscript.

6. Figure 3DE poorly explained-also how many cells counted?

Response;

We counted at least 100 GFP- or Ly6G-positive cells in each lung section of each mouse. We explained these in Methods and Figure legends in the revised manuscript.

In the revised text;

For quantification of perivascular cellular infiltration, more than 100 cells were counted around the distal pulmonary arteries with a diameter of 50-100 μm in each mouse. (*page 38, lines 627-629*)

More than 100 GFP⁺ cells and Ly6G⁺ cells were counted in each section and expressed as the percentage of the cells. (*Figure legends*)

7. Figure 8 should show allele fractions- instead of only having in supplement

Response;

We have shown the allele fractions in the main Figure (Figure 9b in the revised manuscript), instead of the Supplementary Figure.

8. There is some pretty relevant literature not cited or discussed regarding transfer to PH phenotype in various murine PH genetic models, as well as via engraftment of HSPC from PH patients into xenografts resulting in PH (10.1182/blood-2012-03-419275). So the myeloid/HSPC intrinsic capability to produce PH not limited to JAK2 activating mutations.

Response;

We thank the reviewer for the excellent suggestion. It is reported that engraftment of hematopoietic progenitors from PH patients into xenografts resulted in PH, suggesting the intrinsic capability of hematopoietic progenitors is associated with PH, which may not be limited to JAK2 activating mutations. We have cited the literature as a new ref. #35, and described this point in the Discussion section.

In the revised text;

Engraftment of hematopoietic progenitors from PH patients who did not display any hematological disorder into xenografts showed increases in the growth of myeloid colonies and the expression of myeloid transcription factors, resulting in pulmonary vascular remodeling and right heart hypertrophy (ref. #35), suggesting that the intrinsic capability of hematopoietic progenitors is associated with PH. In line with our JAK2^{V617F}-BMT model that did not show elevation of white blood cells or platelets, the activation of the myeloid cells including JAK-STAT may lead to PH phenotypes even without elevation of leukocyte or platelet counts. (*pages 26-27, lines 437-444*)

Thank you again for your thoughtful comments to our manuscript.

To Reviewer #2:

Thank you very much for reviewing our manuscript (NCOMMS-20-18526) and providing us with valuable comments. Our responses to your comments are as follows,

1. The authors wanted to know the involvement of JAK-STAT pathway in PH development and saw that STAT3 phosphorylation was significantly increased in PH treated mice. To clarify this effect, they tested their JAK2^{V617F} mice, whose mutation is the most frequent mutation among MPNs and that causes JAK-STAT activation, in a model of PH which chronic hypoxia (10%O₂). Strikingly, they found that JAK2^{V617F} exacerbated PH in mice and that JAK2^{V617F}-expressing neutrophils were specifically accumulated in pulmonary arterial regions. This is a very interesting result that would be strengthened if confirmed in another PH model such as monocrotaline injection.

Response;

Thank you for the reviewer's helpful suggestion. We performed additional experiments using the monocrotaline as another PH model referred to the protocol from the previous literature (Cardiovasc Res 2016; 110: 319–330). However, using this protocol (monocrotaline 600 mg/kg weekly for 4 weeks), the monocrotaline-injected mice did not develop PH either in WT mice or JAK2^{V617F} mice as shown below. Monocrotaline mouse model of severe PH has not fully been established as mice metabolize monocrotaline differently from other species (Am J Physiol Lung Cell Mol Physiol 2012; 302: L363-L369).

Instead, we applied a Sugen-hypoxia model as the other PH model (Am J Respir Crit Care Med 2011; 84: 1171-1182, Nat Commun 2019; 10: 4143, Nat Commun 2019; 10: 5183). Using a single weekly injection of VEGF inhibitor SU-5416 at 20 mg/kg followed by 2 weeks of hypoxia (10% O₂), this

Sugen-hypoxia protocol significantly induced both PH in WT mice and JAK2^{V617F} mice, and JAK2^{V617F} mice showed more exaggerated PH than WT mice. We have included these data in Supplementary Fig. 8 and described these in the Results section.

In the revised text;

We confirmed that in a Sugen-hypoxia model, which is another PH model (ref. #25), RVSP and RV hypertrophy were significantly elevated in JAK2^{V617F} mice compared to WT mice (Supplementary Fig. 8). *(page 10, lines 149-152)*

2. To elucidate the underlying mechanisms, gene expression profiling of neutrophils, at several stages of differentiation, was performed by RNA sequencing in sorted Ly6G⁺ cells from BM, peripheral blood (PB) and lungs in JAK2^{V617F} mice in comparison to WT mice. A gene set enrichment analysis revealed the canonical IL6-JAK-STAT3 pathway and interestingly the receptor ALK1 was found to be the highest upregulated gene in this pathway in Ly6G⁺ neutrophils of the lung as well as PB. This result is very interesting but rather surprising as ALK1 is an endothelial-specific receptor. This unexpected result needs to be supported by further experiments.

- First, the purity of the Ly6G⁺ cells enrichment (98%) should be checked by other means than just May-Giemsa staining. They should test some endothelial cell markers in this purification by PCR and /or IF, to check if they find any vascular enrichment.

Response;

We checked the purity of the lung Ly6G⁺ cell enrichment by immunofluorescence using an anti-Ly6G antibody and an anti-myeloperoxidase (MPO) antibody, which are neutrophil-specific markers. We found that nearly 100% of the MACS-sorted Ly6G⁺ cells were positive for both Ly6G and MPO. These data were consistent with those of May-Giemsa staining. In contrast, the MACS-sorted Ly6G⁺ cells were not stained by an endothelial marker, an anti-CD31 antibody. We have included these data in Supplementary Fig. 18, and described these in the Results and Methods sections.

In the revised text;

The purity of the lung Ly6G⁺ cell enrichment was confirmed by immunofluorescence (Supplementary Fig. 18). *(page 16, lines 264-265)*

The purity of the neutrophils was >98% as determined by May-Giemsa staining and the specificity was confirmed with the positive immunostaining by anti-Ly6G (ab25377, Abcam) and anti-

Myeloperoxidase (ab9535, Abcam) antibodies and with the negative immunostaining by an anti-CD31 antibody (102401, BioLegend). (pages 41-42, lines 694-697)

- Second, they show that *Acvr11* mRNA level in whole lung extract is also increased in JAK2V617F mice under hypoxia (Fig5a). As *Acvr11* expression in the lungs is predominantly expressed by endothelial cells, it is rather surprising to observe such a high increase in *Acvr11* that would be due to neutrophils. ALK1 levels are particularly high in lungs as this tissue contains a very high number of endothelial cells.

It is interesting to note, although not discussed in the text, that *Acvr11* levels in this figure is also increase in wt mice under hypoxia. The authors should isolate endothelial cells and neutrophils under normoxia and hypoxia (the RNAseq analysis was performed under normoxia) from the lungs of these mice and check in which cell type is *Acvr11* increased. It would be interesting to compare the number of *Acvr11* copies in neutrophils versus endothelial cells.

Response;

Firstly, we noted that *Acvr11* mRNA expression levels from whole lung homogenates were increased in WT mice after exposure to chronic hypoxia in the revised manuscript. Secondly, we isolated the neutrophils and endothelial cells from the lung after exposure to normoxia and chronic hypoxia by MACS with an anti-Ly6G antibody and an anti-CD31 antibody, respectively. *Acvr11* mRNA levels in JAK2V617F-Ly6G⁺ cells were significantly increased compared to WT-Ly6G⁺ cells after normoxia, which were consistent with the data from RNA-seq. *Acvr11* mRNA levels in both WT- Ly6G⁺ and JAK2V617F-Ly6G⁺ cells were increased after hypoxic exposure, but the *Acvr11* levels in the hypoxia-exposed JAK2V617F-Ly6G⁺ cells were significantly elevated compared to those in the hypoxia-exposed WT-Ly6G⁺ cells. In contrast, *Acvr11* mRNA levels in CD31⁺ cells were not different among the groups. These data suggest that the differences of *Acvr11* in the lung homogenates among the 4 groups resulted from different expression levels of *Acvr11* in Ly6G⁺ cells, although *Acvr11* expressions were higher in CD31⁺ cells than in Ly6G⁺ cells. We have included these data in Fig. 6a and Supplementary Fig. 19, and described in the Results section.

In the revised text;

In response to chronic hypoxia, *Acvr11* levels were increased in both WT and JAK2^{V617F} lungs, but these levels in JAK2^{V617F} lungs were greater than those in WT lungs. *Acvr11* mRNA levels in sorted Ly6G⁺ neutrophils were significantly elevated in JAK2^{V617F} lungs compared to WT lungs after both normoxia and hypoxia, but not in CD31⁺ endothelial cells, suggesting that the changes in *Acvr11* in the lungs resulted from different expression levels of *Acvr11* in Ly6G⁺ neutrophils although *Acvr11*

expressions were higher in CD31⁺ cells than in Ly6G⁺ cells (Fig. 6a, Supplementary Fig. 19a). (pages 18-19, lines 301-308)

- The authors should comment on the difference between the 10-fold increase in *Acvr11* mRNA levels versus the 2-fold increase in pSmad1/5 levels (Fig5a and b).

Response;

We have commented in the Results section that there was a difference between the 10-fold increase in *Acvr11* mRNA levels versus the 2-fold increase in phosphorylated Smad1/5/8 levels in chronic hypoxia-exposed JAK^{2V617F} lungs. This finding suggests that the relationship of *Acvr11* mRNA expression and the phosphorylation of Smad1/5/8 was not completely linear, and Smad1/5/8 phosphorylation may be regulated by multiple potential pathways.

In the revised text;

There was a difference between the 10-fold increase in *Acvr11* mRNA levels versus the 2-fold increase in phosphorylated Smad1/5/8 levels in chronic hypoxia-exposed JAK^{2V617F} lungs, indicating that the relationship of *Acvr11* mRNA expression and the phosphorylation of Smad1/5/8 was not completely linear, and Smad1/5/8 phosphorylation may be regulated by multiple potential pathways. (page 19, lines 310-314)

- The authors should measure the level of *Acvr1* mRNA encoding the receptor *Alk2* expression, which highly related to ALK1 and phosphorylates Smad1/5/8, and in contrast to ALK1 ubiquitous. A pSmad1/5/8 immunostaining in lungs would be interesting to discriminate neutrophils pSmad1/5/8 from endothelial pSmad1/5/8 staining.

Response;

We measured the level of *Acvr1* mRNA encoding ALK2 from the lung homogenates, showing that *Acvr1* mRNA was significantly increased in response to chronic hypoxia in both WT and JAK^{2V617F} mice, but there were no differences between WT and JAK^{2V617F} lungs after normoxia or chronic hypoxia. *Acvr1* mRNA in the lung Ly6G⁺ cells was rather decreased in WT and JAK^{2V617F} mice after hypoxia, and the changes in *Acvr1* levels were observed in the opposite direction to those seen in *Acvr11*. There were no differences in *Acvr1* mRNA in the lung CD31⁺ cells among the groups. We have included these data in Supplementary Fig. 19 and described in the Results section. We have tried the p-smad1/5/8 staining immunostaining to discriminate neutrophils from endothelial cells in the lung

tissue, but we met the technical issues of non- or weak-staining and non-specific staining even though we tested the several commercially available antibodies.

In the revised text;

Acvr1 mRNA encoding the ALK2 in the lung homogenates was significantly increased after chronic hypoxia in both WT and JAK2^{V617F} mice, but there were no differences between the groups after normoxia or hypoxia. However, *Acvr1* mRNA in Ly6G⁺ cells was rather decreased in WT and JAK2^{V617F} mice after hypoxia, and the changes in *Acvr1* levels were observed in the opposite direction to those seen in *Acvr11* (Supplementary Fig. 19b). (page 19, lines 314-319)

3) To confirm ALK1 upregulation by JAK2V617F, they then use the commercial cell line HCT116 carrying the JAK2V617F mutation. HCT116 is colorectal carcinoma cell line that should not express Acvr11 so the choice of the cell line is rather surprising in particular when they want to study the promoter of this gene which is endothelial specific. Please comment. Again, it will be important to know the number of copies of Acvr11 in these cells as compared to endothelial cells (Fig 6a). In order to check whether there is a significant level of active/functional ALK1 on these cells, the authors could stimulate these cells with BMP9, the high affinity ALK1 ligand.

Response;

We agree with the reviewer that HCT116 cells are a colorectal carcinoma cell line. We aimed to examine the *Acvr11* promoter regulation induced by JAK2V617F and we needed a cell line for this purpose. Commercially available JAK2V617F knock-in cell lines were limited to the epithelial carcinoma cell lines including HCT116 cells and SW48 cells (colorectal adenocarcinoma cell lines). We thought that using JAK2V617F knock-in cell lines has more impact than overexpression by plasmid transfection of JAK2V617F. Moreover, we could demonstrate that the molecular mechanisms of *Acvr11* regulation by JAK2-STAT3 were commonly present when we used the HCT116 cells. As the reviewer suggested, we measured and compared the *Acvr11* mRNA levels in wild-type (JAK2^{+/+}) HCT116 cells as well as the other cell lines such as human pulmonary artery endothelial cells (HPAEC). As the reviewer pointed, *Acvr11* mRNA expression levels in HPAEC were higher than in HCT116 cells, but those in HCT116 cells were higher than human embryonic kidney 293 (HEK293) cells,

suggesting that HCT116 cells indeed express *Acvr11* mRNA as shown below.

Accordingly, we stimulated HCT116 cells by BMP9 and demonstrated that Smad1/5/8 was phosphorylated by stimulation of BMP9 in HCT116 cells similarly to HPAEC (Supplementary Fig. 23). Thus, although HCT116 cells are not endothelial cells, HCT116 cells indeed express *Acvr11* and can be used for the present study. We have included the data in the Results section.

In the revised text;

Smad1/5/8 was phosphorylated by stimulation of BMP9, a high affinity in ALK1 ligand, in HCT116 cells (Supplementary Fig. 23). (pages 20-21, lines 340-342)

4) The in-silico analysis identified several putative STAT3 binding sites in ACVRL1 promoter region in both humans and mice (Fig. 6d), that were not mentioned in the publication cited by the authors and only one STAT3 binding site is present in the putative promoter sequence from -1035 bp to +210 bp of the transcriptional start that the author studied. So, the involvement of STAT3 in the regulation of this promoter is not that strongly supported by in-silico data. Although the experiment with Ruxolitinib, a specific JAK1/2 inhibitor (Fig. 6f), supports a role for the JAK pathway in ACVRL1 promoter activity, no experiment was performed to conclude that it is STAT3 as mentioned in the discussion (lane 361). The authors should just conclude that ALK1 promoter activity in HCT116 cells seems to be JAK1/2 dependent.

Response;

We performed additional experiments to support to our findings that the *Acvr11* promoter activity was regulated by STAT3. The chromatin immunoprecipitation (ChIP) coupled with qPCR showed that the bindings of STAT3 and the putative *Acvr11* promoter regions were significantly increased in

JAK2^{V617F/+} HCT116 cells compared to *JAK2*^{+/+} cells. In addition, the administration of sttatic, an inhibitor of STAT3, attenuated the *ACVRL1* promoter activity in *JAK2*^{V617F/+} HCT116 cells. We have included these data in Fig. 7 and described in the Results section.

In the revised text;

The chromatin immunoprecipitation (ChIP) coupled with qPCR showed that the bindings of STAT3 and the putative *ACVRL1* promoter regions were significantly increased in *JAK2*^{V617F/+} HCT116 cells compared to *JAK2*^{+/+} HCT116 cells (Fig. 7e). (page 21, lines 348-351)

In addition, the administration of Sttatic, an inhibitor of STAT3, attenuated the *ACVRL1* promoter activity (Fig. 7h). (page 21, lines 357-358)

5) Very interestingly, to support their ALK1 hypothesis, the authors performed some in vivo experiments in order to block ALK1 and test whether it would reverse their phenotype. For this, they used the inhibitor K02288. First, it is quite surprising to see no reference concerning this inhibitor in the manuscript and this should be corrected (Santivale et al. 2013, Plos One) and second, they do not mention that this inhibitor is not specific to ALK1 but also inhibits ALK2 and that this inhibitor is most of the time proposed as an ALK2 inhibitor with a slightly higher affinity for ALK2 than ALK1. This is why, it is important to measure ALK2 levels in neutrophils as ALK2 is a more widely expressed receptor. BMP2 levels which plays a much preponderant role than ALK1 in PH should also be measured.

Response;

Thank you for your helpful comments. Firstly, we cited the reference (Santivale et al. 2013, Plos One) as a ref. #30. To date, the inhibitors specific to ALK1 are not commercially available, and such ALK1 inhibitors are likely to cross-react ALK2. As the reviewer suggested, we noted that K02288 was inhibitors to ALK1 as well as ALK2. As we described in your comment #2, ALK1 in the neutrophils was significantly increased in *JAK2*^{V617F} lungs compared to WT lungs in both normoxia and chronic hypoxia (Fig. 6a), but ALK2 was not. Thus, ALK1 may play an important role in the *JAK2*^{V617F} PH pathogenesis, rather than ALK2. Also, we measured the BMP2 mRNA levels in the lung homogenates as well as MACS-isolated Ly6G⁺ neutrophils and CD31⁺ endothelial cells from the lung, showing that there were no differences in BMP2 mRNA between WT and *JAK2*^{V617F} mice in the lung homogenates or each cell type. We have included these data in the Results section and Supplementary Fig. 19c.

In the revised text;

K02288, a chemical inhibitor of ALK1/2 (ref. #30, 31) clearly decreased the phosphorylation levels of Smad1/5/8 in chronic hypoxia-exposed $JAK2^{V617F}$ lungs as well as $JAK2^{V617F/+}$ HCT116 cells. (page 22, lines 365-367)

There were no differences on *Bmpr2* mRNA between WT and $JAK2^{V617F}$ mice in the lung homogenates, Ly6G⁺, or CD31⁺ cells (Supplementary Fig. 19c). (page 19, lines 320-321)

Together, from these experiments it can only be concluded that the ALK1/2 pathway is important for PH development in accordance with a recent paper that they cite (Li et al., Circ Res, 2019).

Response;

Accordingly, we stated that the ALK1/2 pathway is important for PH development in accordance with a recent paper (Li et al., Circ Res, 2019. ref. #45) in the Results section.

In the revised text;

Collectively, these results suggest that the ALK1/2 pathway was involved in chronic hypoxia-induced PH in $JAK2^{V617F}$ mice. (page 23, lines 385-386)

It is however surprising that K02288 administration did not significantly change the levels of RVSP or RV hypertrophy in chronic hypoxia-exposed WT mice as recently published in the above cited paper. These experiments need to be performed using other ALK1 inhibitors and the effect of these inhibitors need to be also tested under normoxia.

Response;

With the reviewer's kind suggestion, we tested another ALK1 inhibitor, LDN-212854, after normoxia and chronic hypoxia. Again, ALK1-specific inhibitors are not commercially available, and therefore we selected this LDN-212854, which is the second most specific chemical inhibitor to ALK1 from ref. #31 and technical product information. We observed that the treatment of LDN-212854 significantly decreased RVSP and RV/LV+S in chronic hypoxia-exposed $JAK2^{V617F}$ mice, similar to K02288. K02288 or LDN-212854 did not affect the levels of RVSP and RV/LV+S in WT or $JAK2^{V617F}$ mice after normoxia. We have described these in the Results section.

In the revised text;

Of note, we found that the treatment of LDN-212854, another ALK1/2 inhibitor (ref. #31), significantly decreased RVSP and RV/LV+S in chronic hypoxia-exposed $JAK2^{V617F}$ mice, similar to K02288 (Supplementary Fig. 27, 28). K02288 or LDN-212854 did not affect the levels of RVSP and RV/LV+S in WT or $JAK2^{V617F}$ mice after normoxia (Supplementary Fig. 29). (page 23, lines 379-383)

Minor points:

1) Lane 257: please indicate in the text what cell line is used.

Response;

We stated the HCT116 cell lines.

In the revised text;

Heterozygous $JAK2^{V617F}$ knock-in ($JAK2^{V617F/+}$) HCT116 cell lines were analyzed. (page 20, lines 339-340)

2) Fig6e, please precise the cell type transfected in the figure legend.

Response;

We precisely indicated the $JAK2^{V617F/+}$ HCT116 cells for transfection in the Figure legend (Figure 7f in the revised manuscript).

In the revised text;

The pGL3-basic vector containing the putative *ACVRL1* promoter region (TSS -875 bp) and pNL1.1.TK [Nluc/TK] as a control vector were co-transfected in $JAK2^{V617F/+}$ HCT116 cells. (Figure legend)

Thank you again for your thoughtful comments to our manuscript.

To Reviewer #3:

Thank you very much for reviewing our manuscript (NCOMMS-20-18526) and providing us with valuable comments. Our responses to your comments are as follows,

General comments:

In this study, Kimishima et al. have investigated the effect of clonal somatic mutation JAK2V617F on PH development. The JAK-STAT3 pathway has since long been known to be involved in PH. JAK2V617F, even though not the most common mutation among patients suffering from myeloproliferative neoplasms, it is associated with vascular complications. The authors have demonstrated that clonal hematopoiesis with JAK2V617F could exacerbated chronic hypoxia-induced PH and pulmonary vascular remodeling in mice. Both JAK2V617F transgenic and wild type recipient mice transplanted with JAK2V617F developed an exacerbated PH upon chronic hypoxia via accumulation neutrophils in perivascular regions of lungs. They show that JAK2V617F-mediated STAT3 phosphorylation could upregulate ALK1-Smad1/5/8 signaling resulting in enhanced PH phenotype. The findings are interesting.

Response;

We appreciate for your constructive comments.

Major comments:

1. A main concern is a lack of data showing how chronic hypoxia can trigger JAK2V617F PH phenotype. Is it HIF regulated activation? Are these neutrophils more likely to invade the perivascular area in the lung?

Response;

We thank the reviewer for suggesting such important points. Accordingly, we performed additional experiments and analyses. Firstly, we investigated the protein levels of HIF1 α in the mouse lung, which is an important molecule for the development of PH in a variety of cell types. HIF1 α expression levels in the lung were increased in both WT and JAK2^{V617F} mice after chronic hypoxia, but there was no difference between the groups (Supplementary Fig. 20). We showed that STAT3 phosphorylation levels were significantly increased in JAK2^{V617F} lungs after chronic hypoxia compared to normoxia (Fig. 6). To confirm the association of STAT3 with HIF1 α , immunoprecipitation analysis showed that STAT3 protein weakly interacted with HIF1 α in JAK2^{V617F} lungs at normoxia, and chronic hypoxia for 2 weeks increased the bindings of STAT3 and HIF1 α (Supplemental Fig. 21). JAK2V617F PH

phenotype was characterized by pulmonary arterial remodeling, which was associated with smooth muscle cell proliferation (in relation to your comment #4) and thickening of medial walls of pulmonary arteries. The conditioned medium from cultured hypoxia-exposed Ly6G⁺ neutrophils of JAK2^{V617F} mice induced the smooth muscle cell proliferation (Supplemental Fig. 22). The increased proliferation was partly but significantly attenuated by the conditioned medium from those pretreated with an HIF1 α inhibitor, Echinomycin. It has been reported that STAT3 is involved in the HIF1 α -mediated hypoxic transcriptional response (Oncogene 2014; 33: 1670–1679, ref. #39). Thus, HIF1 α may be associated with the hypoxia-related activation in the JAK2V617F neutrophils potentially via its STAT3 interactions. However, we need to clarify further mechanisms of how chronic hypoxia could trigger JAK2V617F PH in the future.

When we categorized the infiltrated Ly6G⁺ neutrophils in perivascular and non-perivascular regions in JAK2^{V617F} lungs, the numbers of JAK2V617F Ly6G⁺ neutrophils were increased in both perivascular and non-perivascular regions after chronic hypoxia (Supplementary Fig. 7, 14). The proportion was not different between normoxia and hypoxia exposure. Thus, the chronic hypoxia-induced neutrophil invasions were not limited to perivascular regions in JAK2^{V617F} lungs. We have included these data and described these in the Results section.

In the revised text;

HIF1 α expression levels in the lung were increased in both WT and JAK2^{V617F} mice after chronic hypoxia, but there was no difference between the groups (Supplementary Fig. 20). Immunoprecipitation analysis showed that STAT3 protein weakly interacted with HIF1 α in JAK2^{V617F} lungs at normoxia, and chronic hypoxia increased the bindings (Supplementary Fig. 21). The conditioned medium from hypoxia-exposed JAK2^{V617F} neutrophils pretreated with a HIF1 α inhibitor partly attenuated the increases in the proliferation of pulmonary arterial smooth muscle cells (Supplementary Fig. 22). *(page 20, lines 325-331)*

Increased physical interactions of HIF1 α and STAT3 (ref. #39) in response to hypoxia might trigger PH phenotypes in JAK2^{V617F} mice, but further mechanisms need to be clarified. *(page 28, lines 467-469)*

Of note, the numbers of Ly6G⁺ cells in perivascular regions, as well as non-perivascular regions, of JAK2^{V617F} lungs were further increased after chronic hypoxia exposure compared to those after normoxia exposure (Supplementary Fig. 7). *(page 9, lines 138-141)*

the numbers of Ly6G⁺ cells in both perivascular and non-perivascular regions in chronic hypoxia-exposed JAK2^{V617F}-BMT mice were further increased compared to normoxia-exposed JAK2^{V617F}-

BMT mice (Fig. 2g, h, Supplementary Fig. 13, 14). (page 12, lines 188-191)

2. The authors showed that approximately 40% of the perivascular GFP positive cells are Ly6G positive (neutrophils). However, it remains unclear what are the remaining GFP positive cells. In order to not underestimate the contribution of other BM-derived cell types, I would suggest to use either IF staining for other markers such as CD68 or extensive flow cytometry to identify these cells.

Response;

In line with the reviewer's suggestion, we additionally stained the GFP-positive cells using an anti-F4/80 antibody (instead of CD68) or an anti-CD45R antibody in the lungs to clarify the contribution of other BM-derived cell types. In JAK2^{V617F}-GFP-BMT mice, Ly6G⁺ cells predominantly contributed to BM-derived cells rather than other cell types of F4/80⁺ or CD45R⁺ cells. We have included these data and described in the Results section.

In the revised text;

In JAK2^{V617F}-GFP-BMT mice, nearly half of the GFP⁺ cells expressed Ly6G in pulmonary arterial regions and Ly6G⁺ cells predominantly contributed to BM-derived cells rather than F4/80⁺ or CD45R⁺ cells (Fig. 3c, d, Supplementary Fig. 15). (pages 13-14, lines 215-217)

Furthermore, CD68 IHC staining shown in Supplemental Figures 2 and 3 is not convincing. Double immune staining including CD45 and presenting different cell types of interest as a percentage of CD45 positive perivascular cells is recommended.

Response;

We apologize that our original figures might give you confusion. Accordingly, we carried out IHC staining using another macrophage marker of F4/80, instead of CD68. Additionally, we performed the double immune staining including CD45 and presenting different cell types of interest as a percentage of CD45-positive perivascular cells. We have replaced the original images with new images in Supplementary Fig. 6 and 13, and described these in the Results section.

In the revised text;

Ly6G-expressing cells within CD45⁺ cells were dominantly found rather than F4/80⁺ macrophages or CD45R⁺ B cells (Supplementary Fig. 6). (page 9, lines 137-138)

Ly6G⁺ cells significantly contributed to CD45⁺ cells rather than F4/80⁺ or CD45R⁺ cells in hypoxia-exposed JAK2^{V617F}-BMT lungs (Supplementary Fig. 13). (page 12, lines 191-193)

3. A more detailed characterization of pulmonary hemodynamics and cardiac function would be helpful. It is not clear if cardiac function is affected and whether changes in cardiac output contribute to the changes observed for RVSP. Also, it would be interesting to see whether rheological effects of leukocytes and thrombocytes affects PH.

Response;

We evaluated the pulmonary hemodynamics and cardiac function by echocardiography (Circ Cardiovasc Imaging 2010; 3: 157-163. Circulation 2018; 138: 600-623, ref. #58). The ratio of pulmonary arterial acceleration time to pulmonary arterial ejection time, which inversely correlates with pulmonary arterial resistance, were significantly decreased in both WT mice and JAK2^{V617F} mice after exposure to chronic hypoxia, but that in hypoxia-exposed JAK2^{V617F} mice was significantly decreased compared to that in hypoxia-exposed WT mice. Right ventricular fractional change, which indicates right ventricular systolic function, was significantly lower in JAK2^{V617F} mice than in WT mice after chronic hypoxia. Accordingly, cardiac output was significantly decreased in JAK2^{V617F} mice compared to WT mice after chronic hypoxia. In contrast, left ventricular fractional shortening was not different between WT mice and JAK2^{V617F} mice, suggesting that left ventricular systolic function was not affected by the PH phenotypes in JAK2^{V617F} mice. We found that the recipient mice transplanted with JAK2^{V617F} BM cells showed similar parameters to JAK2^{V617F} mice. We have included these data in Supplementary Fig. 2 and 10, and described in the Results section.

In the present study, the recipient mice transplanted with JAK2^{V617F} BM cells did not show increases in leukocyte or platelet counts, but developed PH in response to chronic hypoxia, suggesting that JAK2 activation in the myeloid cells was associated with PH independently of the leukocyte or platelet counts. The rheological effects of leukocytes and thrombocytes on PH need to be clarified in the future. We described this point in the Discussion section.

In the revised text;

we found that RVSP was significantly elevated in JAK2^{V617F} mice compared to WT mice in response to continuous hypoxia (Fig. 1c) in line with echocardiographic indices for evaluation of pulmonary hemodynamics (Supplementary Fig. 2). (page 8, lines 112-115)

LV fractional shortening or LV+S values was not different among the groups, suggesting that chronic

hypoxia was not associated with LV systolic dysfunction or LV hypertrophy in JAK2^{V617F} mice (Supplementary Fig. 2, 3). (page 8, lines 117-120)

JAK2^{V617F}-BMT mice showed significant increases in both RVSP and RV/LV+S compared to WT-BMT mice in response to exposure to chronic hypoxia for 3 weeks (Fig. 2d), consistent with data from echocardiography to assess pulmonary hemodynamics (Supplementary Fig. 10). (page 11, lines 175-178)

In line with our JAK2^{V617F}-BMT model that did not show elevation of white blood cells or platelets, the activation of the myeloid cells including JAK-STAT may lead to PH phenotypes even without elevation of leukocyte or platelet counts. The rheological effects of leukocytes and thrombocytes on PH need to be clarified. (pages 26-27, lines 441-445)

4. In the experimental mouse model of chronic hypoxia-induced PH, it has been shown that the number of proliferating cells increased in the pulmonary vasculature. These proliferating cells are usually found in the vessel wall and could be characterized by PCNA positive nuclei. The IHC staining for PCNA in Supplemental Figures 2 and 3 was not successful and it could not be quantified (cytoplasm and even blood seems to be stained). I suggest repeating the staining or using another proliferation marker. Accordingly, the authors should change the statements through the manuscript such as line 130-133: Proliferating cell nuclear antigen (PCNA)-positive cells in peri-vascular regions were comparable between WT lungs and JAK2^{V617F} lungs, suggesting that the accumulated Ly6G+ cells were not on the proliferative state (Supplementary Fig. 2b).

Response;

Thank you for the reviewer's helpful suggestion. To characterize the proliferation of smooth muscle cells in pulmonary arteries, we performed IHC staining with an anti-Ki67 antibody, which is another proliferation marker, in addition to PCNA. Double immunostaining was performed using an anti-Ki67 antibody and an anti- α SMA antibody to identify the smooth muscle cell proliferation of pulmonary arteries. In line with the observation of pulmonary arterial remodeling in JAK2^{V617F} mice, the numbers of Ki67-positive nuclei within α SMA-positive cells were significantly increased in JAK2^{V617F} mice and JAK2^{V617F}-BMT mice compared to WT mice or WT-BMT mice, respectively. We have replaced the original images with new ones in Supplementary Fig. 5 and 12. Furthermore, an ALK1 inhibitor significantly decreased the numbers of Ki67-positive nuclei within α SMA-positive cells of pulmonary arteries in chronic hypoxia-exposed JAK2^{V617F} mice, and we have included these data in

Supplementary Fig. 26. Accordingly, we have omitted the original sentences as you advised, and have described these findings in the Results section.

In the revised text;

The numbers of proliferating smooth muscle cells in the pulmonary arteries were significantly increased in JAK2^{V617F} mice compared to WT mice after chronic hypoxia (Supplementary Fig. 5a). *(page 9, lines 127-129)*

Likewise, medial wall thickness, the percentage of muscularized vessels and the numbers of proliferating smooth muscle cells of pulmonary arteries were significantly increased in JAK2^{V617F}-BMT mice compared to WT-BMT mice after hypoxia exposure (Fig. 2e, f, Supplementary Fig. 12a). *(pages 11-12, lines 179-182)*

There were significant decreases in medial wall thickness and muscularization and the numbers of proliferating smooth muscle cells in pulmonary arteries of K02288-treated JAK2^{V617F} mice compared to DMSO-treated JAK2^{V617F} mice (Fig. 8d, e, Supplementary Fig. 26). *(page 22, lines 373-376)*

5. It is not clear why the authors used only female mice.

Response;

Considering the clinical relevance of PH patients, women are 1.8-3.6 times more likely to be affected than men (Chest 2015; 148: 1043-1054, Chest 2011; 1390: 128-137). Thus, we first examined female JAK2^{V617F} mice and used the female mice in a whole series of the present study. Nevertheless, we investigated male JAK2^{V617F} mice, and we found that male JAK2^{V617F} mice also showed significant increases in RVSP and RV/LV+S compared to male WT mice 2 weeks after chronic hypoxia. We have included these data in the Results section and Supplementary Fig. 4.

In the revised text;

Of note, we found that even male JAK2^{V617F} mice showed significant increases in RVSP and RV/LV+S compared to male WT mice 2 weeks after chronic hypoxia (Supplementary Fig. 4). Considering the clinical relevance of PH patients that women are more likely to be affected than men (ref. # 23), we thereafter used female mice in a whole series of the present study otherwise indicated. *(page 8, lines 120-124)*

Is there any particular reason why different time points of hypoxic exposure (2 and 3 weeks respectively) were used in different parts of this study?

Response;

Starting from 2 weeks after chronic hypoxia exposure, we observed the noticeable signs of cardio-respiratory distress such as reduced activity, diminished appetite, and piloerection in $JAK2^{V617F}$ mice, but not in WT mice. In this case, a humane endpoint had to be applied in accordance with our Animal Research Committee, and we needed to terminate the sick mice immediately when we realized such findings. When we analyzed mice at 2 weeks after chronic hypoxia, we found that $JAK2^{V617F}$ mice displayed significant increases in RVSP and RV hypertrophy compared to WT mice. Also, to minimize the secondary alternation for investigation of the molecular mechanisms that cause PH, we determined to analyze the mice at 2-week point. In contrast, the recipient mice transplanted with $JAK2^{V617F}$ bone marrow cells (JAK^{V617F} -BMT mice) did not show such signs of cardio-respiratory distress 2 weeks after chronic hypoxia, but the levels of PH were not sufficiently different in comparison to WT-BMT mice. Thus, we analyzed the BMT mice at 3-week point, showing that JAK^{V617F} -BMT mice displayed significant increases in RVSP and RV hypertrophy. The severity and development of PH phenotypes between $JAK2^{V617F}$ mice and JAK^{V617F} -BMT mice potentially due to the presence or absence of hematological phenotypes. We have described this point in the Results section.

In the revised text;

Starting from 2 weeks after chronic hypoxia exposure, we observed the noticeable signs of cardio-respiratory distress such as reduced activity, diminished appetite, and piloerection in $JAK2^{V617F}$ mice, but not in WT mice. We determined to analyze the mice at 2-week point to minimize the secondary alternation for investigation of the molecular mechanisms that cause PH. (page 7, lines 101-105)

The myeloproliferative neoplasms (MPNs) are diseases occurring in later ages (Srour, PMID: 28387461). Therefore, it would be worth to characterise $JAK2^{V617F}$ transgenic mice in later age to identify if somatic mutations in JAK could drive PH development alone, without hypoxia stimulus.

Response;

We examined the levels of PH in $JAK2^{V617F}$ mice in later age without hypoxia stimulus. We found that the mean of RVSP and RV hypertrophy was slightly increased in $JAK2^{V617F}$ mice at the age of 8- to 9-month-old, and 2 mice (16.7%) out of 12 $JAK2^{V617F}$ mice in later age showed comparatively high RVSP and RV/LV+S (Supplementary Fig. 9). However, there was no statistical significance on RVSP

and RV/LV+S between the aged WT and JAK2^{V617F} mice without hypoxia stimulus. These findings may be consistent with the human MPN patients, all of whom do not develop PH (ref. #6). An additional genetic and/or environmental hit in addition to JAK2V617F is needed for the onset and development of PH in the predisposed subjects. We have included these data in Supplementary Fig. 9 and described in the Results and Discussion sections.

In the revised text;

There was no statistical significance on RVSP and RV/LV+S between aged WT and JAK2^{V617F} mice at 8- to 9-month-old without hypoxia stimulus, but some of the aged JAK2^{V617F} mice displayed comparatively high RVSP and RV/LV+S (Supplementary Fig. 9). *(page 10, lines 152-155)*

JAK2^{V617F} mice developed PH pathology in response to chronic hypoxia but did not develop PH in normoxia, indicating that JAK2V617F alone is not sufficient to induce PH and that a trigger such as chronic hypoxia is required for PH phenotypes in JAK2^{V617F} mice. In contrast, patients with MPNs can develop PH in the setting of normoxia. However, not all MPN patients develop PH. As MPN occurs in later ages (ref. #36), an additional genetic and/or environmental hit in addition to JAK2V617F may be needed for the onset and development of PH in the predisposed subjects. *(page 27, lines 445-451)*

6. The authors should provide data from normoxic experiments in figure 7.

Response;

Accordingly, we have included data from normoxic groups in Supplementary Fig. 29 and described the Result section.

In the revised text;

K02288 or LDN-212854 did not affect the levels of RVSP and RV/LV+S in WT or JAK2^{V617F} mice after normoxia (Supplementary Fig. 29). *(page 23, lines 382-383)*

7. As nicely pointed by authors, the role of ALK1 in PH development is complex. Recent publications of Tu et al. (PMID: 30636542) showed that ALK1 ligand trap counterintuitively inhibited PH. Could the authors explain why inhibition of ALK1 by K02288 attenuated chronic hypoxia-induced PH only in JAK2V617F mice and did not affect WT mice? How was the dose of K02288 chosen for? Is it possible that in high dose K02288 could also affect EC or PSMC

proliferation? Could the authors prove that K02288 specifically affect neutrophils and did not affect directly EC/SMC?

Response;

We speculate that the effects of K02288 might specifically modify the upregulated ALK1 by JAK2^{V617F} in neutrophils. Because there has been no adequate literature using K02288 in mice, we referred to the previous reports using the other ALK inhibitors (Nat Med 2008; 14: 1363–1369, Cancer Res 2011;71:5194-5203) in mice. We estimated the reasonable dosage of K02288 to treat the mice comparing the chemical activities among the ALK inhibitors. We preliminarily tested several doses of K02288 and found that the dose of K02288 (12 mg/kg) we used in the present study could attenuate RVSP and RV hypertrophy in chronic hypoxia-exposed JAK2^{V617F} mice. According to the reviewer's suggestion, we performed additional experiments on whether a higher dose of K02288 could affect PH even in WT mice. We also set JAK2^{V617F} mice for this experiment. The high dose of K02288 (24 mg/kg) indeed reduced RVSP and RV hypertrophy in chronic hypoxia-exposed JAK2^{V617F} mice (as shown below) but did not decrease RVSP and RV hypertrophy in chronic hypoxia-exposed WT mice. Histological analysis showed that the chronic hypoxia-induced SMC proliferation in pulmonary arteries was not decreased in a high dose of K02288-treated WT mice. The contribution of EC proliferation was unlikely in our chronic hypoxia model by Ki67-immunostaining. Pulmonary arterial muscularization slightly decreased in a high dose of K02288-treated WT mice. It is possible that a high dose of K02288 could directly affect the histological remodeling, and a much higher dose of K02288 may attenuate the PH levels in hypoxia-exposed WT mice, but it was not evident that a reasonable dose of K02288 did not contribute to the improvement of PH in chronic hypoxia-exposed WT mice. We believe that K02288 even with a lower dose had a higher affinity to JAK2^{V617F}-related

upregulation of ALK1 in neutrophils, resulting in the prevention of PH phenotypes. We have included these data in Supplementary Fig. 30 and described in the Results section.

In the revised text;

A higher dose of K02288 did not attenuate PH levels in hypoxia-exposed WT mice (Supplementary Fig. 30). (*page 23, lines 383-384*)

8. Please clarify how pulmonary arteries were distinguished from veins for the quantification of pulmonary vascular remodelling.

Response;

This is anatomically distinguishable. Pulmonary arteries are distributed along the bronchi and show an eccentric morphology. Wall of pulmonary arteries is thick and elastic while that of pulmonary veins is comparatively thinner. We have described these in the Methods section.

In the revised text;

Based on the anatomical characteristics, pulmonary arteries, distributed along the bronchi and displayed an eccentric morphology with thick and elastic walls, are distinguishable from pulmonary veins. (*page 37, lines 610-612*)

Furthermore, what was the selection criteria for characterization of non-, partially- or fully-muscularized?

Response;

Based on α SMA-immunostaining, the pulmonary arteries with a diameter of less than 50 μ m were classified into three groups; the artery with α SMA-positive throughout the entire circumference of the vessel cross-section was defined as “fully” muscularized, and the artery with α SMA-positive of 5 to 99% around the vessel was defined as “partially” muscularized, and the artery with α SMA-positive of less than 5% around the vessel was classified as “non” muscularized (Circulation 2018; 138: 600-623, ref. #58). We have described these in the Methods section.

In the revised text;

the pulmonary arteries with a diameter of less than 50 μ m were classified into three groups; the artery with α SMA-positive throughout the entire circumference of the vessel cross-section was defined as

“fully” muscularized, and the artery with α SMA-positive of 5 to 100% around the vessel was defined as “partially” muscularized, and the artery with α SMA-positive of less than 5% around the vessel was classified as “non” muscularized (ref. #58). (pages 36, lines 605-610)

9. Since GFP positive cells are stained using GFP antibody, it would be useful to include WT animals without GFP as a negative control.

Response;

In the present study, we used an anti-GFP antibody to enhance the immuno-specificities and to increase the fluorescent signals. According to the reviewer’s suggestion, we have included the images stained by an anti-GFP antibody in the lung sections from WT recipient mice transplanted with WT BM cells without GFP as a negative control in Supplementary Fig.15c.

Minor comments:

1. Please provide LV+S values as a change in LV+S weight affects RV/(LV+S) and can thus suggest a false positive or negative effect on RV hypertrophy

Response;

We have provided these data in Supplementary Fig. 3, 11, 25, and described in the Results section.

In the revised text;

LV fractional shortening or LV+S values was not different among the groups, suggesting that chronic hypoxia was not associated with LV systolic dysfunction or LV hypertrophy in JAK2^{V617F} mice (Supplementary Fig. 2, 3). (page 8, lines 117-120)

LV+S values were not different among the groups (Supplementary Fig. 11). (page 11, lines 178-179)

2. Please unify the presentations of statistics through the figures.

Response;

We have unified the presentations of statistics through the Figure legends.

Thank you again for your thoughtful comments to our manuscript.

REVIEWER COMMENTS

Reviewer #1 (Remarks to the Author):

Greatly improved by the additional experiments, including the dissection of the impact of varying fractions of JAK2 mutant HSPCs transplanted, and the analysis of mutant vs WT neutrophils from the pulmonary vasculature.

Remaining issues are continued poor English syntax and usage, lack of reasonable paragraph structure (some go on for pages), and some redundancy with certain points being made again and again in both results and discussion.

Reviewer #2 (Remarks to the Author):

The present study claims that clonal haematopoiesis with JAK2V617F mutation accelerates pulmonary hypertension (PH) in mice via neutrophils accumulation in pulmonary arterial regions through the receptor ALK1. Although the PH phenotype of these mice seems clear, the mechanism of action presented by the authors involving ALK1 is far from being demonstrated and requires further experiments to be validated. The manuscript addresses a very important question for the scientific community as pulmonary hypertension is a life-threatening disease without cure and a better understanding of its underlying mechanisms is an essential point for cardiovascular community. Here the authors address the link of myeloproliferative neoplasms (MPNs) and PH. My major concern was the validation of the involvement of ALK1 versus another ALK receptor in this process and the answers obtained and the new experiments performed still don't convince me. See below my concerns, in blue, related to their responses.

Major points

1) The authors wanted to know the involvement of JAK-STAT pathway in PH development and saw that STAT3 phosphorylation was significantly increased in PH treated mice. To clarify this effect, they tested their JAK2V617F mice, whose mutation is the most frequent mutation among MPNs and that causes JAK-STAT activation, in a model of PH which chronic hypoxia (10%O₂). Strikingly, they found that JAK2V617F exacerbated PH in mice and that JAK2V617F-expressing neutrophils were specifically accumulated in pulmonary arterial regions. This is a very interesting result that would be strengthened if confirmed in another PH model such as monocrotaline injection.

The authors have tried to use monocrotaline in order to have another PH model but they failed to induce PH as they did not use monocrotaline-pyrrole that would have allowed to bypass the problem of monocrotaline metabolization. They thus used the mouse Sugen-hypoxia model which is not a very well accepted model in the field of PH still with this model they could confirm that JAK2V617 showed a more pronounced PH than WT mice.

2) To elucidate the underlying mechanisms, gene expression profiling of neutrophils, at several stages of differentiation, was performed by RNA sequencing in sorted Ly6G⁺ cells from BM, peripheral blood (PB) and lungs in JAK2V617F mice in comparison to WT mice. A gene set enrichment analysis revealed the canonical IL6-JAK-STAT3 pathway and interestingly the receptor ALK1 was found to be the highest upregulated gene in this pathway in Ly6G⁺ neutrophils of the lung as well as PB. This result is very interesting but rather surprising as ALK1 is an endothelial-specific receptor. This unexpected result needs to be supported by further experiments.

- First, the purity of the Ly6G⁺ cells enrichment (98%) should be checked by other means than just May-Giemsa staining. They should test some endothelial cell markers in this purification by PCR and /or IF, to check if they find any vascular enrichment.

OK

- Second, they show that *Acvr1* mRNA level in whole lung extract is also increased in JAK2V617F mice under hypoxia (Fig5a). As *Acvr1* expression in the lungs is predominantly expressed by endothelial cells, it is rather surprising to observe such a high increase in *Acvr1* that would be due to neutrophils. ALK1 levels are particularly high in lungs as this tissue contains a very high number of endothelial cells.

It is interesting to note, although not discussed in the text, that *Acvr1* levels in this figure is also increase in wt mice under hypoxia. The authors should isolate endothelial cells and neutrophils under normoxia and hypoxia (the RNAseq analysis was performed under normoxia) from the lungs of these mice and check in which cell type is *Acvr1* increased. It would be interesting to compare the number of *Acvr1* copies in neutrophils versus endothelial cells.

OK

- The authors should comment on the difference between the 10-fold increase in *Acvr1* mRNA levels versus the 2-fold increase in pSmad1/5 levels (Fig5a and b).

- The authors should measure the level of *Acvr1* mRNA encoding the receptor *Alk2* expression, which highly related to ALK1 and phosphorylates Smad1/5/8, and in contrast to ALK1 ubiquitous. A pSmad1/5/8 immunostaining in lungs would be interesting to discriminate neutrophils pSmad1/5/8 from endothelial pSmad1/5/8 staining.

OK

3) To confirm ALK1 upregulation by JAK2V617F, they then use the commercial cell line HCT116 carrying the JAK2V617F mutation. HCT116 is colorectal carcinoma cell line that should not express *Acvr1* so the choice of the cell line is rather surprising in particular when they want to study the promoter of this gene which is endothelial specific. Please comment. Again, it will be important to

know the number of copies of *Acvrl1* in these cells as compared to endothelial cells (Fig 6a). In order to check whether there is a significant level of active/functional ALK1 on these cells, the authors could stimulate these cells with BMP9, the high affinity ALK1 ligand.

The authors have measured *Acvrl1* levels in HCT116 versus HPAEC and indeed found a 100-fold difference between in favour of endothelial cells supporting that the *Acvrl1* levels is low in these cells. They have stimulated the cells with BMP9 and found an increase in Smad1/5 phosphorylation but the concentrations of BMP9 used (50 and 100 ng/mL) are huge when compared to the EC50 of BMP9 for ALK1 (50 pg/mL) and thus it cannot be concluded from this result that HCT116 cells respond to BMP9 via ALK1 as presented in their response but would support a role for ALK2. Lower doses of BMP9 should have been tested. SiRNA against ALK1 would have been the right approach to demonstrate that these cells express an active ALK1 receptor. From these results it cannot be excluded that what is observed could be due to ALK2. I am also septic about the ALK1 expression in HCT116 cells detected by western blot using an antibody from Abcam. First this antibody is not anymore commercially available may be due to specificity problems and second the detection of ALK1 by Western blot is already very challenging in endothelial cells so it must really be difficult to detect in cells like HCT116 that express much less ALK1.

4) The in-silico analysis identified several putative STAT3 binding sites in *ACVRL1* promoter region in both humans and mice (Fig. 6d), that were not mentioned in the publication cited by the authors and only one STAT3 binding site is present in the putative promoter sequence from -1035 bp to +210 bp of the transcriptional start that the author studied. So, the involvement of STAT3 in the regulation of this promoter is not that strongly supported by in-silico data. Although the experiment with Ruxolitinib, a specific JAK1/2 inhibitor (Fig. 6f), supports a role for the JAK pathway in *ACVRL1* promoter activity, no experiment was performed to conclude that it is STAT3 as mentioned in the discussion (lane 361). The authors should just conclude that ALK1 promoter activity in HCT116 cells seems to be JAK1/2 dependent.

OK. The addition of the specific STAT3 inhibitor (stattic) comforts their hypothesis.

5) Very interestingly, to support their ALK1 hypothesis, the authors performed some in vivo experiments in order to block ALK1 and test whether it would reverse their phenotype. For this, they used the inhibitor K02288. First, it is quite surprising to see no reference concerning this inhibitor in the manuscript and this should be corrected (Santivale et al. 2013, Plos One) and second, they do not mention that this inhibitor is not specific to ALK1 but also inhibits ALK2 and that this inhibitor is most of the time proposed as an ALK2 inhibitor with a slightly higher affinity for ALK2 than ALK1. This is why, it is important to measure ALK2 levels in neutrophils as ALK2 is a more widely expressed receptor. *BMP2* levels which plays a much preponderant role than ALK1 in PH should also be measured. Together, from these experiments it can only be concluded that the ALK1/2 pathway is important for PH development in accordance with a recent paper that they cite (Li et al., Circ Res, 2019). It is however surprising that K02288 administration did not significantly change the levels of RVSP or RV hypertrophy in chronic hypoxia-exposed WT mice as recently published in the above cited paper. These experiments need to be performed using other ALK1 inhibitors and the effect of these inhibitors need to be also tested under normoxia.

I appreciate the modification made in the text describing K02288 as an inhibitor of both ALK1 and ALK2 and adding the reference asked. However, they are still statements that do not specify this

point, for example: Statements like “In the current study, inhibition of ALK1 prevented chronic hypoxia-induced PH as well as pulmonary arterial remodeling in JAK2V617F mice. (p29, line 484) and Thus, inhibition of ALK1, which was directly upregulated by JAK-STAT activation, may be effective especially in the lung neutrophils. (p32, line 537).

They have also used another drug LDN-212854 but unfortunately as K02288 this drug inhibits both ALK1 and ALK2 but with a higher selectivity for ALK2 versus ALK1 (ref 31). This inhibitor gave the same result as K02288 (Supplementary Fig. 29) supporting their previous data. However, this inhibitor is not better than K02288 and does not allow to better conclude on the role of ALK1 versus ALK2.

Together, although the authors have replied to many of my comments and have performed supplementary experiments, I still remain unsatisfied concerning their conclusion of a specific role of ALK1 in this work. Apart from the fact that *Acvrl1* expression is upregulated in neutrophils in JAK2V617 mice and that *Stat3* might regulate *Acvrl1* expression, I do not see a clear demonstration that clonal hematopoiesis with JAK2V617 causally leads to PH development through ALK1 in this manuscript. I was also not convinced by their response to reviewer 1 concerning their response to point 5: “Although it was reported that ACVRL1 mutations in hereditary hemorrhagic telangiectasia led to a loss of function (ref. #42, 43), the functional relevance of congenital ACVRL1 mutations in PH remains undetermined whether the mutations result in a gain of function or a loss of function. » This hypothesis is not convincing as ACVRL1 mutations found in PAH are the same mutations described in HHT which result in loss of function. I would thus be very surprised that in PAH these mutations would not lead to a loss of function.

Minor points:

1) Lane 257: please indicate in the text what cell line is used.

OK

2) Fig6e, please precise the cell type transfected in the figure legend.

OK

Reviewer #3 (Remarks to the Author):

The authors have thoroughly revised the manuscript and addressed almost all concerns except my previous question no. 8: "Please clarify how pulmonary arteries were distinguished from veins for the quantification of pulmonary vascular remodelling."

The anatomical difference between arteries and veins is not always clearly visible. This is especially the case with small vessels (diameter $<50\mu\text{m}$) that have been analyzed. Therefore, I would ask to refer in text to "pulmonary vessels" rather than to "pulmonary arteries".

To Reviewer #1:

Thank you very much for reviewing our manuscript (NCOMMS-20-18526A) and providing us with valuable comments. Our responses to your comments are as follows,

Comments;

Greatly improved by the additional experiments, including the dissection of the impact of varying fractions of JAK2 mutant HSPCs transplanted, and the analysis of mutant vs WT neutrophils from the pulmonary vasculature.

Response;

Thank you very much for your positive comments.

Comments;

Remaining issues are continued poor English syntax and usage, lack of reasonable paragraph structure (some go on for pages), and some redundancy with certain points being made again and again in both results and discussion.

Response;

The manuscript was proofread by two native English-speaking scientific editors again. We generated reasonable paragraphs and omitted the redundancy with certain points in the discussion section overlapped with the results section.

To Reviewer #2:

Thank you very much for reviewing our manuscript (NCOMMS-20-18526A) and providing us with valuable comments. Our responses to your comments are as follows,

Comments;

The present study claims that clonal haematopoiesis with JAK2V617F mutation accelerates pulmonary hypertension (PH) in mice via neutrophils accumulation in pulmonary arterial regions through the receptor ALK1. Although the PH phenotype of these mice seems clear, the mechanism of action presented by the authors involving ALK1 is far from being demonstrated and requires further experiments to be validated. The manuscript addresses a very important question for the scientific community as pulmonary hypertension is a life-threatening disease without cure and a better understanding of its underlying mechanisms is an essential point for cardiovascular community. Here the authors address the link of myeloproliferative neoplasms (MPNs) and PH. My major concern was the validation of the involvement of ALK1 versus another ALK receptor in this process and the answers obtained and the new experiments performed still don't convince me. See below my concerns, in blue, related to their responses.

Response;

Thank you again for your important suggestions. Given the lack of specific ALK1 inhibitors that can confirm the proposed *in vivo* mechanism, we have toned down throughout the manuscript the conclusion that the PH phenotype is specifically mediated by ALK1 signaling. Accordingly, the article title was changed to "Clonal hematopoiesis with JAK2V617F promotes pulmonary hypertension with ALK1 upregulation in lung neutrophils". Moreover, we have performed additional *in vitro* experiments in line with the reviewer's specific comments.

Major points

Comments;

1) The authors wanted to know the involvement of JAK-STAT pathway in PH development and saw that STAT3 phosphorylation was significantly increased in PH treated mice. To clarify this effect, they tested their JAK2V617F mice, whose mutation is the most frequent mutation among MPNs and that causes JAK-STAT activation, in a model of PH which chronic hypoxia (10%O₂). Strikingly, they found that JAK2V617F exacerbated PH in mice

and that JAK2V617F-expressing neutrophils were specifically accumulated in pulmonary arterial regions. This is a very interesting result that would be strengthened if confirmed in another PH model such as monocrotaline injection.

>The authors have tried to use monocrotaline in order to have another PH model but they failed to induce PH as they did not used monocrotaline-pyrrole that would have allowed to bypass the problem of monocrotaline metabolization. They thus used the mouse Sugen-hypoxia model which is not a very well accepted model in the field of PH still with this model they could confirm that JAK2V617 showed a more pronounced PH than WT mice.

Response;

We totally agree with the reviewer that the monocrotaline-pyrrole is an alternative to sufficiently induce PH in mice to bypass the problems of monocrotaline. Although the Sugen-hypoxia model we used may not be a very well accepted model in the field of PH, recently, this model is widely used to study PH in mice (Am J Respir Crit Care Med 2011; 184: 1171-1182, ref. #25, J Clin Invest 2015; 125: 1228-1242, Circ Cardiovasc Genet 2017; 10: e001591, Nat Commun 2019; 10: 4143, Nat Commun 2019; 10: 5183, Circ Res 2019; 124: 52-65, Arterioscler Thromb Vasc Biol 2019; 39: 2505-2519, J Am Coll Cardiol 2019; 73: 2567-2580, Cardiovasc Res 2020; 116: 1500-1513) and rats (Circulation 2014; 130: 168-179, Am J Respir Crit Care Med 2015; 191: 1273-1286, Circulation 2017; 135: 1532-1546, Am J Respir Crit Care Med 2018; 197: 373-385, Am J Respir Crit Care Med 2019; 200: 617-627, Basic Res Cardiol 2020; 115: 68, Sci Transl Med 2021; 13: eaba6480). Using this Sugen-hypoxia model, we revealed that JAK2^{V617F} mice showed more exaggerated PH than WT mice. As the reviewer kindly suggested, we think that it is important to confirm further the role of JAK2V617F on PH using other PH models such as monocrotaline-pyrrole injection. We have discussed this point as future work in the Discussion section.

In the texts;

investigation by other PH models such as the monocrotaline-pyrrole need to be clarified
(page 21, line 470)

Comments;

2) To elucidate the underlying mechanisms, gene expression profiling of neutrophils, at several stages of differentiation, was performed by RNA sequencing in sorted Ly6G+ cells from BM, peripheral blood (PB) and lungs in JAK2V617F mice in comparison to WT mice. A gene set enrichment analysis revealed the canonical IL6-JAK-STAT3 pathway and interestingly the receptor ALK1 was found to be the highest upregulated gene in this

pathway in Ly6G⁺ neutrophils of the lung as well as PB. This result is very interesting but rather surprising as ALK1 is an endothelial-specific receptor. This unexpected result needs to be supported by further experiments.

- First, the purity of the Ly6G⁺ cells enrichment (98%) should be checked by other means than just May-Giemsa staining. They should test some endothelial cell markers in this purification by PCR and /or IF, to check if they find any vascular enrichment.

>OK

- Second, they show that Acvr11 mRNA level in whole lung extract is also increased in JAK2V617F mice under hypoxia (Fig5a). As Acvr11 expression in the lungs is predominantly expressed by endothelial cells, it is rather surprising to observe such a high increase in Acvr11 that would be due to neutrophils. ALK1 levels are particularly high in lungs as this tissue contains a very high number of endothelial cells. It is interesting to note, although not discussed in the text, that Acvr11 levels in this figure is also increase in wt mice under hypoxia. The authors should isolate endothelial cells and neutrophils under normoxia and hypoxia (the RNAseq analysis was performed under normoxia) from the lungs of these mice and check in which cell type is Acvr11 increased. It would be interesting to compare the number of Acvr11 copies in neutrophils versus endothelial cells.

>OK

- The authors should comment on the difference between the 10-fold increase in Acvr11 mRNA levels versus the 2-fold increase in pSmad1/5 levels (Fig5a and b).

- The authors should measure the level of Acvr1 mRNA encoding the receptor Alk2 expression, which highly related to ALK1 and phosphorylates Smad1/5/8, and in contrast to ALK1 ubiquitous. A pSmad1/5/8 immunostaining in lungs would be interesting to discriminate neutrophils pSmad1/5/8 from endothelial pSmad1/5/8 staining.

>OK

Response;

Thank you for accepting the additional experiments we performed in Revision #1.

Comments;

3) To confirm ALK1 upregulation by JAK2V617F, they then use the commercial cell line HCT116 carrying the JAK2V617F mutation. HCT116 is colorectal carcinoma cell line that should not express Acvr11 so the choice of the cell line is rather surprising in particular when they want to study the promoter of this gene which is endothelial specific. Please comment. Again, it will be important to know the number of copies of Acvr11 in these cells as compared

to endothelial cells (Fig 6a). In order to check whether there is a significant level of active/functional ALK1 on these cells, the authors could stimulate these cells with BMP9, the high affinity ALK1 ligand.

>The authors have measured *Acvrl1* levels in HCT116 versus HPAEC and indeed found a 100-fold difference between in favour of endothelial cells supporting that the *Acvrl1* levels is low in these cells. They have stimulated the cells with BMP9 and found an increase in Smad1/5 phosphorylation but the concentrations of BMP9 used (50 and 100 ng/mL) are huge when compared to the EC₅₀ of BMP9 for ALK1 (50 pg/mL) and thus it cannot be concluded from this result that HCT116 cells respond to BMP9 via ALK1 as presented in their response but would support a role for ALK2. Lower doses of BMP9 should have been tested. SiRNA against ALK1 would have been the right approach to demonstrate that these cells express an active ALK1 receptor. From these results it cannot be excluded that what is observed could be due to ALK2.

Response;

Thank you very much for your important suggestions. Accordingly, we stimulated HCT116 cells with lower concentrations of BMP9 (12.5, 50, 200, 800, and 3200 pg/mL) in addition to 50 ng/mL. We found that phosphorylation levels of Smad1/5/8 were significantly increased in HCT116 cells in a dose dependent manner, and it is likely that 800 pg/mL was the concentration of BMP9 that gave the nearly maximal response. The BMP9 EC₅₀ in HCT116 cells was estimated to be 46.1 pg/mL. Next, to demonstrate that these cells express an active ALK1 receptor, we transfected HCT116 cells with ALK1-specific siRNA (siALK1). Decreases in *ACVRL1* mRNA and ALK1 protein levels in siALK1-transfected HCT116 cells were verified by RT-qPCR with 78% reduction and by Western blotting and flow cytometry analysis with about 60% reduction in comparison to the control siRNA-transfected cells (siCTRL). Then, we stimulated the ALK1-knocked down HCT116 cells with BMP9 (200 pg/mL). Phosphorylation levels of Smad1/5/8 in the siALK1-transfected cells were partly but significantly decreased compared to siCTRL after BMP9 stimulation. In the revised manuscript, we have replaced Supplementary Figure 23 with the new data.

According to the public database for *ACVRL1* expression in cell lines at the baseline, HCT116 cells express rather moderate *ACVRL1* levels. The Cell Lines Project (https://cancer.sanger.ac.uk/cell_lines) with the Affymetrix Human Genome U219 categorizes the HCT116 as one of the 918 cell lines with normally expressing *ACVRL1*, while there are 24 and 28 cell lines with under- and over-expressed *ACVRL1* in this database, respectively. HCT116 is ranked 203rd out of 798 cell lines in the *ACVRL1* expression levels in GSE68950 (GEO in NCBI database, Sanger cell line Affymetrix gene expression project by the Affymetrix U133A), which

we evaluated through CellExpress [<http://cellexpress.cgm.ntu.edu.tw>, Database (Oxford). 2018; 2018: bax101]. Additionally, at the tissue level, the colon presents relatively high expression levels of *ACVRL1* with the mean RPKM (reads per kilobase of transcript per million mapped reads) 18.4, which is not far behind from the lung with mean RPKM 28.5 (NCBI database, Mol Cell Proteomics 2014; 13: 397-406). Importantly, the expression levels of *ACVRL1* were functionally altered in the HCT116 cell line (Cell Death Dis 2019; 10: 895. Oncotarget 2016; 7: 22077-22091) as well as in clinical colorectal cancer cells (Clin Colorectal Cancer 2018;17: e471-e488) in recent studies. Taken together, these suggest that HCT116 cells express an active ALK1.

We examined the expressions of *ACVRL1* mRNA and ALK2 protein in *JAK2*^{V617F/+} HCT116 cells, demonstrating that both *ACVRL1* (ALK2) expression levels were not significantly different between *JAK2*^{V617F/+} and *JAK2*^{+/+} HCT116 cells (new Supplementary Figure 25). Also, we have shown that *Acvr1l* mRNA expression levels were significantly upregulated in the lung neutrophils in *JAK2*^{V617F} mice compared to WT mice, but *Acvr1* mRNA expression levels were not (Supplementary Figure 19). Our data from RNA sequencing revealed that *Acvr1l* levels in Ly6G⁺ cells in bone marrow and peripheral blood of *JAK2*^{V617F} mice were significantly upregulated (Figure 5), but *Acvr1* levels were similar (0.94-fold), compared to those of WT mice. Given that there were no increases in ALK2 expressions of *JAK2*^{V617F/+} HCT116 cells and *JAK2*^{V617F} Ly6G⁺ cells compared to WT cells, together with the results of siALK1 of HCT116 with BMP9 stimulation, we think the role of ALK2 in *JAK2*^{V617F}-associated pulmonary hypertension is less likely. However, as suggested by the reviewer, we cannot exclude the potential effects of ALK2 on PH. This point is discussed as a limitation in the Discussion section.

In the texts;

JAK2^{V617F/+} cells exhibited significant increases in the expression levels of *ACVRL1* mRNA as well as ALK1 protein and phosphorylation levels of Smad1/5/8 compared to *JAK2*^{+/+} cells (Fig. 7b, c, Supplementary Fig. 24), but not in *ACVRL1* (ALK2) expressions (Supplementary Fig. 25). (page 15, lines 347-351)

Although we cannot exclude the potential effects of ALK2 on PH, (page 24, lines 542-543)

Comments;

I am also septic about the ALK1 expression in HCT116 cells detected by western blot using an antibody from Abcam. First this antibody is not anymore commercially available may be due to specificity problems and second the detection of ALK1 by Western blot is already

very challenging in endothelial cells so it must really be difficult to detect in cells like HCT116 that express much less ALK1.

Response;

Firstly, the ALK1 antibody (ab37807) from Abcam had been commercially available when fixing our results and during the preparation of the manuscript. According to the official answer from Abcam, the vendor discontinued supplying Abcam with it for some commercial reasons, not for the specificity problem of the antibody, since January 2020. In line with the reviewer's suggestion, we have used the other two commercially available ALK1 antibodies. Western blotting demonstrated that siALK1 significantly decreased the protein levels of ALK1 using both ALK1 antibodies from Proteintech (14745-1-AP) and R&D systems (AF370) as shown below. Using the ALK1 antibodies, we showed that $JAK2^{V617F/+}$ HCT116 cells exhibited significant increases in the expression levels of ALK1 compared to $JAK2^{+/+}$ cells, which was consistent with our results we have shown previously. Flow cytometry analysis using the ALK1 antibody from Proteintech (14745-1-AP) confirmed increased levels in ALK1-positives in $JAK2^{V617F/+}$ HCT116 cells compared to $JAK2^{+/+}$ cells. We have replaced the original images of Western blotting and quantitative data with the new ones (Figure 7c), and added the data from flow cytometry in Supplemental Figure 24. Also, we have noted the ALK1 antibody information (14745-1-AP, Proteintech) in the Methods section (*page 28, line 643, page 31, lines 710-713*).

Comments;

4) The in-silico analysis identified several putative STAT3 binding sites in ACVRL1 promoter region in both humans and mice (Fig. 6d), that were not mentioned in the publication cited by the authors and only one STAT3 binding site is present in the putative promoter sequence from -1035 bp to +210 bp of the transcriptional start that the author studied. So, the involvement of STAT3 in the regulation of this promoter is not that strongly supported by in-silico data. Although the experiment with Ruxolitinib, a specific JAK1/2 inhibitor (Fig. 6f), supports a role for the JAK pathway in ACVRL1 promoter activity, no experiment was performed to conclude that it is STAT3 as mentioned in the discussion (lane 361). The authors should just conclude that ALK1 promoter activity in HCT116 cells seems to be JAK1/2 dependent.

>OK. The addition of the specific STAT3 inhibitor (stattic) comforts their hypothesis.

Response;

Thank you for your positive comments.

Comments;

5) Very interestingly, to support their ALK1 hypothesis, the authors performed some in vivo experiments in order to block ALK1 and test whether it would reverse their phenotype. For this, they used the inhibitor K02288. First, it is quite surprising to see no reference concerning this inhibitor in the manuscript and this should be corrected (Santivale et al. 2013, Plos One) and second, they do not mention that this inhibitor is not specific to ALK1 but also inhibits ALK2 and that this inhibitor is most of the time proposed as an ALK2 inhibitor with a slightly higher affinity for ALK2 than ALK1. This is why, it is important to measure ALK2 levels in neutrophils as ALK2 is a more widely expressed receptor. BMP2 levels which plays a much preponderant role than ALK1 in PH should also be measured. Together, from these experiments it can only be concluded that the ALK1/2 pathway is important for PH development in accordance with a recent paper that they cite (Li et al., Circ Res, 2019). It is however surprising that K02288 administration did not significantly change the levels of RVSP or RV hypertrophy in chronic hypoxia-exposed WT mice as recently published in the above cited paper. These experiments need to be performed using other ALK1 inhibitors and the effect of these inhibitors need to be also tested under normoxia.

>I appreciate the modification made in the text describing K02288 as an inhibitor of both ALK1 and ALK2 and adding the reference asked. However, they are still statements that do not specify this point, for example: Statements like “In the current study, inhibition of ALK1

prevented chronic hypoxia-induced PH as well as pulmonary arterial remodeling in JAK2V617F mice. (p29, line 484) and Thus, inhibition of ALK1, which was directly upregulated by JAK-STAT activation, may be effective especially in the lung neutrophils. (p32, line 537).

Response;

We thank again for your helpful comments. We have carefully reviewed and revised the manuscript to specify ALK1 and ALK2.

Comments;

They have also used another drug LDN-212854 but unfortunately as K02288 this drug inhibits both ALK1 and ALK2 but with a higher selectivity for ALK2 versus ALK1 (ref 31). This inhibitor gave the same result as K02288 (Supplementary Fig. 29) supporting their previous data. However, this inhibitor is not better than K02288 and does not allow to better conclude on the role of ALK1 versus ALK2.

Response;

In Revision #1, we performed further experiments according to the reviewer's suggestion that "these experiments need to be performed using other ALK1 inhibitors", although we had realized that ALK1 super-selective inhibitors were not commercially available and such ALK1 inhibitors were likely to cross-react ALK2. We agree with the reviewer that LDN-212854 might not be actually better than K02288 for selectivity toward ALK1 versus ALK2. Among the existing ALK1/2 inhibitors, LDN-212854 was the second most specific chemical inhibitor to ALK1 and was less likely to cross-react ALK2. Importantly, we observed that the treatment of LDN-212854 significantly decreased RVSP and RV/LV+S in chronic hypoxia-exposed JAK2V617F mice, similar to K02288. As we concur with the reviewer that the lack of specific ALK1 inhibitors was the limitation of the present study, a conditional ALK1 knockout model of hematopoietic cells in JAK2^{V617F} mice is needed to clarify the role of ALK1 on JAK2V617F-mediated PH in the future. Accordingly, we have toned down throughout the manuscript the conclusion that the PH phenotype is specifically mediated by ALK1 signaling. Again, we have carefully reviewed and revised the manuscript to specify ALK1 and ALK2.

Comments;

Together, although the authors have replied to many of my comments and have performed supplementary experiments, I still remain unsatisfied concerning their conclusion of a specific role of ALK1 in this work. Apart from the fact that Acvr11 expression is upregulated in neutrophils in JAK2V617mice and that Stat3 might regulate Acvr11 expression, I do not see a clear demonstration that clonal hematopoiesis with JAK2V617 causally leads to PH development through ALK1in this manuscript.

Response;

Thank you for your important suggestions. Accordingly, we have toned down throughout the manuscript the conclusion that the PH phenotype is specifically mediated by ALK1 signaling in neutrophils, starting from the title and the abstract.

Comments;

I was also not convinced by their response to reviewer 1 concerning their response to point 5: “Although it was reported that ACVRL1 mutations in hereditary hemorrhagic telangiectasia led to a loss of function (ref. #42, 43), the functional relevance of congenital ACVRL1 mutations in PH remains undetermined whether the mutations result in a gain of function or a loss of function. » This hypothesis is not convincing as ACVRL1 mutations found in PAH are the same mutations described in HHT which result in loss of function. I would thus be very surprized that in PAH these mutations would not lead to a loss of function.

Response;

We apologize for this description. We agree with the reviewer that ACVRL1 mutations found in PAH are the same mutations described in HHT which result in loss of function. The loss-of-function mutations in ACVRL1 are important causes of heritable PAH (N Engl J Med 2001; 345: 325–334, ref. #44). However, several germline ACVRL1 mutations have been reported in PAH patients which were not described in HHT database (Am J Respir Crit Care Med 2010; 181: 851–861. Genome Med 2021; 13: 80. bioRxiv, doi: <https://doi.org/10.1101/2020.05.29.124255>). Somatic mutations in ACVRL1 in HHT were reported (Am J Hum Genet 2019; 105: 894-906), but those in PH patients have not been fully investigated. Although the endothelial cell-specific deletion of ALK1 in mice led to vascular malformations mimicking the features of HHT (Blood 2008; 111: 633–642), its effect on PH was not fully determined. Heterozygous ALK1 knockout mice developed PH in adulthood (Cardiovasc Res 2011; 92: 375-384, ref. #45). In contrast, the inhibition of BMP-9 partly protected chronic hypoxia-induced PH in the adult mice and systemic

administration of ALK1 inhibitor, a ligand trap targeting ALK1, prevented the monocrotaline and Sugen hypoxia-induced PH in the adult rats (Circ Res 2019; 124: 846–855, ref. #46), suggesting that systemic blockade of the BMP-9/ALK1 pathways is beneficial for PH in the adult rodents. The discrepancy that the inhibition of ALK1 was beneficial or detrimental for PH has been on the debate (Circ Res 2019; 124: e81, Circ Res 2019; 124: e82–e83). In the present study, we showed that systemic administration of ALK1/2 inhibitors prevented the progression of chronic hypoxia-induced PH in JAK2^{V617F} mice, indicating that upregulation of ALK1 in hematopoietic cells with JAK2V617F had detrimental effects on PH. Since the role of ALK1 in PH may have a different impact in different cell types at different time points, the functional relevance of ALK1 in PH is not fully understood. ALK1 expressions in myeloid cells may have a different impact on PH from the lung endothelial cells. A conditional knockout model of hematopoietic cells is needed to clarify the role of ALK1 on PH in the hematopoietic system. We have revised the texts for further discussion in the Discussion section.

In the texts;

It has been reported that *ACVRL1* mutations in hereditary hemorrhagic telangiectasia led to a loss of function^{42,43}. Most of *ACVRL1* mutations found in pulmonary arterial hypertension are the same mutations described in HHT which result in a loss of function. The loss-of-function mutations in *ACVRL1* are important causes of heritable pulmonary arterial hypertension.⁴⁴ Consistently, heterozygous ALK1 knockout mice developed PH in adulthood⁴⁵. In contrast, the inhibition of BMP9 partly protected chronic hypoxia-induced PH in the adult mice and systemic administration of ALK1 inhibitor, a ligand trap targeting ALK1, prevented the monocrotaline and Sugen hypoxia-induced PH in the adult rats⁴⁶, suggesting that systemic blockade of the BMP9/ALK1 pathways is beneficial for PH in the adult rodents. In the present study, we showed that ALK1/2 inhibitor administration prevented the progression of chronic hypoxia-induced PH in JAK2^{V617F} mice, indicating that JAK2V617F-related ALK1 upregulation in myeloid cells had detrimental effects in PH. Although the molecular roles of ALK1 have been investigated particularly in endothelial cells, ALK1 expressions in myeloid cells may have a different impact on PH from the lung endothelial cells. As the functional relevance of ALK1 in PH is not fully understood, a conditional knockout model of hematopoietic cells is needed to clarify the role of ALK1 on PH in the hematopoietic system. (pages 21-22, lines 478-495)

Comments;

Minor points:

1) Lane 257: please indicate in the text what cell line is used.

>OK

2) Fig6e, please precise the cell type transfected in the figure legend.

>OK

Response;

We thank the reviewer for these comments.

Thank you very much again for your thoughtful comments to our manuscript.

To Reviewer #3:

Thank you very much for reviewing our manuscript (NCOMMS-20-18526A) and providing us with valuable comments. Our responses to your comments are as follows,

Comments;

The authors have thoroughly revised the manuscript and addressed almost all concerns except my previous question no. 8: “Please clarify how pulmonary arteries were distinguished from veins for the quantification of pulmonary vascular remodelling.” The anatomical difference between arteries and veins is not always clearly visible. This is especially the case with small vessels (diameter <50µm) that have been analyzed. Therefore, I would ask to refer in text to “pulmonary vessels” rather than to “pulmonary arteries”.

Response;

Thank you very much for your important suggestion. We have referred to the “pulmonary vessels” rather than to “pulmonary arteries” in the texts, when small vessels (diameter < 50 µm) were analyzed.

REVIEWERS' COMMENTS

Reviewer #2 (Remarks to the Author):

The authors have now addressed all the points raised by the reviewer.

To Reviewer #2:

Thank you very much for reviewing our revised manuscript (NCOMMS-20-18526B) and providing us with your comments. Our responses to your comments are as follows,

Comments;

The authors have now addressed all the points raised by the reviewer.

Response;

Thank you very much again for your positive comments.